**Role of a key microphysical factor in mixed-phase stratocumulus clouds and their**
**interactions with aerosols**
Seoung Soo Lee[1,2,3], Chang Hoon Jung[4], Jinho Choi[5], Young Jun Yoon[6], Junshik Um[5,7],
Youtong Zheng[8], Jianping Guo[9], Manguttathil. G. Manoj[10], Sang-Keun Song[11]
[1]Science and Technology Corporation, Hampton, Virginia
[2]Earth System Science Interdisciplinary Center, University of Maryland, College Park,
Maryland, USA
[3]Research Center for Climate Sciences, Pusan National University, Busan, Republic of
Korea
[4]Department of Health Management, Kyungin Women's University, Incheon, Republic of
Korea
[5]Department of Atmospheric Sciences, Pusan National University, Busan, Republic of
Korea
[6]Korea Polar Research Institute, Incheon, Republic of Korea
[7]Institute of Environmental Studies, Pusan National University, Busan, Republic of Korea
[8]Department of Earth and Atmospheric Sciences, University of Houston, Houston, Texas,
USA
[9]State Key Laboratory of Severe Weather, Chinese Academy of Meteorological Sciences,
Beijing 100081, China
[10]Advanced Centre for Atmospheric Radar Research, Cochin University of Science and
Technology, Kerala, India

[11]Department of Earth and Marine Sciences, Jeju National University, Jeju, Republic of

Korea

Corresponding author: Seoung Soo Lee, Chang Hoon Jung and Sang-Keun Song

Office: (303) 497-6615

Cell: (609) 375-6685

Fax: (303) 497-5318

E-mail: cumulss@gmail.com, slee1247@umd.edu

**Abstract**
This study examines the ratio of ice crystal number concentration (ICNC) to cloud droplet
number concentration (CDNC), which is ICNC/CDNC, in mixed-phase stratocumulus
clouds. This examination is performed using a large-eddy simulation (LES) framework and
one of efforts toward a more general understanding of mechanisms controlling cloud
development, aerosol-cloud interactions and impacts of ice processes on them in mixed-
phase stratocumulus clouds. For the examination, this study compares a case of polar
mixed-phase stratocumulus clouds to that of midlatitude mixed-phase stratocumulus
clouds with weak precipitation. It is found that ICNC/CDNC plays a critical role in making
differences in cloud development with respect to the relative proportion of liquid and ice
mass between the cases by affecting in-cloud latent-heat processes. Note that this
proportion has an important implication for cloud radiative properties and thus climate. It
is also found that ICNC/CDNC plays a critical role in making differences in interactions
between clouds and aerosols and impacts of ice processes on clouds and their interactions
with aerosols between the cases by affecting in-cloud latent-heat processes. Findings of
this study suggest that ICNC/CDNC can be a simplified general factor that contributes to
a more general understanding and parameterizations of mixed-phase clouds, their
interactions with aerosols and roles of ice processes in them.












**1. Introduction**

Stratiform clouds (e.g., stratus and stratocumulus clouds) have significant impacts on
climate (Warren et al. 1986; Stephens and Greenwald 1991; Hartmann et al. 1992; Hahn
and Warren 2007; Wood, 2012; Dione et al., 2019; Zheng et al., 2021). Since
industrialization, aerosol concentrations have increased and this has had impacts on
stratiform clouds and climate (Twomey, 1974; Albrecht, 1989; Ackerman et al., 2004).
However, our level of understanding of these clouds and impacts has been low and this has
caused the highest uncertainty in the prediction of future climate (Ramaswamy et al., 2001;
Forster et al., 2007; Knippertz et al., 2011; Hannak et al., 2017). Stratiform clouds can be
classified into warm and mixed-phase clouds. Mixed-phase stratiform clouds involve ice
processes and frequently form in midlatitude and polar regions. When mixed-phase clouds
are associated with convective clouds, they can form even in the tropical region. Most
previous studies have focused on warm clouds and their interactions with aerosols, whereas
the mixed-phase stratiform clouds and their interactions with aerosols are poorly
understood mainly due to the more complex ice processes. Hence, mixed-phase stratiform
clouds and their interactions with aerosols account for the uncertainty more than warm
clouds and their interactions with aerosols (Ramaswamy et al., 2001; Forster et al., 2007;
Wood, 2012; IPCC, 2021; Li et al., 2022).
The relative proportion of liquid mass, which can be represented by liquid-water
content (LWC) or liquid-water path (LWP), and ice mass, which can be represented by ice-
water content (IWC) or ice-water path (IWP), in mixed-phase stratiform clouds plays a
critical role in cloud radiative properties and thus their climate feedbacks (Tsushima et
al., 2006; Choi et al., 2010 and 2014; Gettelman et al., 2012; Zhang et al., 2019). The
relative proportion is defined to be IWC (IWP) over LWC (LWP) or IWC/LWC
(IWP/LWP) in this study. Motivated by this and the above-mentioned uncertainty, this
study aims to improve our understanding of mixed-phase stratiform clouds and their
interactions with aerosols with the emphasis on ice processes and IWC/LWC (or
IWP/LWP).
Lee et al. (2021) have investigated mixed-phase stratocumulus clouds in a midlatitude
region and found that microphysical latent-heat processes are more important in the
development of mixed-phase stratiform clouds and their interactions with aerosols than
entrainment and sedimentation processes. Lee et al. (2021) have found that a microphysical
factor, the ratio of ice crystal number concentration (ICNC) to cloud droplet number
concentration (CDNC) or ICNC/CDNC, play an important role in latent processes, the
development of mixed-phase stratiform clouds and their interactions with aerosols. In
particular, Lee et al. (2021) have found that IWC/LWC or IWP/LWP is strongly affected
by ICNC/CDNC. This is because water vapor deposits on the surface of ice crystals, while
it condenses on droplets. As a result, ice crystals act as sources of deposition and droplets
act as sources of condensation. Consequently, ice crystals act as sources of IWC (or IWP)
and droplets act as sources of LWC (or LWP). More ice crystals and droplets provide the
greater integrated surface area of ice crystals and droplets and induce more deposition and
condensation, respectively, for a given environmental condition (Lee et al., 2009; Khain et
al., 2012; Fan et al., 2018; Chua and Ming, 2020; Lee et al., 2021). The higher
ICNC/CDNC means more ice crystals or sources of deposition per a droplet as a source of
condensation in a given group of ice crystals and droplets. Thus, the higher ICNC/CDNC
enables more deposition per unit condensation to occur, which can raise IWC/LWC or
IWP/LWP.
Mixed-phase stratocumulus clouds in different regions are known to have different
IWC/LWC or IWP/LWP and aerosol-cloud interactions (e.g., Choi et al., 2010 and 2014;
Zhang et al., 2019). Lots of factors such as environmental conditions, which can be
represented by variables such as temperature, humidity and wind shear, and macrophysical
factors one of which is the relative locations of ice-crystal and droplet layers, can explain
those differences. Choi et al. (2010 and 2014) and Zhang et al. (2019) have shown that as
temperature lowers, IWC/LWC or IWP/LWP tends to increase and indicated that
temperature is a primary environmental condition to explain the differences in IWC/LWC
among different regions or clouds. However, Choi et al. (2010 and 2014) and Zhang et al.
(2019) have not discussed process-level mechanisms that govern the role of temperature in
those differences.
It is important to establish a general principle that explains the differences in
LWC/LWC and aerosol-cloud interactions among regions, since the general principle is
useful in the development of a more general or comprehensive parameterization of
stratocumulus clouds and their interactions with aerosols for climate models. This
contributes to the better prediction of future climate, considering that the absence of the
comprehensive parameterization has been considered one of the biggest obstacles to the
better prediction (Ramaswamy et al., 2001; Foster et al., 2007; Stevens and Feingold, 2009).
As a way of contributing to the establishment of the general principle, this study
attempts to take ICNC/CDNC as a general factor, which can constitute the general principle,
to explain the differences in IWC/LWC (or IWP/LWP) and aerosol-cloud interactions
among clouds. This study also attempts to elucidate how ice processes differentiate mixed-
phase stratiform clouds from warm clouds in terms of cloud development and its
interactions with aerosols, and how this differentiation varies among cases of mixed-phase
stratiform clouds with different ICNC/CDNC values. This attempt is valuable, considering
that in general, the establishment of the general principle for stratocumulus clouds and their
interactions with aerosols has been progressed much less than that for other types of clouds
such as convective clouds and their interactions with aerosols. The attempt is valuable, also
considering that our level of understanding of how ice processes differentiate mixed-phase
stratiform clouds and their interactions with aerosols from much-studied warm clouds and
their interactions with aerosols has been low. Here, we want to emphasize that this study
does not aim to gain a fully established general principle, but aims to test the factor that
can be useful to move ahead on our path to a more complete general principle. Hence, this
study should be regarded a steppingstone to the established principle, and should not be
considered a perfect study that get us the fully established principle. Taking into account
the fact that even attempts to provide general factors for the general principle have been
rare, the fulfilment of the aim is likely to provide us with valuable preliminary information
that streamlines the development of a more established general principle.
For the attempt, this study investigates a case of mixed-phase stratiform clouds in the
polar region. Via the investigation, this study aims to identify process-level mechanisms
that control the development of those clouds and their interactions with aerosols, and the
impact of ice processes on the development and interactions using a large-eddy simulation
(LES) framework. Then, this study compares the mechanisms in the case of polar clouds
to those in a case of midlatitude clouds which have been examined by Lee et al. (2021).
This comparison is based on Choi et al. (2010 and 2014) and Zhang et al. (2019) which

178 have shown that temperature is an important factor which explains the differences in

179 IWC/LWC among regions or clouds. Due to significant differences in latitudes, noticeable

180 differences in the temperature of air are between the polar and midlatitude cases. Hence,

181 through this comparison, this study looks at the role of temperature in those differences in

182 IWC/LWC and associated aerosol-cloud interactions. More importantly than that, as a way

183 of identifying process-level mechanisms that control the role of temperature, this study

184 tests how ICNC/CDNC as the general factor is linked to the role of temperature, using the

185 LES framework. Through this test, this study also identifies process-level mechanisms that

186 control how ICNC/CDNC affects roles of ice processes in the differentiation between

187 mixed-phase stratiform and warm clouds in terms of cloud development and its interactions

188 with aerosols, and causes the variation of the differentiation between the cases of mixed-

189 phase stratiform clouds.

190

191 **2. Case, model and simulations**

192

193 **2.1 LES model**

194

195 LES simulations are performed by using the Advanced Research Weather Research and

196 Forecasting (ARW) model. A bin scheme, which is detailed in Khain et al. (2000) and

197 Khain et al. (2011), is adopted by the ARW for the simulation of microphysics. Size

198 distribution functions for each class of hydrometeors, which are classified into water drops,

199 ice crystals (plate, columnar and branch types), snow aggregates, graupel and hail, are

200 represented with 33 mass doubling bins, i.e., the mass of a particle $m_k$ in the kth bin is

201 determined as $m_k = 2m_{k-1}$. Each of hydrometeors has its own terminal velocity that varies

202 with the hydrometeor mass and the sedimentation of hydrometeors is simulated using their

203 terminal velocity.

204  Size distribution functions for aerosols, which act as cloud condensation nuclei

205 (CCN) and ice-nucleating particles (INP), adopt the same mass doubling bins as for

206 hydrometeors. The evolution of aerosol size distribution and associated aerosol

207 concentrations at each grid point is controlled by aerosol sinks and sources such as aerosol

208 advection, turbulent mixing, activation and aerosol regeneration via the evaporation of

droplets and the sublimation of ice crystals. Aerosol regeneration follows the method
similar to that as described in Xue et al. (2010). It is assumed that aerosols do not fall down
by themselves and move around by airflow that is composed of horizontal flow, updrafts,
downdrafts and turbulent motions. When aerosols move with airflow, it is assumed that
they move with the same velocity as airflow. Taking activation as an example of the
evolution of aerosol size distribution, the bins of the aerosol spectra that correspond to
activated particles are emptied. Activated aerosol particles are included in hydrometeors
and move to different classes and sizes of hydrometeors through collision-coalescence. In
case hydrometeors with aerosol particles precipitate to the surface, those particles are
removed from the atmosphere.
The large energetic turbulent eddies are directly resolved by the LES framework, and
the effects of the smaller subgrid-scale turbulent motions on the resolved flow are
parameterized based on a most widely used method that Smagorinsky (1963) and Lilly
(1967) proposed. In this method, the mixing time scale is defined to be the norm of the
strain rate tensor (Bartosiewicz and Duponcheel, 2018). A cloud-droplet nucleation
parameterization based on Köhler theory represents cloud-droplet nucleation. Arbitrary
aerosol mixing states and aerosol size distributions can be fed to this parameterization. To
represent heterogeneous ice-crystal nucleation, the parameterizations by Lohmann and
Diehl (2006) and Möhler et al. (2006) are used. In these parameterizations, contact,
immersion, condensation-freezing, and deposition nucleation paths are all considered by
taking into account the size distribution of INP, temperature and supersaturation.
Homogeneous aerosol (or haze particle) and droplet freezing is
also considered following the theory developed by Koop et al. (2000).
The bin microphysics scheme is couped to the Rapid Radiation Transfer Model
(RRTM; Mlawer et al., 1997). The effective sizes of hydrometeors, which are calculated
in the bin scheme, are fed into the RRTM as a way of considering effects of the effective
sizes on radiation. The surface process and resultant surface heat fluxes are simulated by
the interactive Noah land surface model (Chen and Dudhia, 2001).

**2.2 Case and simulations**

**2.2.1 Case and standard simulations**


In the Svalbard area, Norway, a system of mixed-phase stratocumulus clouds existed over
the horizontal domain marked by a red rectangle in Figure 1 and a period between 02:00
and 10:00 local solar time (LST) on March 29[th], 2017. These clouds are observed by a
ground station which is a part of the Cloudnet observation network and marked by a dot in
Figure 1. The Cloudnet observation has been established to provide a systematic evaluation
of clouds in forecast and climate models. The Cloudnet observation aims to establish a
number of ground-based remote sensing sites, which would all be equipped with a specific
array of instrumentation, using sensors such as radiometer, lidar and Dopplerized mm-
wave radar, in order to provide vertical profiles of the main cloud variables (e.g., LWC and
IWC) (Hogan et al., 2006). In the Cloudnet observation, particularly, LWC is measured by
radiometer with a spatial resolution of ~50 m in the vertical direction and a temporal
resolution of 30 seconds. The retrieval of IWC is performed by using radar reflectivity and
lidar backscatter in the Cloudnet observation with a spatial resolution of ~10 m in the
vertical direction and a temporal resolution of 30 seconds as described in Donovan et al.
(2001), Donovan and Lammeren (2001), Donovan (2003) and Tinel et al. (2005). In the
retrieval, the lidar signal and radar reflectivity profiles are combined and inverted using a
combined lidar/radar equation as a function of the light extinction coefficient and radar
reflectivity. The combined equation is detailed in Donovan and Lammeren (2001). In the
Cloudnet data, LWC data with the coarser spatial resolution than IWC data are interpolated
to observation locations of IWC data, and IWP and LWP data are obtained from these IWC
and interpolated LWC data, respectively. The Cloudnet observation data including these
IWC, LWC, IWP and LWP data are provided to the public with a temporal resolution of
30 seconds in a continuous manner. This study utilizes these publicized Cloudnet data.

On average, the bottom and top of the observed clouds, which are measured by radar

and lidar in the Cloudnet observation, are at ~400 m and ~3 km in altitude, respectively.
The simulation of the observed system or case, i.e., the control run, is performed three-
dimensionally over the red rectangle and the period between 02:00 and 10:00 LST on
March 29[th], 2017. The horizontal domain adopts a100-m resolution for the control run. The
length of the domain in the horizontal directions is 50 km. The length of the domain in the
vertical direction is ~5 km and the resolution for the vertical domain gets coarsened with
height from ~5 m just above the surface to ~150 m at the model top as detailed in the
supplement. Reanalysis data, which are produced by Met Office Unified Model (Brown et
al., 2012) every 6 hours on a 0.11° × 0.11° grid, provide potential temperature, specific
humidity, and wind as initial and boundary conditions, which represent synoptic-scale
environment, for the control run. The control run employs an open lateral boundary
condition. Figure 2a shows the vertical distribution of the domain-averaged potential
temperature and humidity in those reanalysis data at the first time step. A neutral, mixed
layer is between the surface and 1 km in altitude as an initial condition (Figure 2a). Figure
2b shows the time evolution of the domain-averaged large-scale subsidence or downdraft
in the reanalysis data and at the model top. This large-scale subsidence is imposed on the
control run as a part of background wind fields and interacts with updrafts and downdrafts
generated by relatively small-scale processes including those associated with clouds. The
large-scale subsidence gradually reduces with time (Figure 2b). Figure 2c shows the time
evolution of the domain-averaged surface temperature in the reanalysis data. This evolution
of the surface temperature is strongly controlled by the sea surface temperature considering
that a large portion of the red-rectangle domain is accounted for by the ocean (Figure 1).
Due to the sunrise, the surface temperature starts to increase more rapidly around 08:00
LST (Figure 2c).

The properties of cloud condensation nuclei (CCN) such as the number concentration,

size distribution and composition are measured in the domain (Tunved et al., 2013; Jung et
al., 2018). The measurement of the CCN concentration has been carried out at the location
marked by a dot in Figure 1, using the commercial droplet measurement technologies CCN
counter with one column (CCNC-100), managed by the Korea Polar Research Institute,
since year 2007. The CCNC-100 measures the CCN concentration at supersaturations of
0.2, 0.4, 0.6, 0.8 and 1% (Jung et al., 2018). The aerosol number size distribution is
observed using a closed-loop differential mobility particle sizer (DMPS). The DMPS
charges aerosol particles and exposing them into an electric field, which causes them to
experience a force proportional to their electrical mobility, resulting in their classification
according to size (Tunved et al., 2013). Aerosol composition is measured using aerosol
mass spectrometry (AMS). The AMS measures the composition by vaporizing and ionizing
aerosol particles.
The measurement indicates that on average, aerosol particles are an internal mixture
of 70 % ammonium sulfate and 30 % organic compound. This mixture is assumed to
represent aerosol chemical composition over the whole domain and simulation period for
this study. The observed and averaged concentration of aerosols acting as CCN is ~200
cm$^{-3}$ over the simulation period between 02:00 and 10:00 LST on March 29[th], 2017. Note
that the average of a variable with respect to time in the rest of this paper is performed over
this period between 02:00 and 10:00 LST, unless otherwise stated. 200 cm$^{-3}$ as the averaged
concentration of aerosols acting as CCN is interpolated into all of grid points immediately
above the surface at the first time step.
This study does not take into account aerosol effects on radiation before aerosol is
activated, since no significant amount of radiation absorbers is found in the mixture. Based
on observation, the size distribution of aerosols acting as CCN is assumed to be a tri-modal
log-normal distribution (Figure 3). The shape of distribution, which is a tri-modal log-
normal distribution, as shown in Figure 3 is applied to the size distribution of aerosols
acting as CCN in all parts of the domain during the whole simulation period. The assumed
shape in Figure 3 is obtained by performing the average on the observed size distribution
parameters (i.e., modal radius and standard deviation of each of nuclei, accumulation and
coarse modes, and the partition of aerosol number among those modes) over the simulation
period. Note that although these parameters or the shape of aerosol size distribution does
not vary, associated aerosol concentrations vary over the simulation domain and period via
processes as described in Section 2.1. This study takes an assumption that the interpolated
CCN concentrations do not vary with height in a layer between the surface and the
planetary boundary layer (PBL) top around 1 km in altitude at the first time step, following
the previous studies such as Gras (1991), Jaenicke (1993) and Seinfeld and Pandis (1998).
However, above the PBL top, they are assumed to decrease exponentially with height at
the first time step, based on those previous studies, although the shape of size distribution
and composition do not change with height. It is assumed that the properties of INP and
CCN are not different except for concentrations. The concentration of aerosols acting as
CCN is assumed to be 100 times higher than that acting as INP over grid points at the first
time step based on a general difference in concentrations between CCN and INP
(Pruppacher and Klett, 1978). Hence, the concentration of aerosols acting as INP at the
first time step is 2 cm$^{-3}$ in the control run. This assumed concentration of aerosols acting
as INP is higher than usual (Seinfeld and Pandis, 1998). However, Hartmann et al. (2021)
observed the INP concentration that was at the same order of magnitude as assumed here
in the Svalbard area when strong dust events occur, meaning that the assumed INP
concentration is not that unrealistic.
To examine effects of aerosols on mixed-phase clouds, the control run is repeated by
increasing the concentration of aerosols by a factor of 10. In the repeated (control) run, the
initial concentrations of aerosols acting as CCN and INP at grid points immediately above
the surface are 2000 (200) and 20 (2) cm$^{-3}$, respectively. Reflecting these concentrations in
the simulation name, the control run is referred to as "the 200_2 run" and the repeated run
is referred to as "the 2000_20 run". To isolate effects of aerosols acting as CCN (INP) on
mixed-phase clouds, the control run is repeated again by increasing the concentration of
aerosols acting as CCN (INP) only but not INP (CCN) by a factor of 10. In this repeated
run with the increase in the concentration of aerosols acting as CCN (INP), the initial
concentrations of aerosols acting as CCN and INP at grid points immediately above the
surface are 2000 (200) and 2 (20) cm$^{-3}$, respectively. Reflecting this, the repeated run is
referred to as "the 2000_2 (200_20) run".

**2.2.2 Additional simulations**

To isolate impacts of ice processes on the adopted case and its interactions with aerosols,
the 200_2 and 2000_2 runs are repeated by removing ice processes. These repeated runs
are referred to as the 200_0 and 2000_0 runs. In the 200_0 and 2000_0 runs, all
hydrometeors (i.e., ice crystals, snow, graupel, and hail), phase transitions (e.g., deposition
and sublimation) and aerosols (i.e., INP) which are associated with ice processes are
removed. Hence, in these runs, only droplets (i.e., cloud liquid), raindrops, associated phase
transitions (e.g., condensation and evaporation) and aerosols acting as CCN are present,
regardless of temperature. Stated differently, these noice runs simulate the warm-cloud
counterpart of the selected mixed-phase cloud system. Via comparisons between a pair of
the 200_2 and 2000_2 runs and a pair of the 200_0 and 2000_0 runs, the role of ice
processes in the differentiation between mixed-phase and warm clouds is to be identified.
Along with this identification, the role of the interplay between ice crystals and droplets in
the development of the selected mixed-phase cloud system and its interactions with
aerosols is to be isolated.

As detailed in Sections 3.1.4 and 3.2.2 below, the test of ICNC/CDNC as a general

factor requires more simulations to see impacts of ICNCavg/CDCNavg on clouds and their
interactions with aerosols. Here, ICNCavg and CDNCavg represent the average ICNC and
CDNC over grid points and time steps with non-zero ICNC and CDNC, respectively.
ICNCavg/CDNCavg represents overall ICNC/CDNC over the domain and simulation
period. To respond to this requirement, the 200_0.07, 2000_0.07 and 200_0.7 runs are
performed and their details are given in Sections 3.1.4 and 3.2.2. In addition, all the
simulations above are repeated by turning off radiative processes and Section 3.3 provides
the details of these repeated simulations. These repeated runs are the 200_2_norad,
2000_20_norad, 2000_2_norad, 200_20_norad, 200_0_norad, 2000_0_norad,
200_0.07_norad, 2000_0.07_norad and 200_0.7_norad runs. Moreover, based on the
argument in Section 4.2, the 4000_45, 13_0.1, 4000_1.8 and 12_0.0035 runs are performed
and details of these runs are provided in Section 4.2. Some of the simulations are
summarized in Table 1 for better clarification with a brief description of their configuration.

**3.   Results**

**3.1 The 200_2 run vs. the 200_0 run**


**3.1.1   Model validation**


This study adopts the Cloudnet observation, which has been used to assess cloud
simulations as in Illingworth et al. (2007) and Hansen et al. (2018), to evaluate the 200_2
run. Simulated LWP and IWP, as shown in Figure 4 and Table 2, are compared to the
observed LWP and retrieved IWP in the Cloudnet data, respectively. The average LWP
over all time steps and grid columns for the period between 02:00 and 10:00 LST on March
29$^{th}$, 2017 is 1.23 g m$^{-2}$ in the 200_2 run and 1.12 g m$^{-2}$ in the Cloudnet observation. The
average IWP over all time steps and grid columns over the period is 31.94 g m$^{-2}$ in the
200_2 run and 29.10 g m$^{-2}$ in the retrieval. Cloud-bottom height, which is averaged over
grid columns and time steps with non-zero cloud-bottom height over the period, is 420 m
in the 200_2 run and 440 m in the Cloudnet observation. Cloud-top height, which is
averaged over grid columns and time steps with non-zero cloud-top height over the period,
is 3.5 km in the 200_2 run  and 3.3 km in the Cloudnet observation. Each of LWP, cloud-
bottom and -top heights shows an ~10% difference between the 200_2 run and observation.
IWP also shows an ~10% difference between the 200_2 run and the retrieval. Thus, the
200_2 run is considered performed reasonably well for these variables.
To provide additional information of cloud development, Figure 5 shows the time
evolution of the simulated and observed cloud-top and bottom heights, simulated and
retrieved IWP and simulated and observed LWP together with the evolution of the
simulated surface sensible and latent-heat fluxes; the simulated evolutions in Figure 5 are
from the 200_2 run. This is based on the fact that the cloud-top and bottom heights, IWP
and LWP are considered a good indicative of cloud development and the surface fluxes are
considered important parameters controlling the overall development of clouds. The cloud-
top height increases between 02:00 and ~05:00 LST and after ~05:00 LST, it reduces
gradually. The cloud-bottom height decreases between 02:00 and ~05:00 LST and after
~05:00 LST, it does not change much. IWP and LWP show an overall increase between
02:00 and ~05:30 LST to reach its peak around 05:30 LST and then an overall decrease.
The surface fluxes reduce with time, although the reduction rate of the fluxes starts to
decrease around 08:00 LST in association with the rapid increase in the surface temperature
which starts around 08:00 LST as shown in Figure 2c.
The time- and domain-averaged IWP is ~one order of magnitude greater than LWP, and
the time- and domain-averaged IWC is ~one order of magnitude greater than LWC in the
200_2 run (Figure 4 and Table 2). For the sake of simplicity, the averaged IWC over the
averaged LWC is denoted by IWC/LWC, and the averaged IWP over the averaged LWP
is by IWP/LWP, henceforth.  IWC/LWC is 26.28 and IWP/LWP is 25.96 in the 200_2 run.
Since IWP and LWP are vertically integrated IWC and LWC over the vertical domain,
respectively, the qualitative nature of differences between IWC and LWC is not much

425 different from that between IWP and LWP. Hence, mentioning both a pair of IWC and

426 LWC and that of IWP and LWP is considered redundant, and mentioning either a pair of

427 IWC and LWC or that of IWP and LWP enhances the readability. Henceforth, IWC and

428 LWC are chosen to be mentioned in text, although all of IWC, LWC, IWP and LWP are

429 displayed in Tables 2 and 3.

430  Choi et al. (2014) and Zhang et al. (2019) have obtained the supercooled cloud fraction

431 (SCF), which is basically the ratio of LWC to the sum of LWC and IWC and denoted by

432 LWC/(LWC+IWC), using satellite- and ground-observed data collected over the period of

433 ~1 year to ~5 years. Choi et al. (2014) have shown that SCF is as low as ~0.01 for the

434 temperature range between -16 and -33 ℃. Zhang et al. (2019) have also shown that SCF

435 is as low as ~0.03 for the same temperature range, although the occurrence of SCF of ~0.03

436 or lower is rare. Note that the average air temperature immediately below the cloud base

437 and above the cloud top over the simulation period is -16 and -33 ℃, respectively, in the

438 200_2 run, and SCF in the 200_2 run is 0.04. Hence, based on Choi et al. (2014) and Zhang

439 et al. (2019), we believe that SCF in the 200_2 run is observable and thus not that

440 unrealistic, although it may not occur frequently.

442 **3.1.2 Microphysical processes, sedimentation and entrainment**

444 To understand process-level mechanisms that control the results, microphysical processes

445 are analyzed. As indicated by Ovchinnikov et al. (2011), in clouds with weak precipitation,

446 a high-degree correlation is found between IWC and deposition or between LWC and

447 condensation, considering that deposition is the source of IWC and condensation is the

448 source of LWC. In the 200_2 run, the average surface precipitation rate over the simulation

449 period is ~0.0020 mm hr$^{-1}$, which can be considered weak. Hence, in this case,

450 condensation is considered a proxy for LWC, and deposition is a proxy for IWC. Based on

451 this, to gain a process-level understanding of microphysical processes that control the

452 simulated LWC and IWC, condensation and deposition are analyzed.

453  As seen in Figure 6 and Table 2, the average deposition rate is ~one order of magnitude

454 greater than condensation rate in the 200_2 run, leading to much greater IWC than LWC

455 in the 200_2 run. This is in contrast to the situation in the case of mixed-phase

stratocumulus clouds, which were located in a midlatitude region, in Lee et al. (2021). In
that case, the average IWC and LWC are at the same order of magnitude. For the sake of
brevity, the case in Lee et al. (2021) is referred to as "the midlatitude case", while the case
of mixed-phase clouds, which is adopted by this study, in the Svalbard area is referred to
"the polar case", henceforth. In the midlatitude case, IWC/LWC is 1.55, which is ~ one
order of magnitude smaller than that in the polar case.
Warm clouds in the 200_0 run shows that the time- and domain-averaged condensation
rate that is lower than the time- and the domain-averaged sum of condensation and
deposition rates in the 200_2 run (Figure 6 and Table 2). This leads to a situation where
warm clouds in the 200_0 run shows the time- and domain-averaged LWC that is lower
than the time- and domain-averaged water content (WC), which is the sum of IWC and
LWC, in mixed-phase clouds in the 200_2 run (Figure 4 and Table 2). This is despite the
fact that LWC in the 200_0 run is higher than LWC in the 200_2 run (Figure 4 and Table
2); WC represents the total cloud mass in mixed-phase clouds, while LWC alone represents
the total cloud mass in warm clouds.
It should be noted that the average rate of sedimentation of droplets over the cloud
base and simulation period reduces from the 200_0 run to the 200_2 run (Table 2). This is
mainly due to the decrease in LWC from the 200_0 run to the 200_2 run. The average rate
of sedimentation of ice crystals over the cloud base and simulation period increases from
the 200_0 run to the 200_2 run, since sedimentation of ice crystals is absent in the 200_0
run (Table 2). The average entrainment rate over the cloud top and simulation period
increases from the 200_0 run to the 200_2 run (Table 2). Here, entrainment rate is defined
to be the difference between the rate of increase in cloud-top height and the large-scale
subsidence, following Moeng et al. (1999), Jiang et al. (2002), Stevens et al. (2003a and
2003b) and Ackerman et al. (2004). Entrainment tends to reduce the total cloud mass more
in the 200_2 run than in the 200_0 run. Thus, entrainment should be opted out when it
comes to mechanisms leading to the increase in the total cloud mass from the 200_0 run to
the 200_2 run. Here, the vertical integration of each of condensation and deposition rates
is obtained over each cloudy column in the domain for each of the runs. For the sake of the
brevity, this vertical integrations of condensation and deposition rates are referred to as the
integrated condensation and deposition rates, respectively. Then, each of the integrated
condensation and deposition rates is averaged over cloudy columns and the simulation
period. It is found that the average rate of the droplet sedimentation over the cloud base
and simulation period is ~four orders of magnitude smaller than the average integrated
condensation rate in the 200_2 run (Table 2). The average rate of the ice-crystal
sedimentation over the cloud base and simulation period is ~four orders of magnitude
smaller than the average integrated deposition rate in the 200_2 run (Table 2). It is also
found that the average rate of the droplet sedimentation over the cloud base and simulation
period is ~five orders of magnitude smaller than that in the average integrated condensation
rate in the 200_0 run (Table 2). Changes in the average rate of the droplet sedimentation
over the cloud base and simulation period are ~four to five orders of magnitude smaller
than those in the average integrated condensation rate between the 200_2 and 200_0 runs
(Table 2). Changes in the average rate of the ice-crystal sedimentation over the cloud base
and simulation period are ~four to five orders of magnitude smaller than those in the
average integrated deposition rate between the 200_2 and 200_0 runs (Table 2). Thus,
condensation and deposition, but not the droplet and ice-crystal sedimentation, are main
factors controlling cloud mass, which is represented by LWC and IWC, and the total cloud
mass in the 200_2 and 200_0 runs.  The variation of cloud mass and the total cloud mass
between the runs are also mainly controlled by condensation and deposition, but not by
droplet and ice-crystal sedimentation. These dominant roles of condensation and
deposition over those of droplet and ice-crystal sedimentation are observed in the
midlatitude case and its warm-cloud counterpart as well.

### 3.1.3   Hypothesis

We hypothesized that ICNC/CDNC can be an important factor that determines above-
described differences between the polar and midlatitude cases. Note that both in the polar
and midlatitude cases, pockets of ice particles and those of liquid particles are mixed
together instead of being separated from each other as seen in Figure 4 and Lee et al. (2021).
Remember that ice crystals are more as sources of deposition per a droplet when
ICNC/CDNC is higher. Thus, as ICNC/CDNC increases in a situation where $q_v > q_{sw}$, it
is likely that the portion of water vapor, which is deposited onto ice crystals, increases.

518 This is by stealing water vapor, which is supposed to be condensed onto droplets, from

519 droplets in an air parcel. Here, qv and qsw represent water-vapor pressure and water-vapor

520 saturation pressure for liquid water or droplets, respectively. As ICNC/CDNC increases in

521 a situation where qsi< qv <qsw, the number of ice crystals, which absorb water vapor,

522 increases per a droplet; here, water vapor absorbed by ice crystals includes that which is

523 produced by droplet evaporation, and qsi represents water-vapor saturation pressure for ice

524 water or ice crystals. Thus, as ICNC/CDNC increases, it is likely that the portion of water

525 vapor, which is deposited onto ice crystals in an air parcel, increases as shown in Lee et al.

526 (2021). This is aided by the higher capacitance of ice crystals than that of droplets

527 (Pruppacher and Klett, 1978). Figure 7 shows the time series of the averaged

528 supersaturation over gird points where deposition occurs in the presence of both droplets

529 and ice crystals in the 200_2 run. Figure 7 indicates that on average, supersaturation occurs

530 for both droplets and ice crystals over those grid points. Hence, on average, the above-

531 described situation of qv > qsw is applicable to deposition when droplets and ice crystals

532 coexist in the 200_2 run.

533 ICNCavg/CDNCavg is 0.22 in the control run (i.e., the 200_2 run) for the polar case

534 and 0.019 in the control run for the midlatitude case which is described in Lee et al. (2021).

535 Henceforth, the control run for the midlatitude case is referred to as the control-midlatitude

536 run. ICNCavg/CDNCavg is ~one order of magnitude higher for the polar case than for the

537 midlatitude case. This is despite the fact that the ratio of the initial number concentration

538 of aerosols acting as INP to that of acting as CCN is identical between the 200_2 and

539 control-midlatitude runs. In addition, identical model, model setup such as vertical

540 resolutions, and source of reanalysis data are used between the 200_2 and control-

541 midlatitude runs. However, there are differences in environmental conditions (e.g.,

542 temperature), cloud macrophysical variables such as cloud-top height and horizontal

543 resolutions between the runs. Here, while taking these similarities and differences into

544 account, we hypothesize that the significant differences in ICNCavg/CDNCavg between

545 runs are mainly due to the fact that ice nucleation strongly depends on air temperature

546 (Prappucher and Klett, 1978). When supercooling is stronger, in general, more ice crystals

547 are nucleated for a given group of aerosols acting as INP. The average air temperature

548 immediately below the cloud base over the simulation period is -16 ºC in the 200_2 run

and -5 ºC in the control-midlatitude run. The average air temperature immediately above
the cloud top is -33 ºC in the 200_2 run and -15 ºC in the control-midlatitude run. Hence,
supercooling is greater and this contributes to the higher ICNCavg/CDNCavg in the polar
case than in the midlatitude case. The higher ICNCavg/CDNCavg is likely to induce more
portion of water vapor to be deposited onto ice crystals in the polar case than in the
midlatitude case. It is hypothesized that this in turn enables IWC/LWC in the 200_2 run to
be one order of magnitude greater than that in the control-midlatitude run or in the
midlatitude case. Much higher IWC than LWC, which results in a much higher IWC/LWC
in the polar case than in the midlatitude case, in the 200_2 run overcomes lower LWC in
the 200_2 run than that in the 200_0 run, which leads to the greater total cloud mass in the
200_2 run than in the 200_0 run (Figure 4 and Table 2). However, IWC whose magnitude
is similar to the magnitude of LWC, which results in a much lower IWC/LWC in the
midlatitude case than in the polar case, in the midlatitude case is not able to overcome
lower LWC in the midlatitude case than that in the midlatitude warm clouds, which leads
to the greater total cloud mass in the midlatitude warm clouds than in the midlatitude case;
here, the midlatitude warm clouds are generated by removing ice processes in the
midlatitude case. This means that associated with higher ICNC/CDNC and IWC/LWC, ice
processes enhance the total cloud mass for the polar case as compared to that for the polar
warm-cloud counterpart. However, in the midlatitude case, associated with lower
ICNC/CDNC and IWC/LWC, ice processes reduce the total cloud mass as compared to
that for the midlatitude warm-cloud counterpart.

**3.1.4  Role of ICNC/CDNC**

To test the hypothesis above about the role of ICNC/CDNC in above-described differences
between the polar and midlatitude cases, the 200_2 run is repeated by reducing
ICNCavg/CDNCavg by a factor of 10. This is done by reducing the concentration of
aerosols acting as INP but not CCN in a way that ICNCavg/CDNCavg is lower by a factor
of 10 in the repeated run than in the 200_2 run. In this way, this repeated run has
ICNCavg/CDNCavg at the same order of magnitude as that in the control-midlatitude run.
This repeated run is referred to as the 200_0.07 run. As shown in Figure 8 and Table 2, the
200_0.07 run shows much lower deposition rate and IWC than the 200_2 run does.
However, as we move from the 200_2 run to the 200_0.07 run, the time- and domain-
averaged condensation rate and LWC increases (Figure 8 and Table 2). This is because
reduction in deposition increases the amount of water vapor, which is not consumed by
deposition but available for condensation. Associated with this, in the 200_0.07 run, the
time- and domain-averaged deposition rate and IWC become similar to the average
condensation rate and LWC, respectively (Figure 8 and Table 2). Hence, IWC/LWC
reduces from 26.28 in the 200_2 run to 1.05 in the 200_0.07 run as ICNCavg/CDNCavg
reduces from the 200_2 run to the 200_0.07 run. Here, IWC/LWC in the 200_0.07 run is
similar to that in the midlatitude-control run, which demonstrate that the difference in
ICNC/CDNC is able to explain the difference in IWC/LWC between the polar and
midlatitude cases. It is notable that the reduction in deposition is dominant over the increase
in condensation with the decrease in ICNCavg/CDNCavg. Hence, the sum of condensation
and deposition rates and WC reduce from the 200_2 run to the 200_0.07 run. That the sum
of condensation and deposition rates and WC reduce in a way that the sum and WC in the
mixed-phase clouds in the 200_0.07 run are lower than condensation rate and LWC,
respectively, in the warm clouds in the 200_0 run is also notable (Figure 8 and Table 2).
This is similar to the situation in the midlatitude case and thus demonstrates that the
different relation between the mixed-phase and warm clouds can be associated with the
difference in ICNC/CDNC between the polar and midlatitude cases.

The rate of the sedimentation of ice crystals at the cloud base reduces as

ICNCavg/CDNCavg reduces between the 200_2 and 200_0.07 runs, mainly due to
reduction in the ice-crystal mass (Table 2). The rate of droplet sedimentation at the cloud
base increases as ICNCavg/CDNCavg reduces mainly due to increases in droplet mass and
size in association with the increases in LWC (Table 2). The entrainment rate at the cloud
top reduces as ICNCavg/CDNCavg reduces (Table 2). It is found that those changes in the
average rates of the droplet and ice-crystal sedimentation over the cloud base and
simulation period are ~four to five orders of magnitude smaller than those in the average
integrated condensation and deposition rates between the 200_2 and 200_0.07 runs (Table
2). The entrainment tends to reduce the total cloud mass or WC less with the reducing
ICNCavg/CDNCavg. Hence, changes in the entrainment counters the decrease in WC with
the reducing ICNCavg/CDNCavg between the 200_2 and 200_0.07 runs. Here, we see that
changes in the entrainment are not factors that lead to the increase in LWC, and the
decrease in IWC, and eventually the decrease in WC with the reducing
ICNCavg/CDNCavg. The analysis of the sedimentation and entrainment exclude them
from factors inducing above-described differences between the 200_2 and 200_0.07 runs.
Instead, this analysis grants confidence in the fact that deposition and condensation, which
are strongly dependent on ICNC/CDNC, are main factors inducing those differences.

**3.2 Aerosol-cloud interactions**

Comparisons between the 200_2 and 2000_20 runs show that with the increasing
concentration of both of aerosols acting as CCN and those as INP, IWC increases but LWC
decreases in the polar case (Figures 9 and Table 2). These decreases in LWC are negligible
as compared to these increases in IWC. Hence, the increases in IWC outweigh the
decreases in LWC, leading to aerosol-induced increases in WC (Figures 9 and Table 2).
To identify roles of specific types of aerosols in these aerosol-induced changes,
comparisons not only between the 200_2 and 200_20 runs but also between the 200_2 and
2000_2 runs are performed. Comparisons between the 200_2 and 200_20 runs show that
the increasing concentration of aerosols acting as INP induces increases in IWC but
decreases in LWC (Figure 9 and Table 2). The magnitudes of these increases and decreases
are similar to those between the 200_2 and 2000_20 runs (Figure 9 and Table 2). However,
comparisons between the 200_2 and 2000_2 runs show that the increasing concentration
of aerosols acting as CCN induces negligible changes in either IWC or LWC. Thus, CCN-
induced changes in the total cloud mass are negligible, although the increasing
concentration of aerosols acting as CCN induces a slight decrease in IWC, and a slight
increase in LWC (Figure 9 and Table 2). This demonstrates that INP plays a much more
important role than CCN when it comes to the response of the total cloud mass to increasing
aerosol concentrations. However, in the midlatitude case, the increasing concentration of
aerosols acting as CCN generates changes in the mass as significantly as the increasing
concentration of aerosols acting as INP does.
To identify roles played by ice processes in aerosol-cloud interactions, a pair of the
200_0 and 2000_0 runs are analyzed and compared to the previous four standard
simulations (i.e., the 200_2, 200_20, 2000_2 and 2000_20 runs). The CCN-induced
increases in LWC in those noice runs are much greater than the CCN-induced changes in
WC in the 200_2 and 2000_2 runs (Figure 9 and Table 2). However, these CCN-induced
increases in LWC in the noice runs are smaller than the INP-induced increases in WC in
the 200_2 and 200_20 runs (Figure 9 and Table 2). This is different from the midlatitude
case where changes in the total cloud mass, whether they are induced by the increasing
concentration of aerosols acting as CCN or INP, in the mixed-phase clouds are much lower
than those CCN-induced changes in the warm clouds.

**3.2.1   Deposition, condensation, sedimentation and entrainment**

The CCN-induced increases in condensation rates and decreases in deposition rates are
negligible. This leads to the CCN-induced negligible increases in LWC and negligible
decreases in IWC between the 200_2 and 2000_2 runs (Figure 9 and Table 2). However,
between the 200_2 and 200_20 runs, rather the significant INP-induced increases are in
deposition rate, leading to the significant INP-induced increases in IWC (Figure 9 and
Table 2). Between the 200_2 and 200_20 runs, INP-induced decreases in condensation
rate are negligible, leading to the negligible INP-induced decreases in LWC, as compared
to the INP-induced increases in deposition rate and IWC (Figure 9 and Table 2). With the
increasing concentration of aerosols acting as INP from the 200_2 run to the 200_20 run,
the sedimentation of ice crystals at the cloud base decreases (Table 2). This is mainly due
to decreases in the size of ice crystals in association with increases INP and resultant
increases in ICNC. In Figure 10a, we see that the number concentration of ice crystals with
diameters smaller and larger than ~40 micron increases and decreases, respectively, as we
move from the 200_2 run to the 200_20 run, which indicate a shift of the sizes of ice
crystals to smaller ones. From the 200_2 run to the 200_20 run, the sedimentation of
droplets at the cloud base decreases as shown in Table 2, mainly due to decreases in LWC.
Figure 10b shows that the number concentration of drops decreases throughout almost all
parts of the size range from the 200_2 run to the 200_20 run, which indicates a negligible
shift in the drop size but a reduction in LWC. It is found that changes in the average rates
of the droplet and ice-crystal sedimentation over the cloud base and simulation period are
~three to four orders of magnitude smaller than those in the average integrated
condensation and deposition rates between the 200_2 and 200_20 runs (Table 2). From the
200_2 run to the 200_20 run, the entrainment at the cloud top increases (Table 2). Hence,
the entrainment reduces WC less in the 200_2 run than in the 200_20 run. Here, we see
that changes in entrainment and the sedimentation are not factors that we have to focus on
to explain the changes in LWC, IWC and WC between the 200_2 and 200_20 runs.
In the warm clouds in the 200_0 and 2000_0 runs, the CCN-induced increases in
condensation rate occur, leading to those in LWC (Figure 9 and Table 2). However, the
CCN-induced increases in condensation rate in the warm clouds associated with the polar
case are lower than the INP-induced increases in deposition rate in the polar case (Table
2). This contributes to aerosol-induced smaller changes in the total cloud mass in the polar
warm clouds than in the polar mixed-phase clouds. The sedimentation of droplets at the
cloud base reduces and the entrainment at the cloud top increases from the 200_0 run to
2000_0 run (Table 2). The increasing concentration of aerosols acting as CCN induces
increases in CDNC and decreases in the droplet size, leading to the reduction in the droplet
sedimentation from the 200_0 run to 2000_0 run. The entrainment counters the CCN-
induced increases in LWC from the 200_0 run to 2000_0 run. Hence, the entrainment is
not a factor which induces the CCN-induced increases in LWC between the 200_0 and
2000_0 runs. As seen in Table 2, the changes in the sedimentation rate is ~three orders of
magnitude smaller than those in the integrated condensation rate between the 200_0 and
2000_0 runs. Hence, it is not the sedimentation but condensation that we have to look at to
explain changes in LWC or WC between the 200_0 and 2000_0 runs.

**3.2.2   Understanding differences between the polar and midlatitude cases**

Roughly speaking, the CCN-induced changes in LWC via CCN-induced changes in
autoconversion of droplets are proportional to LWC that changing CCN affect, and INP-
induced changes in IWC via INP-induced changes in autoconversion of ice crystals are
proportional to IWC that changing INPs affect (e.g., Dudhia, 1989; Murakami, 1990; Liu
and Daum, 2004; Morrison et al., 2005, 2009 and 2012; Lim and Hong, 2010; Mansell et
al. 2010; Kogan, 2013; Lee and Baik, 2017). This is for given environmental conditions
(e.g., temperature and humidity) and given CCN- or INP-induced changes in microphysical
factors such as sizes and number concentrations of droplets or ice crystals. Hence, in the
polar case, with a given much lower LWC than IWC, the changing concentration of
aerosols acting as CCN is likely to induce smaller changes in the given LWC via CCN
impacts on the droplet autoconversion. This is as compared to changes in the given IWC
which are induced by the changing concentration of aerosols acting as INP and thus
changing ice-crystal autoconversion.

The smaller changes in the given LWC are related to changes in CDNC. These changes

in CDNC are initiated by those in droplet autoconversion. The larger changes in the given
IWC are related to changes in ICNC. These changes in ICNC are initiated by those in ice-
crystal autoconversion. Changes in integrated droplet surface area, which are induced by
those in CDNC, initiate those in the given LWC. Changes in integrated ice-crystal surface
area, which are induced by those in ICNC, initiate those in the given IWC. Remember that
condensation occurs on droplet surface and thus droplets act as a source of condensation,
and deposition occurs on ice-crystal surface and thus ice crystals act as a source of
deposition. Hence, those changes in CDNC and associated integrated droplet surface area
can lead to changes in condensation and thus feedbacks between condensation and updrafts,
while those changes in ICNC and associated integrated ice-crystal surface area can lead to
changes in deposition and thus feedbacks between deposition and updrafts. The smaller
CCN-induced changes in LWC involve changes in CDNC and associated smaller changes
in condensation and feedbacks between condensation and updrafts in the polar case. This
is as compared to changes in deposition and feedbacks between deposition and updrafts
which are associated with the INP-induced changes in ICNC and the related larger INP-
induced changes in IWC in the polar case. The smaller CCN-induced changes in LWC
involve smaller changes in water vapor that is consumed by droplets in the polar case. The
larger INP-induced changes in IWC involve larger changes in water vapor that is consumed
by ice crystals in the polar case. This leaves the CCN-induced smaller changes in the
amount of water vapor available for deposition, which induce the smaller CCN-induced
changes in IWC in the polar case. This is as compared to the INP-induced changes in the
amount of water vapor which is available for condensation and associated changes in LWC
in the polar case.
The lower LWC in the polar warm clouds than IWC in the polar case contributes to the
INP-induced greater changes in IWC than the CCN-induced changes in LWC in the polar
warm clouds. The lower LWC in the polar case than that in the polar warm clouds
contributes to the CCN-induced greater changes in LWC in the polar warm clouds than
those in LWC and subsequent changes in IWC in the polar case.
In contrast to the situation in the polar case, in the midlatitude case, remember that a
given LWC is at the same order of magnitude of IWC. Hence, the CCN- induced changes
in LWC and subsequent changes in IWC are similar to the INP-induced changes in IWC
and subsequent changes in LWC. The greater LWC in the midlatitude warm cloud than
both of LWC and IWC in the midlatitude case contributes to the greater CCN-induced
changes in LWC in the midlatitude warm cloud. This is as compared to either the CCN-
induced changes in LWC and subsequent changes in IWC or the INP-induced changes in
IWC and subsequent changes in LWC in the midlatitude case.
To confirm above-described mechanisms in this section, which explain different
aerosol-cloud interactions between the polar and midlatitude cases, the 200_0.07 run is
repeated by increasing INP by a factor of 10 in the PBL at the first time step. This repeated
run is referred to as "the 200_0.7 run. Then, the 200_0.07 run is repeated again by
increasing CCN by a factor of 10 in the PBL at the first time step. This repeated run is
referred to as the 2000_0.07 run. These repeated runs are to see the response of IWC and
LWC to the increasing concentration of aerosols acting as INP and CCN. This is when
IWC and LWC are at the same order of magnitude and lower in mixed-phase clouds than
LWC in the warm-cloud counterpart as in the 200_0.07 run and midlatitude case.
Comparisons between the 200_0.07, 200_0.7 and 2000_0.07 runs show that the INP-
induced changes in IWC and LWC are similar to the CCN-induced changes in IWC and
LWC, respectively, as in the midlatitude case (Figure 9 and Table 2). These comparisons
also show that the CCN-induced changes in LWC in the polar warm cloud are greater
(Figure 9 and Table 2). This is as compared to either the CCN-induced changes in LWC
and subsequent changes in IWC between the 200_0.07 and 2000_0.07 runs or the INP-
induced changes in IWC and subsequent changes in LWC between the 200_0.07 and
200_0.7 runs (Figure 9 and Table 2). These comparisons demonstrate that differences in
ICNC/CDNC play a critical role in differences in aerosol-cloud interactions between the
polar and midlatitude cases, considering that differences in ICNC/CDNC between the
200_2 and 200_0.07 runs are at the same order of magnitude of those between the cases.

**3.3  Radiation**

Studies (e.g., Ovchinnikov et al., 2011; Possner et al., 2017; Solomon et al., 2018) have
focused on radiative cooling and subsequent changes in stability and dynamics as a primary
driver for the development of mixed-phase stratocumulus clouds and aerosol-induced
changes in LWC and IWC in those clouds. Motivated by these studies, to isolate the role
of radiative processes in cloud development and aerosol impacts on LWC and IWC, all of
the simulations above are repeated by turning off radiative processes. In these repeated
runs, radiative fluxes over the whole domain and simulation period are zero. The basic
summary of results from these repeated runs is given in Table 3. As seen in comparisons
between Tables 2 and 3, the qualitative nature of results, which are mainly about
differences in IWC/LWC, the relative importance of the impacts of INP on IWC and LWC
as compared to those impacts of CCN, and how warm and mixed-phase clouds are related
between the polar and midlatitude cases, in this study does not vary with whether radiative
processes exist or not. This demonstrates that ICNC, CDNC, deposition and condensation
but not radiative processes drive results in this study.

**4.  Discussion**

**4.1 Examination of the role of ICNC/CDNC in IWC/LWC in 200_2,**
**2000_20, 2000_2, 200_20, 200_0.07, 2000_0.07 and 200_0.7 runs**

So far, comparisons between the set of the 200_2, 2000_20, 2000_2 and 200_20 runs for
the polar case and the other set of the 200_0.07, 2000_0.07 and 200_0.7 runs, which
represents the midlatitude case, have been mainly utilized to understand the role of
ICNC/CDNC. However, even when it comes to all the runs in both the sets, differences in
ICNCavg/CDNCavg and IWC/LWC are shown among them (Tables 1 and 2). For more
robust examination of particularly the role of ICNC/CDNC in IWC/LWC, which is
basically about the increase and decrease in ICNC/CDNC inducing the increase and
decrease in IWC/LWC, respectively, as identified from the comparison between the 200_2
and 200_0.07 runs in Section 3.1.4, all the runs in the sets are utilized by ordering them as
shown in Table 4. This ordering is done in a way that as we move from the first run in the
first row to the last run in the last row of Table 4, ICNCavg/CDNCavg increases. Overall,
with increasing ICNCavg/CDNCavg, IWC/LWC increases in Table 4 as also seen in Figure
11 that shows IWC/LWC as a function of ICNCavg/CDNCavg based on Table 4. This is
despite the fact that the increase in IWC/LWC is highly non-linear in terms of the increase
in ICNCavg/CDNCavg as seen in the percentage increases, and a decrease in IWC/LWC
is seen with an increase in ICNCavg/CDNCavg from the 2000_20 run to the 200_2 run
(Table 4 and Figure 11); this high-degree non-linearity in the increase in IWC/LWC is
associated with the fact that interactions between cloud microphysical, thermodynamic and
dynamic processes are well known to be highly non-linear. Hence, overall, findings
regarding the role of ICNC/CDNC in IWC/LWC from the comparison between the 200_2
and 200_0.07 runs are applicable to all the runs in the sets except for the role between the
2000_20 and 200_2 runs. Here, it is notable that the percentage difference in
ICNCavg/CDNCavg is ~9% between the 2000_20 and 200_2 runs and the smallest among
those differences in Table 4. The other differences are larger than 80%. Hence, the
percentage difference in ICNCavg/CDNCavg for a pair of the 2000_20 and 200_2 runs is
at least ~one order of magnitude smaller than that for the other pairs of the runs in Table 4.
This means that findings from the comparison between the 200_2 and 200_0.07 runs are
not suitable to explain the variation of IWC/LWC among clouds when the variation of
ICNC/CDNC is relatively insignificant. According to Table 4, it seems that the variation
of ICNC/CDNC should be greater than a critical value above which those findings are
useful to account for the IWC/LWC variation among clouds.

The high-degree non-linearity in the variation of IWC/LWC is epitomized by the 1706

percent increase in IWC/LWC for the 163 percent increase in ICNCavg/CDNCavg from
the 200_0.7 run to the 2000_2 run. This 1706 percent increase in IWC/LWC is induced by
increases in both the initial number concentrations of CCN and INP between the runs
(Table 1). In other transition from a simulation in a row to that in the next row in Table 4,
there are decreases in both the initial number concentrations of CCN and INP, or there is
either a change in the initial number condensation of CCN or INP. When either the initial
concentration of CCN or INP changes in the transition, less than a 100% increase in
IWC/LWC is shown. The decreases in both the initial number concentrations of CCN and
INP, which are from the 2000_20 run to the 200_2 run, result in the decrease in IWC/LWC.
Hence, depending on how the initial number concentrations of CCN and INP change, the
magnitude and sign of the change in IWC/LWC can vary substantially.

**4.2 Role of a given ICNC/CDNC in IWC/LWC for different concentrations of**

**aerosols acting as INP and CCN**


Simulations which are compared in Section 4.1 and shown in Table 4 have not only
different ICNCavg/CDNCavg but also the different number concentrations of aerosols
acting as CCN and INP at the first time step (Table 1). To better isolate particularly the
role of ICNC/CDNC in IWC/LWC, we need to show that results in Section 4.1 are valid
regardless of the variation of the number concentration of aerosols. For this need, we focus
on the 200_2 and 200_0.07 runs, since the primary understanding of the role of
ICNC/CDNC in IWC/LWC comes from the comparison between these runs as described
in Section 3.1.4. To fulfill the need, each of these runs are repeated by varying the number
concentration of aerosols acting as CCN and INP in a way that ICNCavg/CDNCavg does
not vary (Tables 1 and 5). The 4000_45 and 13_0.1 runs are the repeated 200_2 run, and
the 4000_1.8 and 12_0.0035 runs are the repeated 200_0.07 run (Tables 1 and 5). The set
of the 200_2, 4000_45 and 13_0.1 runs is referred to as the polar set, and that of the
200_0.07, 4000_1.8 and 12_0.0035 runs is referred to as the midlatitude set in this section.
Among the three runs in each of the sets, less than 4% variation of IWC/LWC is shown
(Table 5). This less-than-4% variation is so small that the start contrast in IWC/LWC
between the 200_2 and 200_0.07 runs as discussed in Section 3.1.4 is also shown between
the polar and midlatitude sets (Table 5). Hence, the role of the difference in a given
ICNC/CDNC in the difference in IWC/LWC between the 200_2 and 200_0.07 runs as
described in Section 3.1.4 is considered robust to the varying concentration of aerosols.

**4.3 Role of environmental factors, sedimentation, aerosol sources and advection**


This study picks ICNC/CDNC as an important factor which differentiates IWC/LWC and interactions among clouds, aerosols and ice processes in the polar case from those in the midlatitude case. However, this does not mean that no other potential factors, which can explain the variation of IWC/LWC and interactions among clouds, aerosols and ice processes between different clouds, exist. For example, differences in environmental factors (e.g., stability and wind shear) between those different clouds can have an impact on the variation. Particularly, differences in stability and wind shear can initiate those in the dynamic development of turbulence. Then, this subsequently induces differences in the microphysical and thermodynamic development of clouds, IWC/LWC and interactions among clouds, aerosols and ice processes. Hence, factors such as stability and wind shear can have different orders of procedures, which involve dynamics, thermodynamics and microphysics, than ICNC/CDNC in terms of differentiation between different clouds. Thus, different mechanisms controlling the differentiation can be expected regarding factors such as stability and wind shear as compared to ICNC/CDNC. The examination of these different mechanisms among stability, wind shear and ICNC/CDNC deserves future study for more comprehensive understanding of the differentiation or for an above-mentioned more fully established general principle explaining the differentiation.

Another point to make is that the cases in this study have weak precipitation and the associated weak sedimentation of ice crystals and droplets. In mixed-phase clouds with strong precipitation and the sedimentation, they can play roles as important as in-cloud latent-heat processes in IWC/LWC and interactions among clouds, aerosols and ice processes. In those clouds with strong precipitation, the sedimentation can take part in the interplay between ICNC/CNDC and latent-heat processes by affecting cloud mass and associated ICNC and CDNC significantly, and play a role in the differentiation of IWC/LWC and interactions among clouds, aerosols and ice processes when it comes to different cases of mixed-phase clouds. For more generalization of results here as a way to the more fully established general principle, this potential role of sedimentation needs to

be investigated by performing more case studies involving cases with strong precipitation
in the future.
It should be emphasized that although this study mentions air temperature as a factor
that affects ICNC/CDNC, ICNC/CDNC can be affected by other factors such as sources of
aerosols acting as INP and those acting as CCN, and/or the advection of those aerosols.
Hence, even for cloud systems that develop with a similar air-temperature condition, for
example, when those systems are affected by different sources of aerosols and/or their
different advection, they are likely to have different ICNC/CDNC, IWC/LWC, relative
importance of impacts of INP on IWC and LWC as compared to those impacts of CCN,
and relation between warm and mixed-phase clouds. Regarding factors, which affect
ICNC/CDNC, such as sources and advection of aerosols together with temperature , it
should be noted that while this study utilizes differences in temperature among those
factors to identify cases exhibiting significant disparities in ICNC/CDNC, its primary
objective does not lie in the role of temperature differences in disparities in ICNC/CDNC,
but in comprehending the inherent role of ICNC/CDNC variations themselves in the
discrepancies observed, for example, in IWC/LWC, across diverse cloud systems.

**4.4 Mixing of droplets and ice crystals**

The representation of mixed-phase clouds in our study relies on the assumption of
homogeneously mixed ice and liquid hydrometeors within the model grid cells, a common
approach in many models. However, recent observational studies (e.g., D'Alessandro et al.,
2021; Korolev and Milbrandt, 2022; Schima et al., 2022; Coopman and Tan, 2023) have
shown that in reality, mixed-phase clouds often exhibit inhomogeneous distributions of ice
and liquid, with distinct pockets or regions of each phase. These observations suggest that
the microphysical processes, such as the Wegener-Bergeron-Findeisen process, may be
influenced by this inhomogeneity, potentially leading to differences in cloud dynamics and
feedbacks compared to what is simulated by models assuming the homogeneous mixing.
While our study, along with the work of Lee et al. (2021), uses a model-based
approach that assumes the homogeneous mixing, it is important to acknowledge that this
representation may not fully capture the complexity observed in real clouds. The
implications of this assumption could affect the accuracy of our simulations, particularly
in scenarios where phase-transition processes in mixed-phase clouds play a significant role.
As such, the results presented should be interpreted with this limitation in mind, and further
work incorporating more detailed representations of inhomogeneous hydrometeor
distributions may be needed to refine our understanding of mixed-phase cloud processes.

**5.    Summary and conclusions**

In this study, a case of mixed-phase stratiform clouds in a polar area, which is referred to
as "the polar case" is compared to that in a midlatitude area, which is referred to as "the
midlatitude case". This is to gain an understanding of how different ICNC/CDNC plays a
role in making differences in cloud properties, aerosol-cloud interactions and impacts of
ice processes on them between two representative areas (i.e., polar and midlatitude areas)
where mixed-phase stratiform clouds form and develop. Among those cloud properties,
this study focuses on IWC/LWC that plays an important role in cloud radiative properties.
To gain the understanding efficiently, the polar case is chosen in a way to make stark
contrast with the midlatitude case in terms of ICNC/CDNC and IWC/LWC. Although such
polar cases may be uncommon, the stark contrast provides an opportunity to elucidate
mechanisms that control the above-mentioned role of different ICNC/CDNC.
Due to lower air temperature, more ice crystals are nucleated, leading to higher
ICNC/CDNC in the polar case than in the midlatitude case. This higher ICNC/CDNC
enables the more efficient deposition of water vapor onto ice crystals in the polar case. This
leads to much higher IWC/LWC in the polar case. The more efficient deposition of water
vapor onto ice crystals enables the polar mixed-phase clouds to have the greater total cloud
mass than the polar warm clouds. However, the less efficient deposition of water vapor
onto ice crystals causes the midlatitude mixed-phase clouds to have less total cloud mass
than the midlatitude warm clouds. With the increasing ICNC/CDNC from the midlatitude
case to the polar case, impacts of CCN and INP on the total cloud mass become less and
more important, respectively.
Previous studies on mixed-phase stratocumulus clouds (e.g., Ovchinnikov et al., 2011;
Possner et al., 2017; Solomon et al., 2018) have primarily focused on investigating the
impacts of cloud-top radiative cooling, entrainment, and sedimentation of ice particles on
these clouds, as well as their interactions with aerosols. However, there are a scarcity of
studies that specifically examine the role of microphysical interactions, involving
processes such as condensation and deposition, as well as factors like cloud-particle
concentrations, between ice and liquid particles in mixed-phase stratocumulus clouds, and
their interactions with aerosols as performed in this study. Therefore, our study contributes
to a more comprehensive understanding of mixed-phase clouds and their intricate interplay
with aerosols.
This study suggests that a microphysical factor, which is ICNC/CDNC, can be a
simplified and useful tool to understand differences among different systems of
stratocumulus clouds in various regions in terms of IWC/LWC and the relative importance
of INP and CCN in aerosol-cloud interactions, and thus to contribute to the development
of general parameterizations of those clouds in various regions for climate models. This
factor can also be a useful tool for a simplified understanding of different roles of ice
processes when mixed-phase clouds are compared to their warm-cloud counterparts in
terms of the cloud development and its interactions with aerosols among those different
systems. It should be noted that warm clouds have been studied much more than mixed-
phase clouds, although mixed-phase clouds play as important roles as warm clouds in the
evolution of climate and its change. This study provides preliminary mechanisms which
differentiate mixed-phase clouds and their interactions with aerosols from their warm-
cloud counterparts, and control the variation of the differentiation in different regions as a
way of improving our understanding of mixed-phase clouds. It should be mentioned that
the efficient way of developing general parameterizations, which are for climate models
and consider all of warm, mixed-phase clouds in various regions and their interactions with
aerosols, can be achieved by just adding those mechanisms to pre-existing
parameterizations of much-studied warm clouds instead of developing brand new
parameterizations from the scratch.
This study finds that the relation between ICNC/CDNC and IWC/LWC is highly non-
linear. This high non-linearity is closely linked to how the number concentrations of CCN
and INP, and associated ICNC/CDNC change. For a specific situation where the
ICNC/CDNC variation is relatively small and both the number concentrations of CCN and
INP reduce, the increase in ICNC/CDNC can reduce IWC/LWC, although it is found that
as a whole, the increase in ICNC/CDNC enhances IWC/LWC. Hence, mechanisms
identified in this study, especially regarding the use of ICNC/CDNC as a simplified and
useful tool to explain differences in IWC/LWC among different cloud systems, are not
complete and entirely general. In addition, results in this study are from only two cases in
two specific locations in the midlatitude and Arctic regions and the more generalization of
these results in this study merits more case studies over more locations in those regions,
for example, in terms of above-mentioned sedimentation intensity, different factors (e.g.,
environmental factors) other than ICNC/CDNC, different sources and advection of
aerosols, the magnitude of the variation of ICNC/CDNC and the way number
concentrations of CCN and INP vary. Hence, findings particularly about relations between
ICNC/CDNC and IWC/LWC in this study should be considered preliminary ones that
initiate future work to streamline the development of the general parameterizations.


















**Code/Data source and availability**

Our private computer system stores private data such as the model code and output, and the CCN data. Upon approval from funding sources, the data will be opened to the public. Projects related to this paper have not been finished, thus, the sources prevent the data from being open to the public currently. However, if information on the data is needed, contact the corresponding author Seoung Soo Lee (slee1247@umd.edu).

The Cloudnet and reanalysis data used in this study are publicly available. The Cloudnet data are obtainable at "https://cloudnet.fmi.fi/search/data", while the reanalysis data can be obtained by contacting Met Office via "https://www.metoffice.gov.uk/about-us/contact"

**Author contributions**

Essential initiative ideas are provided by SSL, CHJ and YJY to start this work. Simulation and observation data are analyzed by SSL, CHJ and JU. YZ, JP, MGM and SKS review the results and contribute to their improvement. JC provides supports to set up and run additional simulations during the review.

**Competing interests**

The authors declare that they have no conflict of interest.

**Acknowledgements**

This study is supported by the National Research Foundation of Korea (NRF) grant funded by the Korea government (MSIT) (Nos. NRF2020R1A2C1003215, NRF2020R1A2C2011081, NRF2023R1A2C1002367, NRF2021M1A5A1065672/KOPRI-PN24011 and 2020R1A2C1013278), and Basic Science Research Program through the NRF funded by the Ministry of Education (No. 2020R1A6A1A03044834).

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

**FIGURE CAPTIONS**

Figure 1. A red rectangle marks the simulation domain in the Svalbard area, Norway, and
a dot in the rectangle marks a ground station which is a part of the Cloudnet observation
network. The light blue represents the ocean and the green the land area.

Figure 2. (a) The vertical distributions of the domain-averaged potential temperature and
humidity at the first time step, (b) the time series of the domain-averaged large-scale
subsidence or downdraft at the model top and (c) the time series of the domain-averaged
surface temperature.

Figure 3. Aerosol size distribution at the surface. N represents aerosol number
concentration per unit volume of air and D represents aerosol diameter.

Figure 4. The vertical distributions of the time- and domain-averaged IWC and LWC in
the 200_2 and 200_0 runs.

Figure 5. The time series of (a) observed and simulated cloud-top and bottom heights, (b)
retrieved and simulated IWP, and observed and simulated LWP, and (c) the simulated
surface sensible and latent heat fluxes. Observed and retrieved values are from the ground
station as marked in Figure 1. For the time series, in the simulation domain, the simulated
cloud-top height is averaged over grid points with cloud tops and the simulated cloud-
bottom height is averaged over grid points with cloud bottoms, while the simulated IWP
and LWP are averaged over grid points with non-zero IWP and LWP, respectively, at each
time step in the 200_2 run. The simulated surface sensible and latent heat fluxes are
averaged over the horizontal domain at the surface and each time step in the 200_2 run.

Figure 6. The vertical distributions of the time- and domain-averaged deposition and
condensation rates in the 200_2 and 200_0 runs.

Figure 7. The time series of the average supersaturation with respect to ice and water over
grid points where deposition occurs in the presence of both droplets and ice crystals in the
200_2 run.
Figure 8. The vertical distributions of the time- and domain-averaged IWC and LWC in
the 200_2, 200_0 and 200_0.07 runs.

Figure 9. The vertical distributions of the time- and domain-averaged (a) IWC in the 200_2,
2000_20, 200_0.07, 200_20, 2000_2, 2000_0.07, and 200_0.7 runs. (b) The vertical
distributions of the time- and domain-averaged LWC in the 200_0 and 2000_0 runs as well
as all the runs shown in panel (a).

Figure 10. The average size distributions of (a) ice crystals over grid points with non-zero
IWC and the simulation period and (b) drops over grid points with non-zero LWC and the
simulation period.

Figure 11. IWC/LWC as a function of ICNCavg/CDNCavg based on Table 4.















| Simulations | The number concentration of aerosols acting as CCN at the first time step in the PBL $(cm^{-3})$ | The number concentration of aerosols acting as INP at the first time step in the PBL $(cm^{-3})$ | ICNCavg/CDNCavg | Ice processes | Radiation |
|---|---|---|---|---|---|
| 200_2 | 200 | 2 | 0.220 | Present | Present |
| 2000_20 | 2000 | 20 | 0.201 | Present | Present |
| 2000_2 | 2000 | 2 | 0.108 | Present | Present |
| 200_20 | 200 | 20 | 0.512 | Present | Present |
| 200_0 | 200 | 2 | 0.000 | Absent | Present |
| 2000_0 | 2000 | 2 | 0.000 | Absent | Present |
| 200_0.07 | 200 | 0.07 | 0.022 | Present | Present |
| 2000_0.07 | 2000 | 0.07 | 0.012 | Present | Present |
| 200_0.7 | 200 | 0.7 | 0.041 | Present | Present |
| 4000_45 | 4000 | 45 | 0.220 | Present | Present |
| 13_0.1 | 13 | 0.1 | 0.220 | Present | Present |
| 4000_1.8 | 4000 | 1.8 | 0.022 | Present | Present |
| 12_0.0035 | 12 | 0.0035 | 0.022 | Present | Present |


Table 1. Summary of simulations
















| Simulations | IWC ($10^{-3}$ g m$^{-3}$) | LWC ($10^{-3}$ g m$^{-3}$) | IWP (g m$^{-2}$) | LWP (g m$^{-2}$) | IWC/LWC | IWP/LWP | Condensation rate | | Deposition rate | | Cloud-base sedimentation ($10^{-3}$ g m$^{-2}$ s$^{-1}$) | | Entrainment (cm s$^{-1}$) |
|---|---|---|---|---|---|---|---|---|---|---|---|---|---|
| | | | | | | | Over grid points ($10^{-2}$ g m$^{-3}$ s$^{-1}$) | Over cloudy columns ( g m$^{-2}$ s$^{-1}$) | Over grid points ($10^{-2}$ g m$^{-3}$ s$^{-1}$) | Over cloudy columns ( g m$^{-2}$ s$^{-1}$) | Ice-crystal | Droplet | |
| 200_2 | 6.57 | 0.25 | 31.94 | 1.23 | 26.28 | 25.96 | 0.11 | 1.98 | 1.30 | 23.40 | 1.17 | 0.17 | 0.25 |
| 2000_20 | 7.82 | 0.21 | 40.91 | 1.08 | 37.24 | 37.91 | 0.09 | 1.62 | 1.57 | 28.26 | 0.94 | 0.06 | 0.53 |
| 2000_2 | 6.55 | 0.29 | 31.85 | 1.46 | 22.58 | 21.81 | 0.12 | 2.16 | 1.28 | 23.04 | 1.11 | 0.08 | 0.28 |
| 200_20 | 7.80 | 0.20 | 40.82 | 1.01 | 39.00 | 40.42 | 0.09 | 1.62 | 1.56 | 28.08 | 0.97 | 0.11 | 0.51 |
| 200_0 | 0.00 | 2.06 | 0.00 | 10.35 | 0.00 | 0.00 | 0.72 | 12.48 | 0.00 | 0.00 | 0.00 | 0.36 | 0.08 |
| 2000_0 | 0.00 | 2.25 | 0.00 | 11.29 | 0.00 | 0.00 | 0.76 | 12.80 | 0.00 | 0.00 | 0.00 | 0.14 | 0.10 |
| 200_0.07 | 0.89 | 0.85 | 4.27 | 4.20 | 1.05 | 1.02 | 0.32 | 5.76 | 0.35 | 6.30 | 0.19 | 0.28 | 0.06 |
| 2000_0.07 | 0.79 | 0.97 | 3.82 | 4.83 | 0.81 | 0.79 | 0.38 | 6.84 | 0.31 | 5.58 | 0.17 | 0.19 | 0.07 |
| 200_0.7 | 0.98 | 0.78 | 4.73 | 3.88 | 1.25 | 1.22 | 0.31 | 5.58 | 0.39 | 7.02 | 0.14 | 0.22 | 0.07 |

Table 2. The averaged IWC, LWC, IWP, LWP, condensation and deposition rates over all of grid points and the simulation period in each of simulations. IWC/LWC (IWP/LWP) is the averaged IWC (IWP) over the averaged LWC (LWP).  Also, as shown are the vertically integrated condensation and deposition rates over each cloudy column which are averaged over those columns and the simulation period. The average cloud-base sedimentation rate, which is for each of ice crystals and droplets, over the cloud base and simulation period, and the average cloud-top entrainment rate over the cloud top and simulation period are shown as well.

| Simulations | IWC ($10^{-3}$ g m$^{-3}$) | LWC ($10^{-3}$ g m$^{-3}$) | IWP (g m$^{-2}$) | LWP (g m$^{-2}$) | IWC/LWC | IWP/LWP | Condensation rate | | Deposition rate | | Cloud-base sedimentation ($10^{-3}$ g m$^{-2}$ s$^{-1}$) | | Entrainment (cm s$^{-1}$) |
|---|---|---|---|---|---|---|---|---|---|---|---|---|---|
| | | | | | | | Over grid points ($10^{-2}$ g m$^{-3}$ s$^{-1}$) | Over cloudy columns ( g m$^{-2}$ s$^{-1}$) | Over grid points ($10^{-2}$ g m$^{-3}$ s$^{-1}$) | Over cloudy columns ( g m$^{-2}$ s$^{-1}$) | Ice-crystal | Droplet | |
| 200_2_norad | 6.42 | 0.24 | 31.21 | 1.22 | 26.75 | 25.58 | 0.10 | 1.96 | 1.29 | 23.35 | 1.16 | 0.16 | 0.24 |
| 2000_20_norad | 7.63 | 0.21 | 40.05 | 1.07 | 36.33 | 37.42 | 0.09 | 1.59 | 1.55 | 29.91 | 0.92 | 0.06 | 0.51 |
| 2000_2_norad | 6.40 | 0.29 | 31.11 | 1.45 | 22.06 | 21.45 | 0.11 | 2.12 | 1.26 | 22.69 | 1.07 | 0.08 | 0.27 |
| 200_20_norad | 7.61 | 0.20 | 39.95 | 0.99 | 38.05 | 40.35 | 0.09 | 1.59 | 1.54 | 27.72 | 0.97 | 0.11 | 0.49 |
| 200_0_norad | 0.00 | 2.03 | 0.00 | 10.20 | 0.00 | 0.00 | 0.72 | 12.31 | 0.00 | 0.00 | 0.00 | 0.34 | 0.08 |
| 2000_0_norad | 0.00 | 2.21 | 0.00 | 11.12 | 0.00 | 0.00 | 0.75 | 12.63 | 0.00 | 0.00 | 0.00 | 0.13 | 0.10 |
| 200_0.07_norad | 0.87 | 0.84 | 4.21 | 4.17 | 1.04 | 1.01 | 0.31 | 5.74 | 0.35 | 6.21 | 0.18 | 0.27 | 0.05 |
| 2000_0.07_norad | 0.78 | 0.96 | 3.78 | 4.80 | 0.81 | 0.79 | 0.36 | 6.81 | 0.30 | 5.50 | 0.16 | 0.18 | 0.06 |
| 200_0.7_norad | 0.97 | 0.76 | 4.70 | 3.85 | 1.25 | 1.22 | 0.30 | 5.55 | 0.38 | 6.91 | 0.13 | 0.21 | 0.06 |


Table 3. Same as Table 2 but for the repeated simulations with radiative processes turned
off.




















| Simulations | ICNCavg/CDNCavg | Percentage increases (+) or decrease (-) in ICNCavg/CDNCavg | IWC/LWC | Percentage increases (+) or decrease (-) in IWC/LWC |
|---|---|---|---|---|
| 2000_0.07 | 0.012 | | 0.81 | |
| 200_0.07 | 0.022 | +83.33% | 1.05 | +29.6% |
| 200_0.7 | 0.041 | +86.36% | 1.25 | +19.0% |
| 2000_2 | 0.108 | +163.4% | 22.58 | +1706.4% |
| 2000_20 | 0.201 | +86.1% | 37.24 | +64.9% |
| 200_2 | 0.220 | +9.4% | 26.28 | -29.4% |
| 200_20 | 0.512 | +132.7% | 39.00 | +48.4% |


Table 4. ICNCavg/CDNCavg and IWC/LWC in the simulations that are related to Section
4.1. The Percentage increases or decreases in ICNCavg/CDNCavg and IWC/LWC as
shown in the $i^{th}$ row are $\frac{(ICNCavg/CDNCavg)_i - (ICNCavg/CDNCavg)_{i-1}}{(ICNCavg/CDNCavg)_{i-1}} \times 100\ (\%)$ and
$\frac{(IWC/LWC)_i - (IWC/LWC)_{i-1}}{(IWC/LWC)_{i-1}} \times 100\ (\%)$, respectively. Here, $(ICNCavg/CDNCavg)_i$ and
$(IWC/LWC)_i$ represent ICNCavg/CDNCavg and IWC/LWC in the $i^{th}$ row, respectively.

















| Simulations | ICNCavg/CDNCavg | IWC/LWC | Percentage increases (+) or decrease (-) in IWC/LWC |
|---|---|---|---|
| Polar case | | | |
| 200_2 | 0.220 | 26.28 | |
| 4000_45 | 0.220 | 27.25 | +3.7% |
| 13_0.1 | 0.220 | 25.62 | -2.5% |
| Representing midlatitude case | | | |
| 200_0.07 | 0.022 | 1.05 | |
| 4000_1.8 | 0.022 | 1.09 | +3.8% |
| 12_0.0035 | 0.022 | 1.02 | -2.9% |


Table 5. ICNCavg/CDNCavg and IWC/LWC in the simulations that are related to Section
4.2. The percentage increases or decreases in IWC/LWC in the 4000_45 run or in the
13_0.1 run are $\frac{(IWC/LWC)_{4000\_45\ or\ 13\_0.1} - (IWC/LWC)_{200\_2}}{(IWC/LWC)_{200\_2}} \times 100\ (\%)$. Here,
(IWC/LWC)$_{4000\_45\ or\ 13\_01}$ represents IWC/LWC in the 4000_45 run or the 13_01 run, while
(IWC/LWC)$_{200\_2}$ represents IWC/LWC in the 200_2 run. The percentage increases or
decreases in IWC/LWC in the 4000_1.8 run or the 12_0.0035 run are
$\frac{(IWC/LWC)_{4000\_1.8\_fac10\ or\ 12\_0.0035\_fac10} - (IWC/LWC)_{200\_2\_fac10}}{(IWC/LWC)_{200\_2\_fac10}} \times 100\ (\%)$. Here,
(IWC/LWC)$_{4000\_1.8\ or\ 12\_0.0035}$ represents IWC/LWC in the 4000_1.8 run or the 12_0.0035
run, while (IWC/LWC)$_{200\_0.07}$ represents IWC/LWC in the 200_0.07 run.













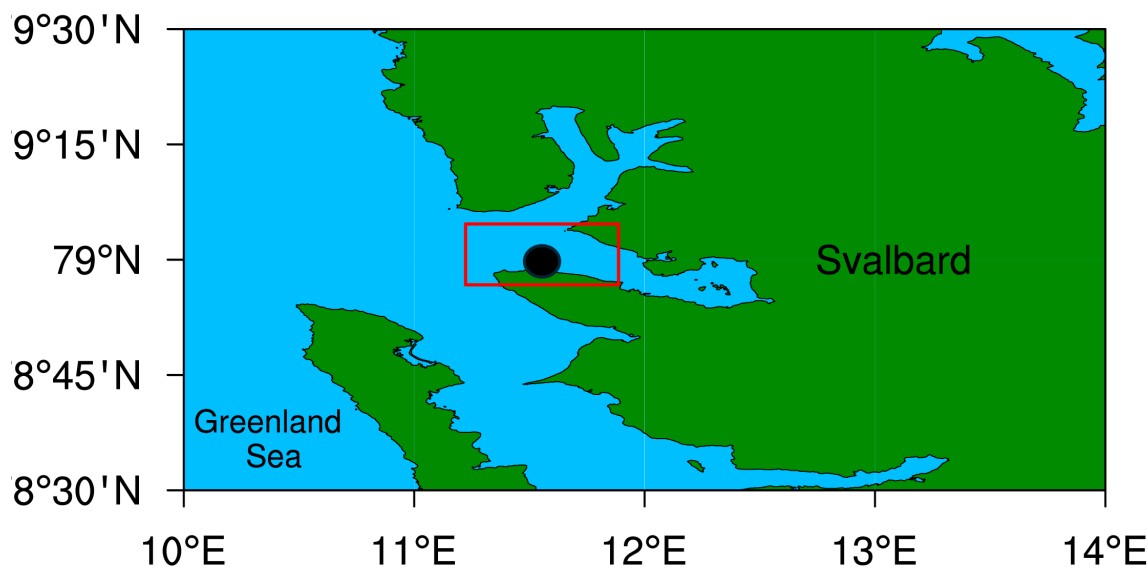

**Figure 1**

(a)

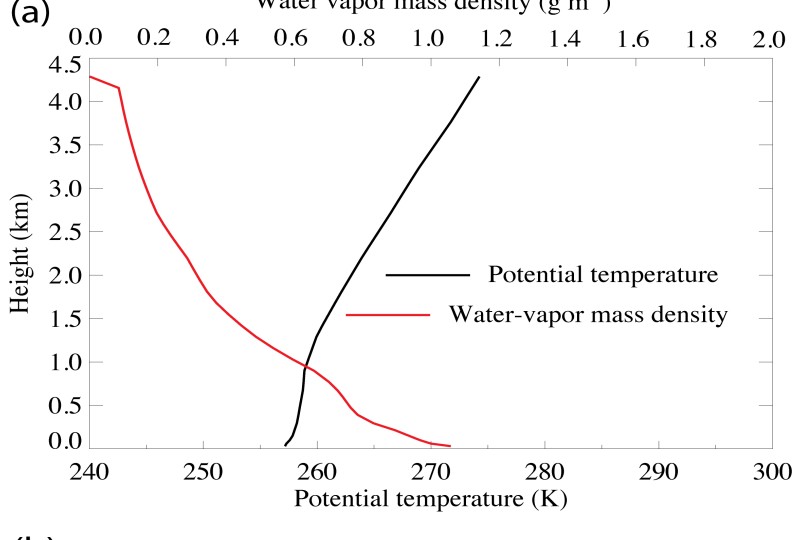

(b)

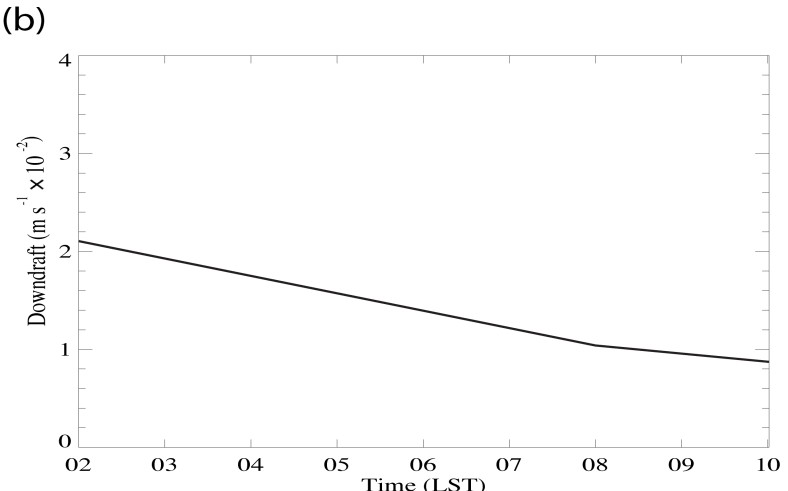

(c)

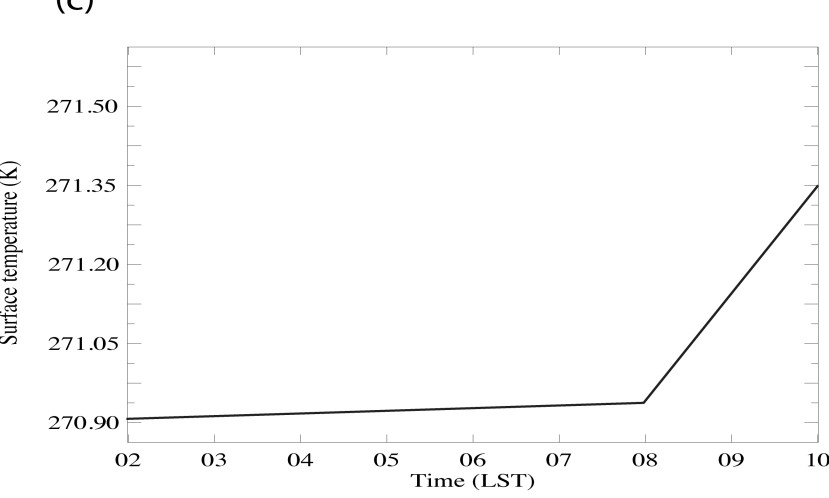



**Figure 2**

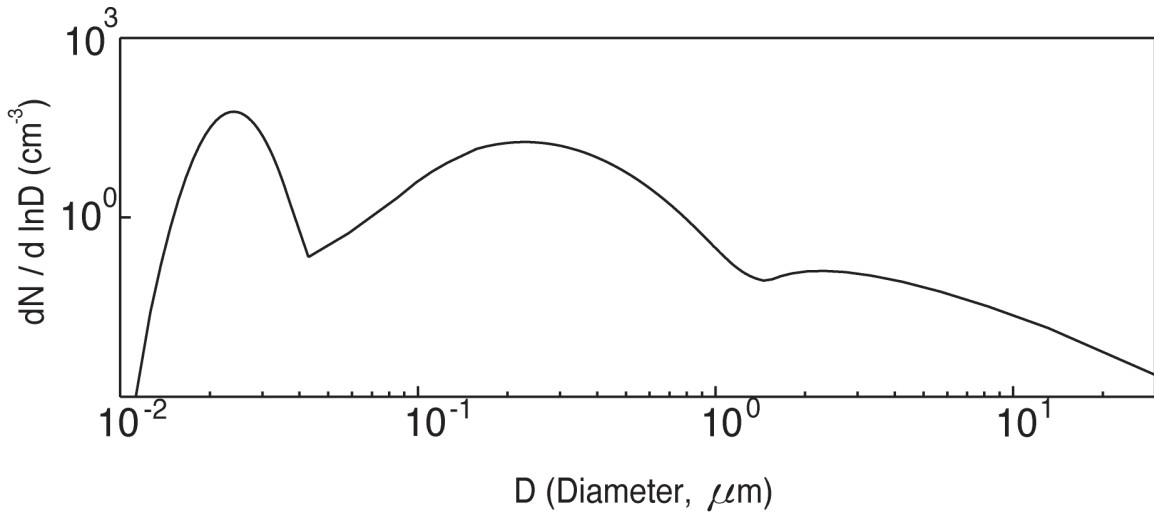


**Figure 3**













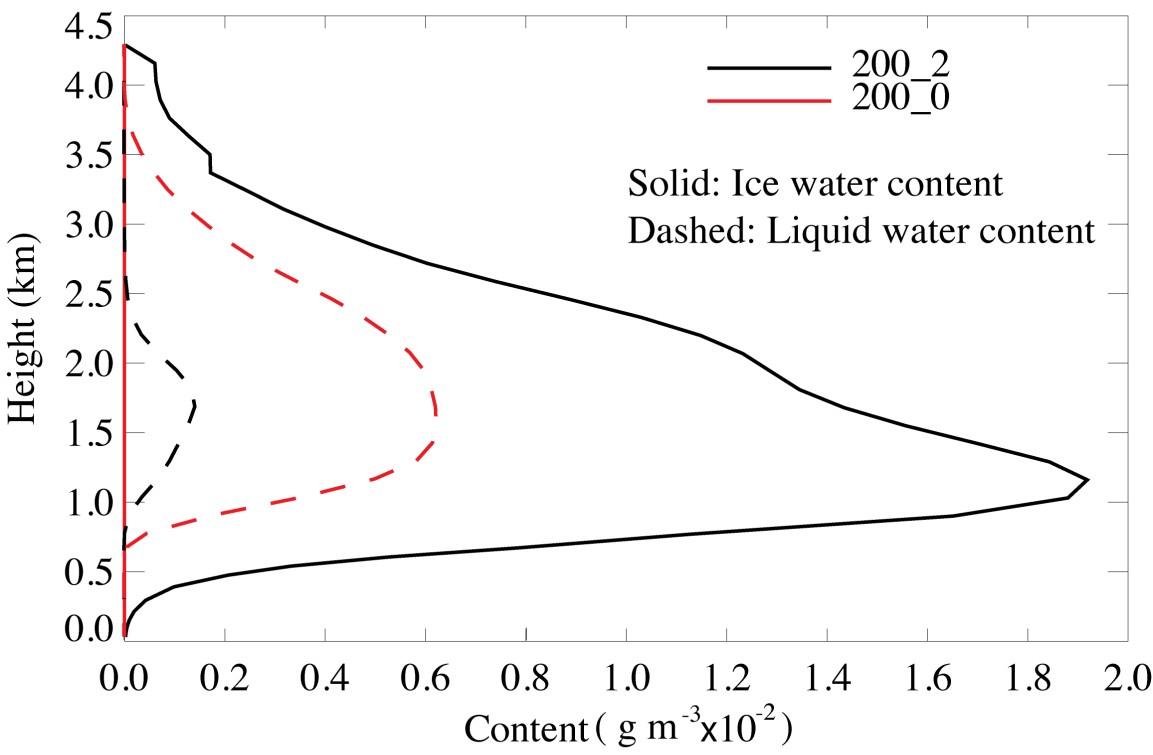

**Figure 4**

(a)

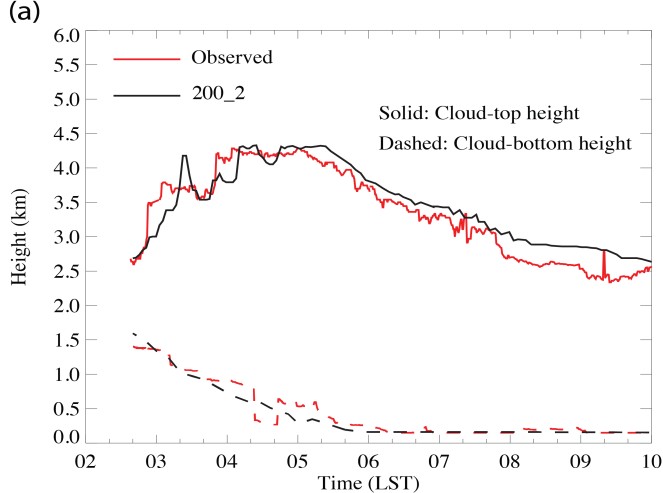

(b)

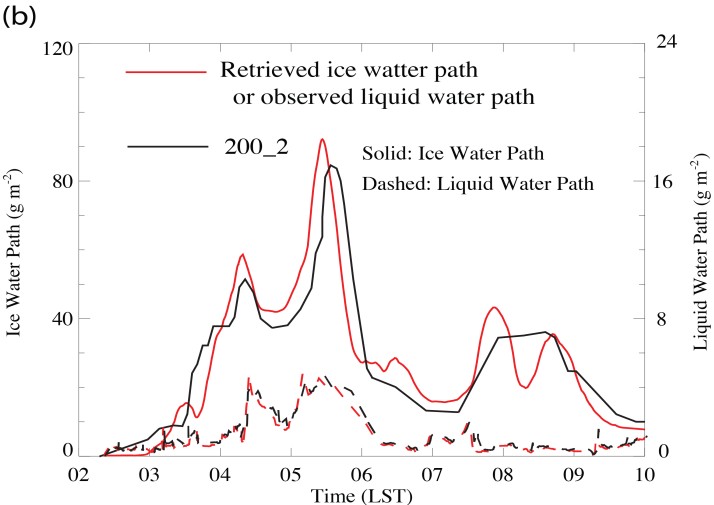

(c)

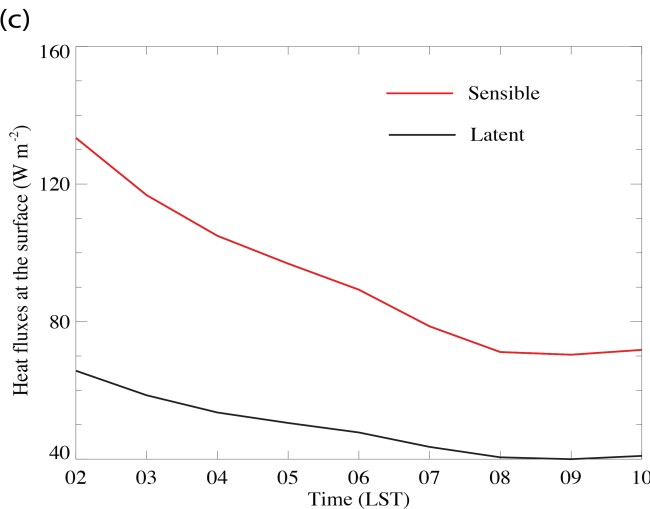


1519                    **Figure 5**

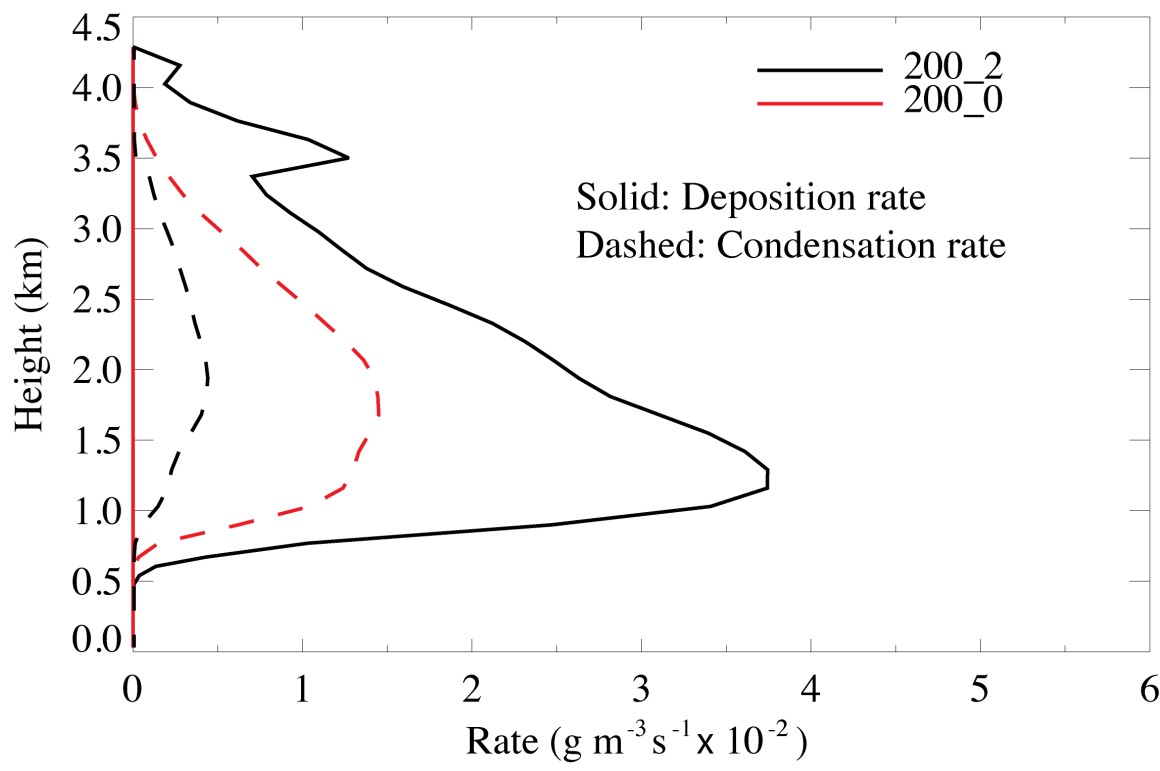

**Figure 6**











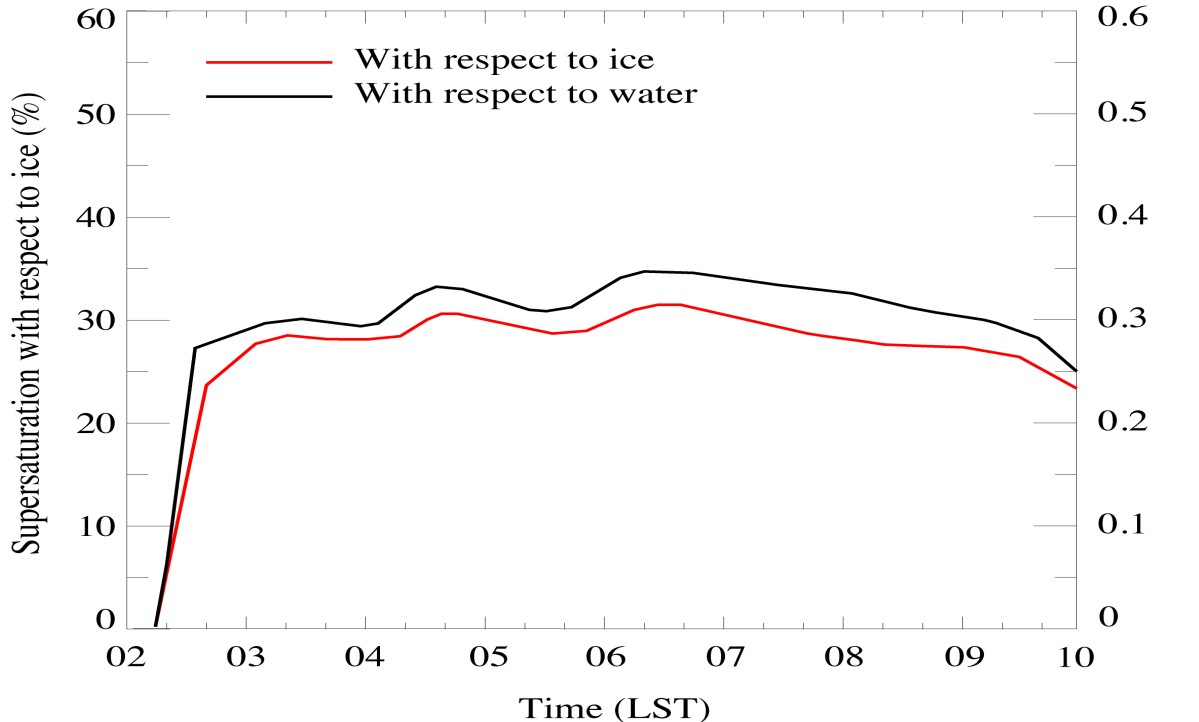


1531          **Figure 7**


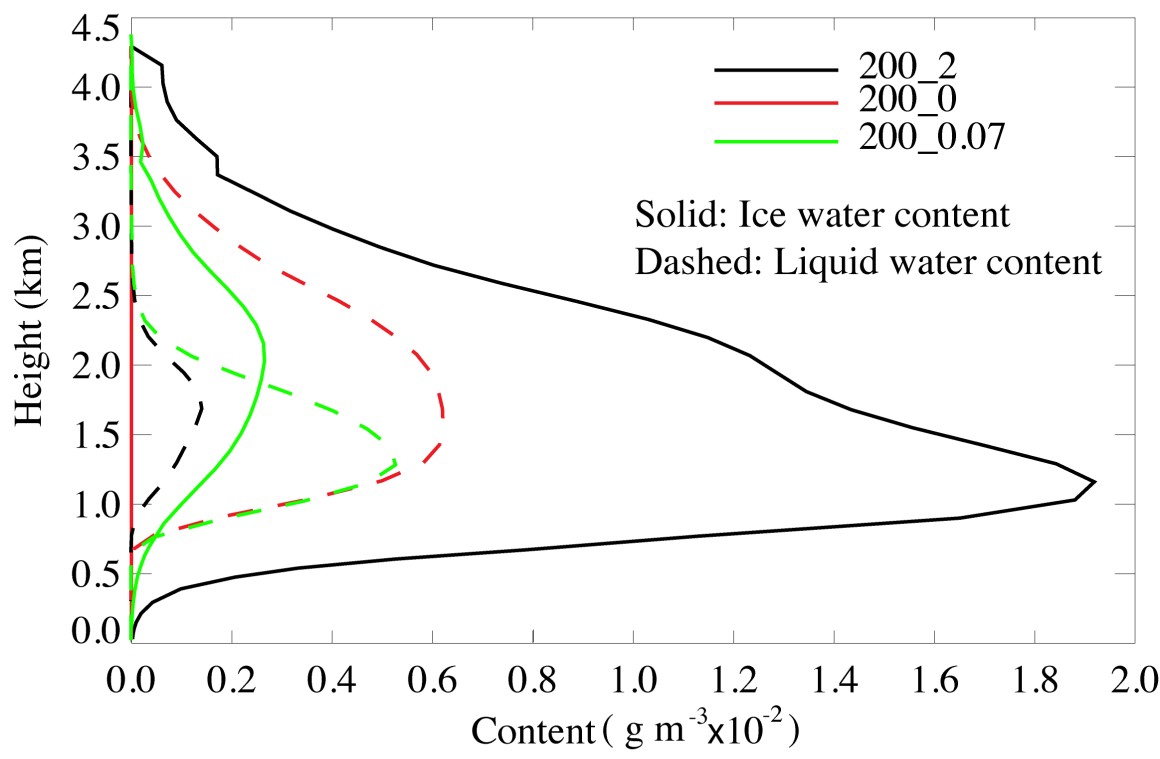

**Figure 8**










(a)

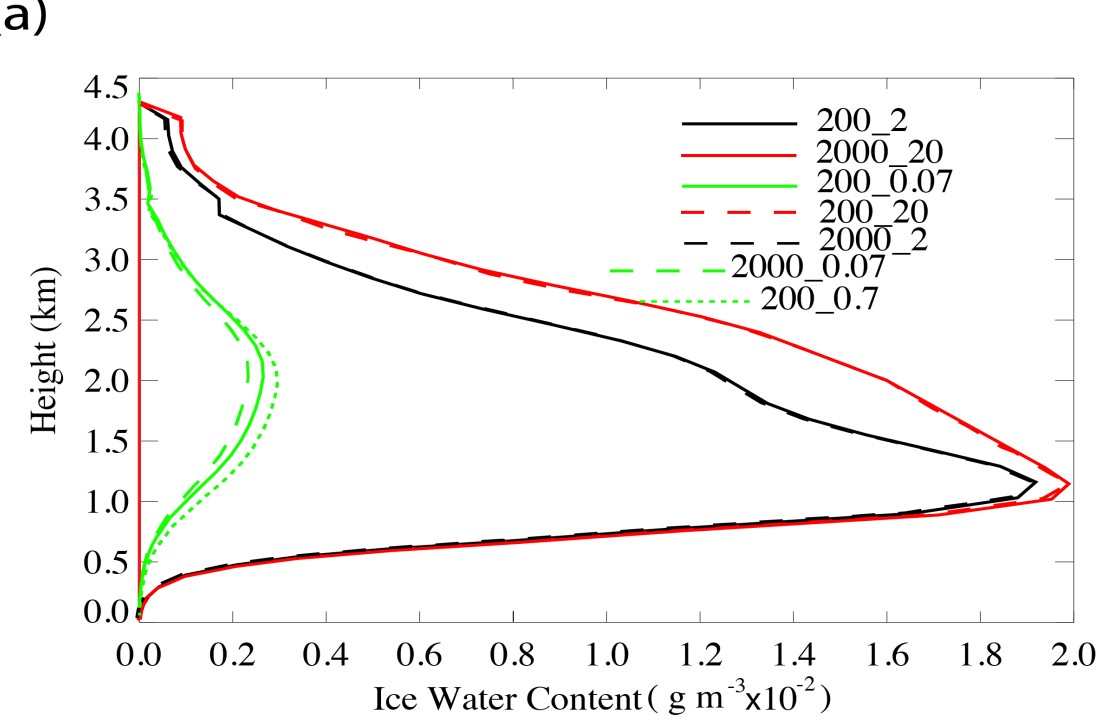

(b)

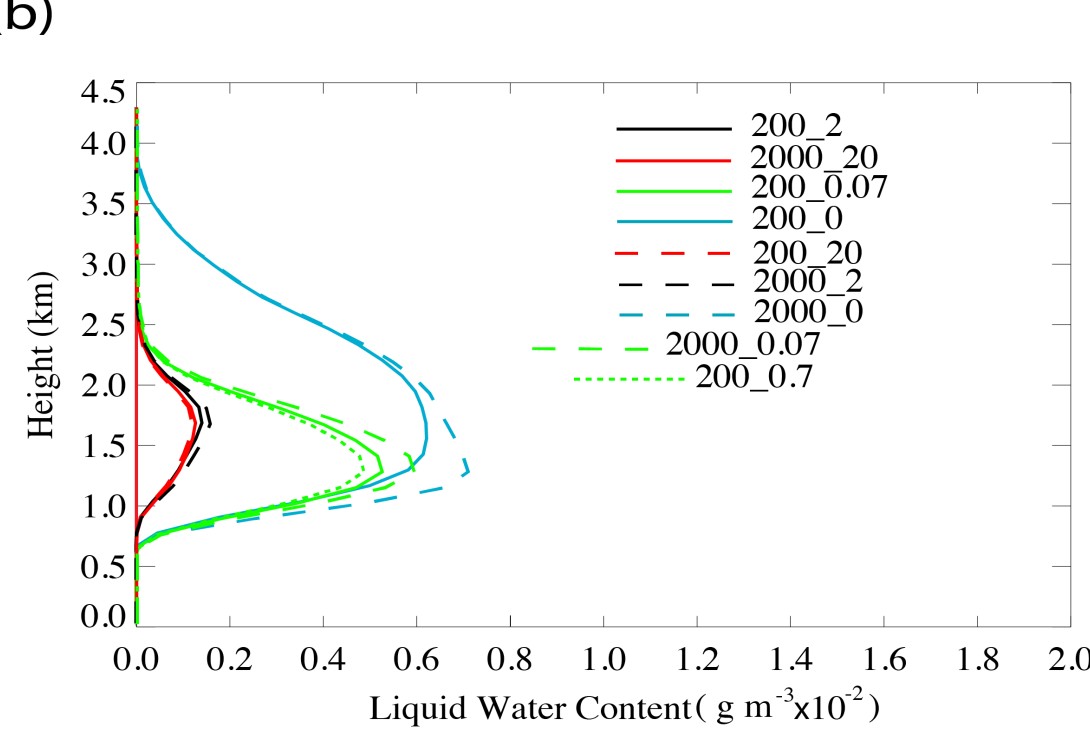



**Figure 9**

(a)

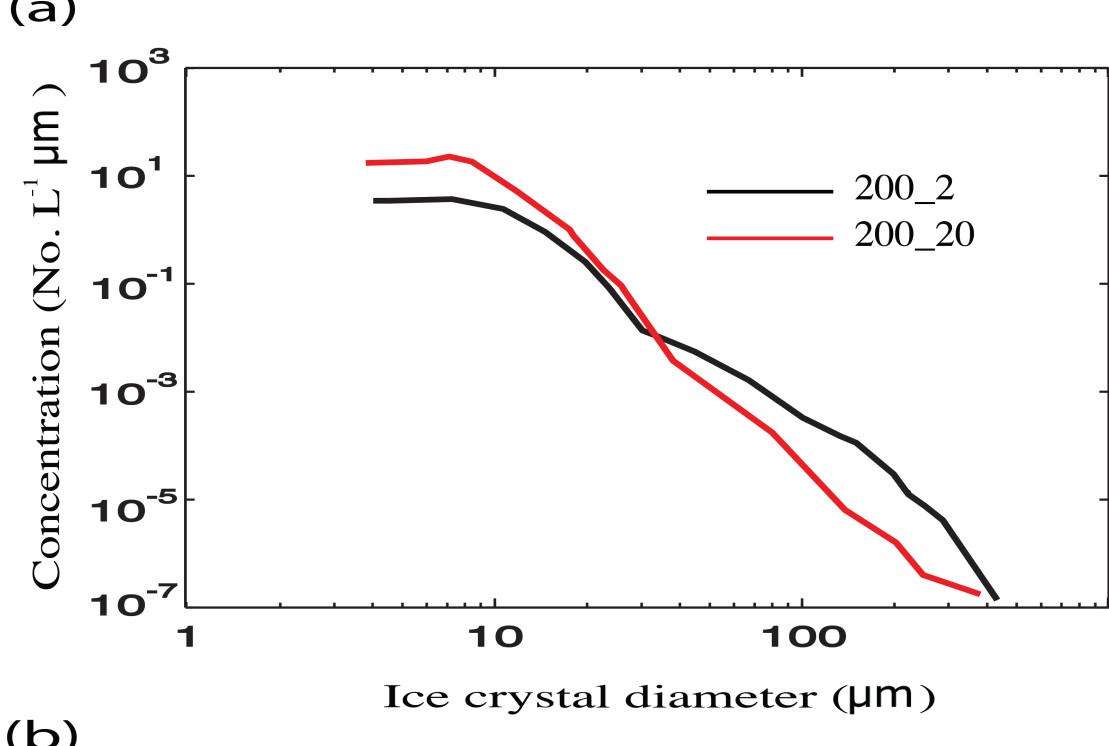

(b)

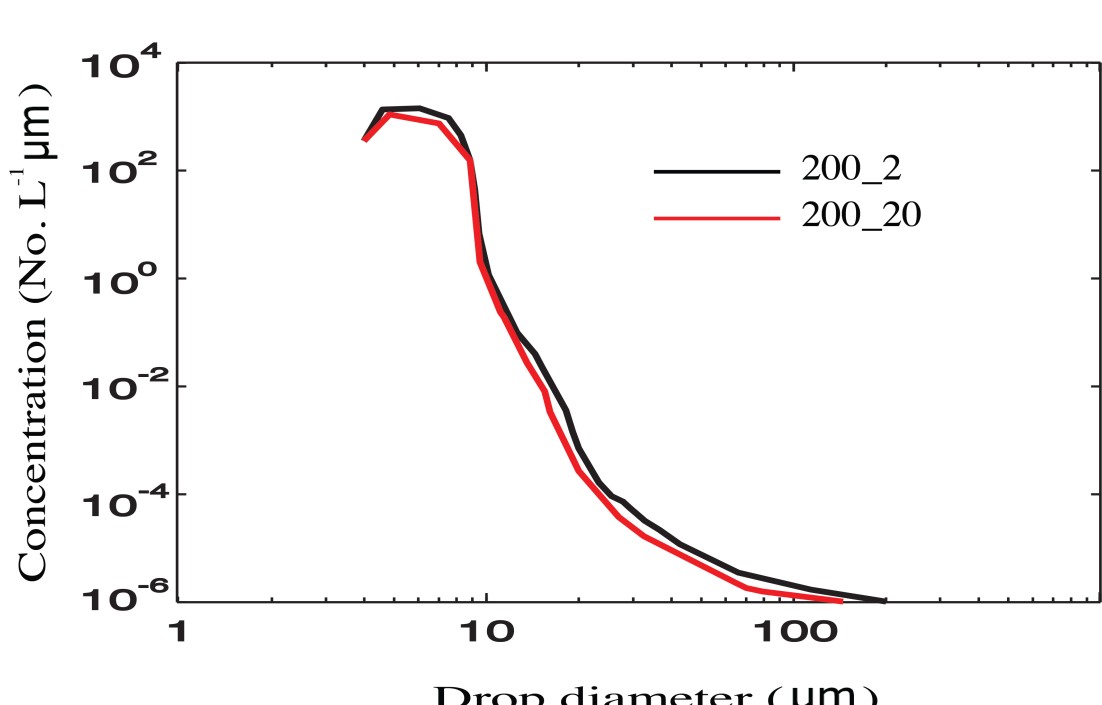


1545                    **Figure 10**

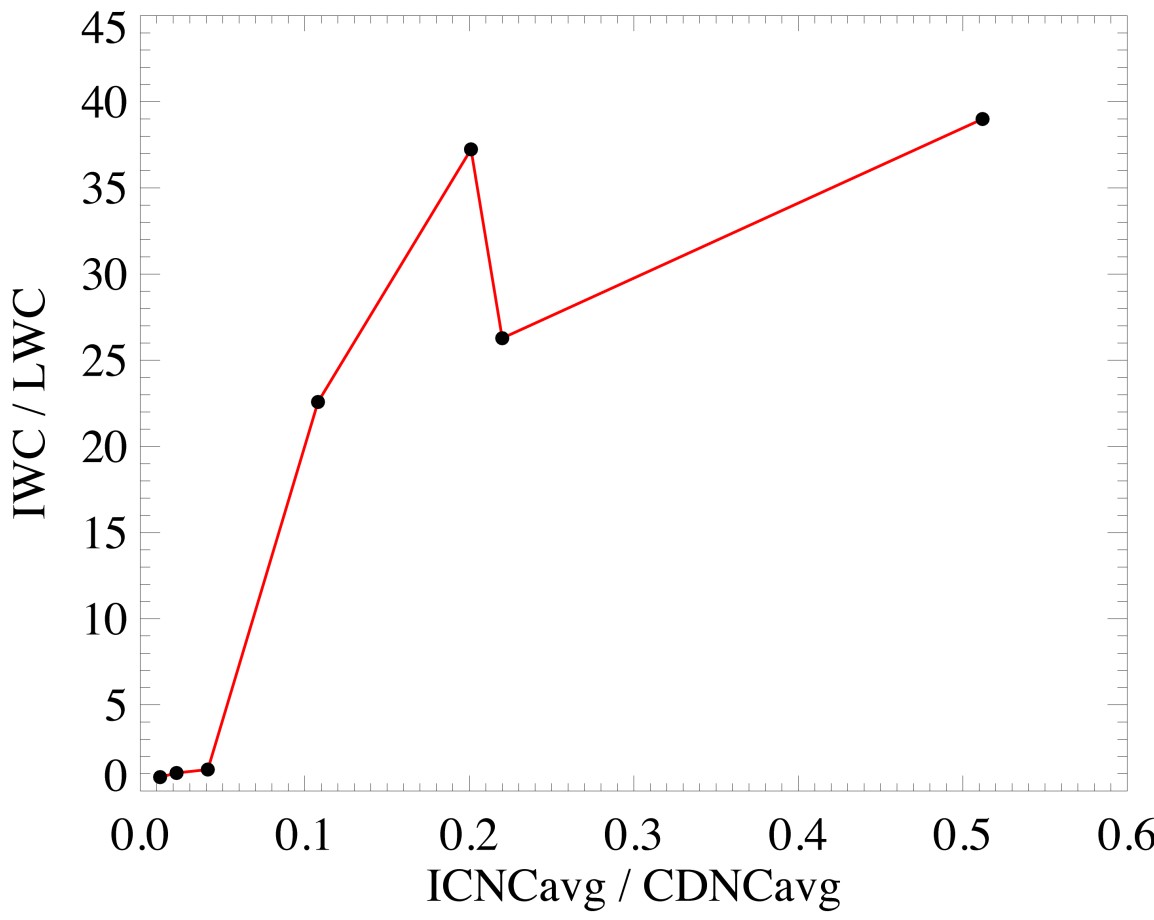

**Figure 11**


