# Peer review of "Role of a key microphysical factor in mixed-phase stratocumulus clouds and their interactions with aerosols"

_EGUsphere, 2023_

## Author Comment (AC1)

First of all, we appreciate the reviewer's comments and suggestion. In response to them, we have made relevant revisions to the manuscript. Listed below are our answers and the changes made to the manuscript according to the questions and suggestions given by the reviewer. The comment of the reviewer (in black) is listed and followed by our responses (in blue).

This study examines the impact of ICNC and CDNC on the properties of mixed-phase clouds using large-eddy simulations. However, I do not see new and exciting findings in this study, and some results are not convincing. Therefore, I recommend rejecting this paper in the current format.

Major comments:

1.  The authors seem to be not familiar with the literature in the field. The impact of ICNC and CDNC on the properties of mixed-phase clouds, which arises from the efficiency of INP and CCN, has been explored extensively in the past. Key conclusions of this study are very similar to some previous studies, e.g, by Solomon et al. 2018 (doi: 10.5194/acp-18-17047-2018). I do not see any new or exciting results out of this study.

This study talks about impacts of INP and CCN on latent-heat processes such as condensation and deposition thorough INP- and CCN-induced changes in ICNC and CDNC as sources of deposition and condensation, respectively. These impacts of INP and CCN on condensation and deposition eventually have a significant impact on the response of LWP and IWP to changing INP and CCN concentrations. In contrast to this, Soloman et al. (2018) have focused on CCN and INP impacts on cloud-top radiative cooling via aerosol-induced changes in droplet sizes. They have focused on the fact that these impacts on radiative cooling eventually alter dynamics and then LWP and IWP. This study focuses only on aerosol-induced changes in deposition and condensation excluding those changes on radiative cooling. In other words, the main driver of results in this study is aerosol-induced changes in deposition and condensation themselves but not aerosol-induced changes in radiative cooling. Hence, we believe that this study can be distinguished from Soloman et al. (2018). Regarding this, we repeated all of the previous simulations in the old manuscript by turning off radiative processes and comparisons between these repeated simulations and the previous simulations have shown that the qualitative nature of results in this study is robust to whether radiative processes, which include cloud-top radiative cooling, are considered for the simulations or not. This confirms that aerosol impacts on radiative cooling are not a main thrust for results in this study and the main thrust is aerosol-induced changes in ICNC, CDNC and associated deposition and condensation.

Also, want to mention that most of the previous studies of mixed-phase stratocumulus clouds have raised entrainment, detrainment, and hydrometeor sedimentation as important factors that control cloud mass and aerosol impacts on it (e.g., Albrecht, 1989; Ackerman, 2004: Ovchinnikov et al., 2011; Possner et al., 2017); cloud mass is represented by IWP and LWP. However, this study finds that entrainment, detrainment, and sedimentation are not important factors that control cloud mass, aerosol-induced changes in cloud mass, and their variation between different cloud systems at different locations. This study finds that CDNC, ICNC and then condensation and deposition are important factors for cloud mass, aerosol impacts on it and their variation between different cloud systems. Hence, this study can be distinguished even from other previous studies focusing on entrainment, detrainment, and hydrometeor sedimentation.

Regarding the argument here, the following is added:

(LL828-837 on p28)

Previous studies on mixed-phase stratocumulus clouds (e.g., Ovchinnikov et al., 2011; Possner et al., 2017; Solomon et al., 2018) have primarily focused on investigating the impacts of cloud-top radiative cooling, entrainment, and sedimentation of ice particles on these clouds, as well as their interactions with aerosols. However, there are a scarcity of studies that specifically examine the role of microphysical interactions, involving processes such as condensation and deposition, as well as factors like cloud-particle concentrations, between ice and liquid particles in mixed-phase stratocumulus clouds, and their interactions with aerosols as performed in this study. Therefore, our study contributes to a more comprehensive understanding of mixed-phase clouds and their intricate interplay with aerosols.

Regarding the simulations with radiation turned off, Section 3.3 is added.

To better put this study in the context of the previous studies of mixed-phase stratocumulus clouds, the following is added in introduction:

(LL136-141 on p5)

Choi et al. (2010 and 2014) and Zhang et al. (2019) have shown that as temperature lowers, IWC/LWC or IWP/LWP tends to increase and indicated that temperature is a primary environmental condition to explain the differences in IWC/LWC among different regions or clouds. However, Choi et al. (2010 and 2014) and Zhang et al. (2019) have not discussed process-level mechanisms that govern the role of temperature in those differences.

(LL175-187 on p6-7)

This comparison is based on Choi et al. (2010 and 2014) and Zhang et al. (2019) which have shown that temperature is an important factor which explains the differences in IWC/LWC among regions or clouds. Due to significant differences in latitudes, noticeable differences in the temperature of air are between the polar and midlatitude cases. Hence, through this comparison, this study looks at the role of temperature in those differences in IWC/LWC and associated aerosol-cloud interactions. More importantly than that, as a way of identifying process-level mechanisms that control the role of temperature, this study tests how ICNC/CDNC as the general factor is linked to the role of temperature, using the LES framework. Through this test, this study also identifies process-level mechanisms that control how ICNC/CDNC affects roles of ice processes in the differentiation between mixed-phase and warm clouds in terms of cloud development and its interactions with aerosols, and causes the variation of the differentiation between the cases of mixed-phase stratiform clouds.

2. The authors stated "ICNC/CDNC can be a simplified general factor that contributes to a more general understanding of mixed-phase clouds". If the authors can establish "a general principle" for the mixed-phase cloud, I think it would be very useful and this study would be worth for a publication. However, this argument is not convincing for the following reasons:

2.1. The author conducted nine idealized simulations of the mixed-phase clouds. It is not convincing to me how results from nine idealized simulations can be helpful to establish a general principle or a general parameterization for the mixed-phase cloud.

Just want to emphasize that as described in text, this study is only about an "attempt" to test a factor that can help us with the development of the general principle but not about the establishment of a perfect, complete general principle. To clarify this, the following is added:

(LL161-168 on p6)

Here, we want to emphasize that this study does not aim to gain a fully established general principle, but aims to test the factor that can be useful to move ahead on our path to a more complete general principle. Hence, this study should be regarded a steppingstone to the established principle, and should not be considered a perfect study that get us the fully established principle. Taking into account the fact that even attempts to provide general factors for the general principle have been rare, the fulfilment of the aim is likely to provide us with valuable preliminary information that streamlines the development of a more established general principle.

Just want to add that this study focuses on a factor, explaining differences in IWC/LWC among different clouds, as a steppingstone to the general principle, motivated by the fact that IWC/LWC plays an important role in cloud radiative properties and thus their climate feedbacks as discussed in Choi et al. (2010 and 2014). Choi et al. (2010 and 2014) and Zhang et al. (2019) have shown that temperature is an important factor which explains the differences in IWC/LWC among different clouds. However, Choi et al. (2010 and 2014) and Zhang et al. (2019) have not discussed process-level mechanisms that govern the role of temperature in those differences. Motivated by this, this study aims to find process-level mechanisms controlling the role of temperature in those differences by using the LES framework and to fulfill the aim, this study tests ICNC/CDNC which potentially can act as the factor or a general factor, explain the differences in IWC/LWC among different clouds and thus contribute to the development of the general principle in connection to the role of temperature. Regarding this, the following is added:

(LL132-141 on p5)

Mixed-phase stratocumulus clouds in different regions are known to have different IWC/LWC or IWP/LWP and aerosol-cloud interactions (e.g., Choi et al., 2010 and 2014; Zhang et al., 2019). Lots of factors such as environmental conditions, which can be represented by variables such as temperature, humidity and wind shear, can explain those differences. Choi et al. (2010 and 2014) and Zhang et al. (2019) have shown that as temperature lowers, IWC/LWC or IWP/LWP tends to increase and indicated that temperature is a primary environmental condition to explain the differences in IWC/LWC among different regions or clouds. However, Choi et al. (2010 and 2014) and Zhang et al. (2019) have not discussed process-level mechanisms that govern the role of temperature in those differences.

(LL169-187 on p6-7)

For the attempt, this study investigates a case of mixed-phase stratiform clouds in the polar region. Via the investigation, this study aims to identify process-level mechanisms that control the development of those clouds and their interactions with aerosols, and the impact of ice processes on the development and interactions using a large-eddy simulation (LES) framework. Then, this study compares the mechanisms in the case of polar clouds to those in a case of midlatitude clouds which have been examined by Lee et al. (2022). This comparison is based on Choi et al. (2010 and 2014) and Zhang et al. (2019) which have shown that temperature is an important factor which explains the differences in IWC/LWC among regions or clouds. Due to significant differences in latitudes, noticeable differences in the temperature of air are between the polar and midlatitude cases. Hence, through this comparison, this study looks at the role of

temperature in those differences in IWC/LWC and associated aerosol-cloud interactions. More importantly than that, as a way of identifying process-level mechanisms that control the role of temperature, this study tests how ICNC/CDNC as the general factor is linked to the role of temperature, using the LES framework. Through this test, this study also identifies process-level mechanisms that control how ICNC/CDNC affects roles of ice processes in the differentiation between mixed-phase and warm clouds in terms of cloud development and its interactions with aerosols, and causes the variation of the differentiation between the cases of mixed-phase stratiform clouds.

In addition, based on comments from both the reviewers, to improve the generality of findings of this study and thus to better streamline the establishment of the general principle, more simulations are performed as described in Sections 3.3, 4.1 and 4.2.

2.2. Observations are missing to justify the model setup and evaluate simulation results. For example, it said that "a system of mixed-phase stratocumulus clouds was observed to exist over a period between 02:00 local solar time (LST) and 20:00 LST on March 29th, 2017. On average, the bottom and top of these clouds are at ~400 m and ~3 km in altitude, respectively." Is there any ground-based observations to support this statement? See Fig. 1 in Solomon's paper as a good example.

The following is added:

(LL235-243 on p8-9)

In the Svalbard area, Norway, a system of mixed-phase stratocumulus clouds existed over the horizontal domain marked by a red rectangle in Figure 1 and a period between ~02:00 local solar time (LST) and 10:00 LST on March 29[th], 2017. These clouds are observed by ground radar and lidar and these radar and lidar are a part of the Cloudnet ground observation that is deployed at a location in the red rectangle. The Cloudnet ground observation is composed of a suite of instruments such as lidar, radar and radiometer and described in Hogan et al. (2006).  On average, the bottom and top of these clouds are at ~400 m and ~3 km in altitude, respectively, according to observation by those radar and lidar.

(LL343-358 on p12)

This study adopts the Cloudnet ground observation to evaluate the 200_2 run. Observed LWP is provided by radiometer.  The retrieval of IWP is performed by using radar reflectivity and lidar backscatter as described in Donovan et al. (2001), Donovan (2003) and Tinel et al. (2005). As mentioned above, observed cloud-bottom and -top heights are obtained from radar and lidar measurements. Simulated LWP and IWP, as

shown in Figure 4 and Table 2, are compared to the observed LWP and retrieved IWP, respectively. The average LWP over all time steps and grid columns is 1.23 in the 200_2 run and 1.12 in observation. The average IWP over all time steps and grid columns is 31.94 in the 200_2 run and 29.10 in retrieval. Cloud-bottom height, which is averaged over grid columns and time steps with non-zero cloud-bottom height, is 420 and 440 m in the 200_2 run and observation, respectively.  Cloud-top height, which is averaged over grid columns and time steps with non-zero cloud-top height, is 3.5 and 3.3 km in the 200_2 run and observation, respectively. Each of LWP, cloud-bottom and -top heights shows an ~10% difference between the 200_2 run and observation. IWP also shows an ~10% difference between the 200_2 run and retrieval. Thus, the 200_2 run is considered performed reasonably well for these variables.

Following the comment by the other reviewer, among the observed variables, the time series of the observed cloud-top height is compared to that of the simulated height as follows:

(LL359-366 on p12-13)

To provide additional or supplementary information of cloud development, the time evolution of the simulated and observed cloud-top height is shown together with the simulated evolution of the surface sensible and latent-heat fluxes in Supplementary Figure 1. This is based on the fact that the cloud-top height is considered a good indicative of cloud development and the surface fluxes are considered important parameters controlling the overall development of clouds. Simulated evolutions in Supplementary Figure 1 are from the 200_2 run. The cloud-top height increases between 02:00 and ~05:00 LST and after ~05:00 LST, it reduces gradually.

2.3. Model setup for the initial CCN and INP measurements is also not convincing. One weird result is the extremely high IWC for the control run. As far as I know, IWC in the mixed-phase stratocumulus cloud is usually much smaller than LWC. However, in the control run, IWC/LWC is 26.28, which is extremely high. Do you have any observations to support this result? Can you find any literature to support such high ratio exist in the mixed-phase cloud? If you cannot find observations to support such high IWC/LWC value, it means that the control run might not be setup correctly, and the goal to establish "a general principle" from those simulation results is not convincing.

First of all, as described above in our response to the reviewer's comment 2.2, the Cloudnet observation is used to evaluate the 200_2 run. As described in the response, the Cloudnet observation shows that the average observed/retrieved IWP is 29.10 and the average observed LWP is 1.12. Hence, the Cloudnet-based IWP/LWP is 25.98.

IWP/LWP in the 200_2 run is 25.96. This demonstrates that the simulated high IWP/LWP is well supported by the Cloudnet observation.

Moreover, Choi et al. (2014) have done work on the supercoold cloud fraction (SCF), which is equivalent to LWP/(LWP+IWP), using satellite-observed data collected over the period of ~5 years. As seen in Figure 1 in Choi et al. (2014), their work has shown that SCF can be lower than 0.05 and as low as 0.01 for the temperature range between -16 and -33 ℃. Regarding the temperature, as stated in the manuscript, the average air temperature immediately below the cloud base over the simulation period is -16 ℃ and the average air temperature immediately above the cloud top is -33 ℃ in the 200_2 run. Note that SCF in the 200_2 run in this study is 0.04.  Zhang et al. (2019) have also shown that for the temperature range between -16 and -33 ℃, SCF can be as low as ~0.03, though clouds with SCF below 0.05 are rare, based on ground observations in the Arctic area over a one-year period; for details, see Figure 7 in Zhang et al. (2019).

In association with Choi et al. (2014) and Zhang et al. (2019), the following is added:

(LL381-391 on p13)

Choi et al. (2014) and Zhang et al. (2019) have obtained the supercooled cloud fraction (SCF), which is basically the ratio of LWC to the sum of LWC and IWC and denoted by LWC/(LWC+IWC), using satellite- and ground-observed data collected over the period of ~5 years and ~1 year, respectively. Choi et al. (2014) have shown that SCF is as low as ~0.01 for the temperature range between -16 and -33 ℃. Zhang et al. (2019) have also shown that SCF is as low as ~0.03 for the same temperature range, although the occurrence of SCF of ~0.03 or lower is rare. Note that the average air temperature immediately below the cloud base and above the cloud top over the simulation period is -16 and -33 ℃, respectively, in the 200_2 run, and SCF in the 200_2 run is 0.04. Hence, based on Choi et al. (2014) and Zhang et al. (2019), we believe that SCF in the 200_2 run is observable and thus not that unrealistic, although it may not occur frequently.

In summary and conclusion, to explain reasoning behind the choice of the possibly rare polar case with SCF of 0.04, the following is added:

(LL774-777 on p26)

To gain the understanding efficiently, the polar case is chosen in a way to make stark contrast with the midlatitude case in terms of ICNC/CDNC and IWC/LWC. Although such polar cases may be uncommon, the stark contrast provides an opportunity to elucidate mechanisms that control the above-mentioned role of different ICNC/CDNC.

Here is one suggestion to improve the paper quality: whenever you refer to the observation (cloud, CCN, INP, LWP, IWP...), you should either cite a reference if the results are published or add it in the paper to support your statement. If you don't have those observations, you should provide reasonable assumptions. If results are quite different from previous studies (e.g., extremely high IWC/LWC), you should provide strong justifications.

As stated in our response above, observed LWP and observed/retrieved IWP are obtained from the Cloundnet observation and these LWP and IWP are compared to simulated counterparts as a way of evaluating the simulation. As stated in our response above, these observed LWP and observed/retrieved IWP also justify the simulated extremely high IWC/LWC. Associated text is added in the new manuscript as described above. Moreover, as stated in our response above, important cloud variables such as cloud-top and cloud-bottom heights are compared between observations and the 200_2 run. As stated above, this comparison demonstrates that the simulation of these cloud variables is performed reasonably well, and text about this is added in the new manuscript.

With respect to the observation of CCN and the associated assumption on INP, the following is added with associated text:

(LL263-271 on p9-10)

The properties of cloud condensation nuclei (CCN) such as the number concentration, size distribution and composition are measured in the domain (Tunved et al., 2013; Jung et al., 2018). The measurement indicates that on average, aerosol particles are an internal mixture of 70 % ammonium sulfate and 30 % organic compound. This mixture is assumed to represent aerosol chemical composition over the whole domain and simulation period for this study. The observed and averaged concentration of aerosols acting as CCN is ~200 cm$^{-3}$ over the simulation period. Based on this, 200 cm$^{-3}$ as an averaged concentration of aerosols acting as CCN is interpolated into all of grid points immediately above the surface at the first time step.

(LL285-288 on p10)

It is assumed that the properties of INP and CCN are not different except for concentrations. The concentration of aerosols acting as CCN is assumed to be 100 times higher than that acting as INP over grid points at the first time step based on a general difference in concentrations between CCN and INP (Pruppacher and Klett, 1978).

---

## Author Comment (AC2)

First of all, we appreciate the reviewer's comments and suggestion. In response to them, we have made relevant revisions to the manuscript. Listed below are our answers and the changes made to the manuscript according to the questions and suggestions given by the reviewer. The comment of the reviewer (in black) is listed and followed by our responses (in blue).

Review of the manuscript "EGUsphere-2023-862" entitled "Examination of varying mixed-phase stratocumulus clouds in terms of their properties, ice processes and aerosol-cloud interactions between polar and midlatitude cases: An attempt to propose a microphysical factor to explain the variation" written by Lee, Jung, Yoon, Um, Zheng, Guo, Manoj, and Song.

This study shows simulation results of idealized clouds that can occur in a polar region using the Weather Research and Forecasting model with a bin cloud microphysics scheme at a spatially fine resolution. The authors endeavor to examine how variations in cloud development are influenced by changes in cloud condensation nuclei (CCN) and ice-nucleating particle (INP) concentrations.

The authors propose that the ratio between the concentrations of ice crystals and cloud droplets may constitute a pivotal factor influencing cloud development. However, substantiating such a claim is challenging due to the scarcity of supporting evidence, and numerical experiments do not appear to be appropriately designed to substantiate this assertion. Furthermore, despite simulations of stratocumuli in a polar region are conducted, there is a notable absence of evidence regarding the adequacy of the model's cloud representation. Additionally, the experimental design lacks sufficient information for simulating stratocumuli adequately. The authors also introduce logical leaps in their arguments at several junctures. Consequently, for these reasons, the reviewer recommends against the publication of this paper. Detailed discussions on specific issues are provided below.

Major:

1. To convincingly demonstrate that the ratio between ICNC and CDNC (ICNC/CDNC) is indeed a critical factor in cloud development, as emphasized by the authors, it is imperative to systematically vary this ratio and conduct numerical experiments. This research, however, has scarcely undertaken such an approach. The ratios presented in Table 1 vary across all experiments, rendering it challenging to discern whether the differences revealed in each experiment stem simply from disparities in CCN and INP or from the ICNC/CDNC. To substantiate the authors' claims, it is essential that similar outcomes emerge in experiments with matched ICNC/CDNC but differing CCN and INP concentrations, thereby providing evidence for the significance of this ratio. Furthermore, as depicted in Figure 7, all four experiments—200_2, 2000_20, 200_20, and 2000_2—exhibit similar profiles for

IWC and LWC. However, the ICNC/CDNC ratios vary significantly among these experiments, ranging from 0.108 to 0.512. Therefore, based on these findings, it is challenging to assert that ICNC/CDNC is a critical factor.

We believe that the suggestion by the reviewer here that "ICNC/CDNC should be systematically varied and simulations should be performed accordingly" can be followed up by looking at simulations in Table 1. For the follow-up, simulations for mixed-phase clouds in Table 1 in the old manuscript are re-arranged as in Table 4. We see that ICNC/CDNC gradually increases from 0.012 in the 200_2_fac10_CCN10 run to 0.512 in the 200_20 run through values in between among the other runs in Table 4. We believe that this achieves the suggested systematic variation of ICNC/CDNC to the reasonable extent. For this variation of ICNC/CDNC, the variation of IWC/LWC among the simulations in Table 4 is examined as detailed in Section 4.1. We think that findings from this examination in Section 4.1 resolves the issue fairly well related to the reviewer's comment "Furthermore, as depicted in Figure 7, all four experiments— 200_2, 2000_20, 200_20, and 2000_2—exhibit similar profiles for IWC and LWC. However, the ICNC/CDNC ratios vary significantly among these experiments, ranging from 0.108 to 0.512. Therefore, based on these findings, it is challenging to assert that ICNC/CDNC is a critical factor"

Regarding the reviewer's comment that "the ratios presented in Table 1 vary across all experiments, rendering it challenging to discern whether the differences revealed in each experiment stem simply from disparities in CCN and INP or from the ICNC/CDNC. To substantiate the authors' claims, it is essential that similar outcomes emerge in experiments with matched ICNC/CDNC but differing CCN and INP concentrations, thereby providing evidence for the significance of this ratio", we performed additional simulations. For these additional simulations, as described in Section 4.2, the 200_2 run for the polar case and the 200_2_fac10 run representing the midlatitude case are selected among the runs in Section 4.1. For the additional simulations, each of these selected runs is repeated by varying the number concentration of aerosols acting as CCN and INP in a way that ICNCavg/CDNCavg does not vary as described in Section 4.2. As detailed in Section 4.2 and Table 5, these additional simulations demonstrate that the basic findings of the role of ICNC/CDNC in IWC/LWC from the comparison between the 200_2 and 200_2_fac10 runs in Section 3.1.2 are robust to whether the concentrations of aerosols acting as CCN and INP vary for a given ICNC/CDNC.

2. The authors present cloud development based solely on profiles of horizontally (and temporally) averaged IWC and LWC. However, it should be noted that cloud development is influenced by a multitude of physical quantities beyond these

metrics. Offering results exclusively in the form of averaged IWC and LWC profiles is not considered appropriate.

As stated in introduction in the old and new manuscripts, this study focuses on the relative proportion of liquid mass and ice mass, since this proportion plays an important role in cloud radiative properties and thus their climate feedbacks, according to Choi et al. (2010 and 2014) and Zhang et al. (2019). Choi et al. (2010 and 2014) and Zhang et al. (2019) quantified the relative proportion with liquid mass (ice mass) represented by LWC or LWP (IWC or IWP). With this quantification, these previous studies have found that the relative proportion, which can be noted to be "IWC/LWC" or "IWP/LWP", varies with regions and clouds. These previous studies have indicated that this variation is closely linked to temperature. However, they have not discussed process-level mechanisms that are associated with cloud-scale microphysics and dynamics and govern the role of temperature in the variation. We believe that understanding these mechanisms is important, since with this understanding, we can come up with process-oriented ideas about how to parameterize or represent the variation of IWC/LWC with varying clouds in varying regions or in varying temperature regimes. These ideas can eventually act as a valuable steppingstone to the development of a more general parameterization of clouds. As stated in introduction, the absence of the general parameterization has been considered one of the biggest obstacles to the better prediction of climate changes. Hence, pursuing those ideas as done by this study is worthy of research efforts.

In summary, this study is motivated by the findings of the variation of IWC/LWC with varying clouds in varying regions in the previous studies (e.g., Choi et al., 2010 and 2014; Zhang et al., 2019). Note that these previous studies basically deal with IWC, which is averaged over the time period and the specific area or region of interest, and LWC, which is also averaged over the time period and the specific area or region of interest, to examine the variation of the relative proportion of liquid mass, which is represented by the average LWC, and ice mass, which is represented by the average IWC. Following these previous studies, this study also adopts the time- and domain-averaged IWC and LWC to examine the variation of the relative proportion of liquid mass and ice mass; as stated in the manuscript, for the sake of simplicity, the averaged IWC over the averaged LWC is denoted by IWC/LWC for this study and IWC/LWC represents the relative proportion in this study, following the previous studies. Then, this study aims to identify process-level mechanisms which control the variation of none other than "IWC/LWC", as a representation of the relative proportion of liquid mass and ice mass, with varying clouds in varying regions or varying temperature regimes and have not been studied in the previous studies. Just want to emphasize that since the previous

studies have focused on the variation of the supercooled cloud fraction (SCF), which is basically a quantity regarding the ratio between the average LWC and the average IWC, this study focuses on IWC/LWC and associated process-level mechanisms. This study wants to make a continuity between itself and the previous studies and via the continuity, this study wants to further develop the findings of the previous studies in terms of IWC/LWC. For this, this study does not adopt other quantities and delve only into IWC/LWC and associated cloud-scale microphysics and dynamics. In this way, our findings can be in line with the previous studies and this can enrich both this study and the previous studies.

Although this study delves into IWC/LWC, to identify above-mentioned process-level mechanisms that are associated with cloud-scale microphysics and dynamics, this study examines other physical and dynamic quantities such as ICNC, CDNC, condensation, deposition, sedimentation and entrainment. Hence, authors do not believe that results here are offered exclusively in the form of the average IWC and LWC.

For your information, to identify the process-level mechanisms efficiently, this study chooses two cases, which are in stark contrast to each other in terms of temperature regimes where they reside and IWC/LWC, and compares them.

Regarding the argument above, the following is added:

(LL132-141 on p5)

Mixed-phase stratocumulus clouds in different regions are known to have different IWC/LWC or IWP/LWP and aerosol-cloud interactions (e.g., Choi et al., 2010 and 2014; Zhang et al., 2019). Lots of factors such as environmental conditions, which can be represented by variables such as temperature, humidity and wind shear, can explain those differences. Choi et al. (2010 and 2014) and Zhang et al. (2019) have shown that as temperature lowers, IWC/LWC or IWP/LWP tends to increase and indicated that temperature is a primary environmental condition to explain the differences in IWC/LWC among different regions or clouds. However, Choi et al. (2010 and 2014) and Zhang et al. (2019) have not discussed process-level mechanisms that govern the role of temperature in those differences.

(LL169-187 on p6-7)

For the attempt, this study investigates a case of mixed-phase stratiform clouds in the polar region. Via the investigation, this study aims to identify process-level mechanisms that control the development of those clouds and their interactions

with aerosols, and the impact of ice processes on the development and interactions using a large-eddy simulation (LES) framework. Then, this study compares the mechanisms in the case of polar clouds to those in a case of midlatitude clouds which have been examined by Lee et al. (2022). This comparison is based on Choi et al. (2010 and 2014) and Zhang et al. (2019) which have shown that temperature is an important factor which explains the differences in IWC/LWC among regions or clouds. Due to significant differences in latitudes, noticeable differences in the temperature of air are between the polar and midlatitude cases. Hence, through this comparison, this study looks at the role of temperature in those differences in IWC/LWC and associated aerosol-cloud interactions. More importantly than that, as a way of identifying process-level mechanisms that control the role of temperature, this study tests how ICNC/CDNC as the general factor is linked to the role of temperature, using the LES framework. Through this test, this study also identifies process-level mechanisms that control how ICNC/CDNC affects roles of ice processes in the differentiation between mixed-phase and warm clouds in terms of cloud development and its interactions with aerosols, and causes the variation of the differentiation between the cases of mixed-phase stratiform clouds.

(LL774-777 on p26)

To gain the understanding efficiently, the polar case is chosen in a way to make stark contrast with the midlatitude case in terms of ICNC/CDNC and IWC/LWC. Although such polar cases may be uncommon, the stark contrast provides an opportunity to elucidate mechanisms that control the above-mentioned role of different ICNC/CDNC.

3. The authors numerically simulate clouds in a polar region using the UM data as initial conditions. However, the adequacy of the model's cloud representation cannot be assessed as there is no comparison between the simulated clouds and those that can form under the actual conditions. For instance, in the control run, the averaged total water path of the simulated clouds is approximately 33 g m$^{-2}$. Without a comparison to the total water path of clouds formed under the corresponding conditions in the corresponding region, it is impossible to gauge the fidelity of the model's cloud simulation.

Comparisons between observation and the 200_2 run have been made in terms of cloud variables including IWP and LWP, the sum of which is the total cloud mass (or total water path), as follows:

(LL343-358 on p12)

This study adopts the Cloudnet ground observation to evaluate the 200_2 run. Observed LWP is provided by radiometer.  The retrieval of IWP is performed by using radar reflectivity and lidar backscatter as described in Donovan et al. (2001), Donovan (2003) and Tinel et al. (2005). As mentioned above, observed cloud-bottom and -top heights are obtained from radar and lidar measurements. Simulated LWP and IWP, as shown in Figure 4 and Table 2, are compared to the observed LWP and retrieved IWP, respectively. The average LWP over all time steps and grid columns is 1.23 in the 200_2 run and 1.12 in observation. The average IWP over all time steps and grid columns is 31.94 in the 200_2 run and 29.10 in retrieval. Cloud-bottom height, which is averaged over grid columns and time steps with non-zero cloud-bottom height, is 420 and 440 m in the 200_2 run and observation, respectively.  Cloud-top height, which is averaged over grid columns and time steps with non-zero cloud-top height, is 3.5 and 3.3 km in the 200_2 run and observation, respectively. Each of LWP, cloud-bottom and -top heights shows an ~10% difference between the 200_2 run and observation. IWP also shows an ~10% difference between the 200_2 run and retrieval. Thus, the 200_2 run is considered performed reasonably well for these variables.

4. According to Figure 2, the potential temperature in the near-surface atmosphere is approximately 257 K, which, assuming a pressure of 1000 hPa, corresponds to an extremely low temperature of approximately –16°C. However, the way SST (and/or surface heat fluxes) is prescribed under these atmospheric conditions remains undisclosed. For example, if the SST is assumed to be near 0°C, this would anticipate significant sensible heat flux. In this situation, it becomes vital to provide details on many cloud-related quantities and synoptic conditions, such as SST evolution, surface heat fluxes, large-scale subsidence, and cloud top height development.

The following is added:

(LL254-262 on p9)

Figure 2b shows the time evolution of the domain-averaged large-scale subsidence or downdraft in the reanalysis data and at the model top. The large-scale subsidence gradually reduces with time (Figure 2b). Figure 2c shows the time evolution of the domain-averaged surface temperature in the reanalysis data. This evolution of the surface temperature is mostly controlled by the sea surface temperature (SST) considering that most portion of the red-rectangle domain is accounted for by the ocean (Figure 1). Between ~06:00 LST around when the sun rises and ~08:00 LST, the surface temperature increases from -2.2 to -1.6 °C, and after that, it does not show significant increase or decrease (Figure 2c).

(LL359-369 on p12-13)

To provide additional or supplementary information of cloud development, the time evolution of the simulated and observed cloud-top height is shown together with the simulated evolution of the surface sensible and latent-heat fluxes in Supplementary Figure 1. This is based on the fact that the cloud-top height is considered a good indicative of cloud development and the surface fluxes are considered important parameters controlling the overall development of clouds. Simulated evolutions in Supplementary Figure 1 are from the 200_2 run. The cloud-top height increases between 02:00 and ~05:00 LST and after ~05:00 LST, it reduces gradually. The surface fluxes reduce with time, although the reduction rate of the fluxes starts to decrease around 08:00 LST in association with the increase in the surface temperature between ~06:00 and ~08:00 LST as shown in Figure 2c.

Supplementary Figure 1 or Figure S1 is as follows:

[Figure]

Figure S1. (a) The time series of observed and simulated cloud-top height. The simulated height is averaged over grid points with cloud tops at each time step in the 200_2 run. (b) The time series of the surface sensible and latent heat surfaces that are averaged over the domain in the 200_2 run.

5. In some points of the paper, the authors present arguments with substantial logical leaps. For instance, at L351, the authors assert that drop sedimentation "increases" total cloud mass. However, this assertion is neither logical nor supported by simulation results. The experimental findings merely demonstrate that in the comparison between the 200_2 run and the 200_2_noice run, the former exhibits greater total cloud mass and greater drop sedimentation. This can be attributed to the inherent fact that denser clouds yield more precipitation.

> Our argument at L351is based on the previous studies (e.g., Albrecht, 1989; Ackerman et al., 2004; Ovchinnikov et al., 2011; Possner et al., 2017) focusing on sedimentation and entrainment to explain the development of cloud mass in stratocumulus clouds. These previous studies have demonstrated that more (less) sedimentation of hydrometeors contributes to greater (smaller) loss of cloud mass by moving more (less) hydrometeors out of clouds when it comes to sedimentation itself or when we restrain our argument only to sedimentation; the argument at L351 is based on this. These studies have indicated that changes in sedimentation are a main driver that controls changes in cloud mass. The purpose of making the argument at L351and after L351 in the corresponding paragraph is to check on how important the much-studied sedimentation is in terms of changes in cloud mass as compared to condensation and deposition. Through this check, we want to show that contrary to the previous studies, sedimentation is not an important factor driving changes in cloud mass but condensation and deposition are the critical factor.

In addition, at L403, the authors describe that a higher IWC/LWC ratio in the 200_2 run than in the 200_2_noice run results in more water content (WC), but no logical rationale for this claim is provided, and it is evident only that the inclusion of INP leads to an increase in total water mass.

> Just want to start with the finding that WC in the polar mixed-phase clouds is higher than LWC in the polar warm clouds, while WC in the midlatitude mixed-phase clouds is lower than LWC in the midlatitude warm clouds. The main reason for this is that IWC in the polar mixed-phase clouds is much higher than LWC in the polar warm clouds, while IWC in the midlatitude mixed-phase clouds is smaller than LWC in the midlatitude warm clouds. Note that LWC in the polar (midlatitude) mixed-phase clouds is lower than that in the polar (midlatitude) warm clouds. Hence, we see that the higher IWC in the polar mixed-phase clouds than LWC in the polar warm clouds overcomes lower LWC in the polar mixed-phase clouds than that in the polar warm clouds to lead to higher WC in the polar mixed-phase clouds than that in the polar warm clouds, while lower

IWC in the midlatitude mixed-phase clouds than LWC in the midlatitude warm clouds is not able to overcome lower LWC in the midlatitude mixed-phase clouds than that in the midlatitude warm clouds to lead to lower WC in the midlatitude mixed-phase clouds than that in the midlatitude warm clouds. The higher IWC in the polar mixed-phase clouds than LWC in the polar warm clouds overcoming lower LWC in the polar mixed-phase clouds than that in the polar warm clouds is associated with a situation where IWC is much higher than LWC "in the polar mixed-phase clouds", thus there is a high IWC/LWC in the polar mixed-phase clouds, while the lower IWC in the midlatitude mixed-phase clouds than LWC in the midlatitude warm clouds not overcoming lower LWC in the midlatitude mixed-phase clouds than that in the midlatitude warm clouds is associated with a situation where the magnitude of IWC is similar to that of LWC "in the midlatitude mixed-phase clouds", thus, there is a low IWC/LWC, which is lower than IWC/LWC "in the polar mixed-phase clouds", in the midlatitude mixed-phase clouds.

In summary, reflecting the comment by the reviewer here, it is not logical to say that higher IWC/LWC itself in the 200_2 run (or in the polar mixed-phase clouds) than that in the midlatitude case (or in the midlatitude mixed-phase clouds) is a reason for WC in the 200_2 run which is greater than LWC in the warm clouds in the 200_2_noice run (or in the polar warm clouds), while it is not logical to say that lower IWC/LWC itself in the midlatitude case than that in the 200_2 run is a reason for WC in the midlatitude mixed-phase clouds which is lower than LWC in their warm-cloud counterpart (or in the midlatitude warm clouds); just want to note that the argument here is not about higher IWC/LWC in the 200_2 run than that in the 200_2_noice run, which is mentioned in the reviewer's comment here about "L403", but about the higher IWC/LWC in the 200_2 run than that in the midlatitude case. However, as described in the first paragraph of authors' response to the reviewer's comment here about "L403", the situation where higher IWC in the polar mixed-phase clouds than LWC in the polar warm clouds overcomes lower LWC in the polar mixed-phase clouds than that in the polar warm clouds is "associated" with higher IWC/LWC "in the 200_2 run than that in the midlatitude case", while the situation where the lower IWC in the midlatitude mixed-phase clouds than LWC in the midlatitude warm clouds is not able to overcome lower LWC in the midlatitude mixed-phase clouds than that in the midlatitude warm clouds is "associated" with lower IWC/LWC "in the midlatitude case than that in the 200_2 run". Based on the argument here and focusing on word "associated" in the argument, corresponding paragraphs are revised as follows:

(LL478-491 on p16-17)

Much higher IWC than LWC, which results in a much higher IWC/LWC in the polar case than in the midlatitude case, in the 200_2 run overcomes lower LWC in the 200_2 run than that in the 200_2_noice run, which leads to the greater total cloud mass in the 200_2 run than in the 200_2_noice run (Figure 4 and Table 2). However, IWC whose magnitude is similar to the magnitude of LWC, which results in a much lower IWC/LWC in the midlatitude case than in the polar case, in the midlatitude case is not able to overcome lower LWC in the midlatitude case than that in the midlatitude warm clouds, which leads to the greater total cloud mass in the midlatitude warm clouds than in the midlatitude case; here, the midlatitude warm clouds are generated by removing ice processes in the midlatitude case. This means that associated with higher ICNC/CDNC and IWC/LWC, ice processes enhance the total cloud mass for the polar case as compared to that for the polar warm-cloud counterpart. However, in the midlatitude case, associated with lower ICNC/CDNC and IWC/LWC, ice processes reduce the total cloud mass as compared to that for the midlatitude warm-cloud counterpart.

Minor:

1. I strongly recommend refine the writing. The authors excessively use sentences beginning with "There" and passive voice constructions.

   We revised sentences pointed out here by removing unnecessary "there" and passive voice constructions.

2. Although the authors have discussed the strong correlation between IWC and IWP in the early sections of the paper (e.g., Table 2), they consistently describe them as "IWC (IWP)" throughout the manuscript, which diminishes the readability of the paper. If the correlation is indeed evident, it is advisable to mention either IWC or IWP alone for clarity.

   Following the comment here, both a pair of IWC and LWC and that of IWP and LWP are mentioned only in the early part of the manuscript and only a pair of LWC and IWC are mentioned in text after the early part. To indicate this, the following is added:

   (LL376-380 on p13)

   Hence, mentioning both a pair of IWC and LWC and that of IWP and LWP is considered redundant, and mentioning either a pair of IWC and LWC or that of

IWP and LWP enhances the readability. Henceforth, IWC and LWC are chosen to be mentioned in text, although all of IWC, LWC, IWP and LWP are displayed in Tables 2 and 3.

---

## Author Response (AR2)

First of all, we appreciate the reviewer's comments and suggestions. In response to them, we have made relevant revisions to the manuscript. Listed below are our answers and the changes made to the manuscript according to those comments and suggestions. Each comment of the reviewer (black) below is followed by our response (blue).

I have carefully read the paper "Examination of varying. Iced-phase stratocumulus clouds in terms of their properties, ice processes and aerosol-cloud interactions between polar and midlatitude cases: An attempt to propose a microphysical factor to explain the variation". The paper proposes a series of simulations to compare the mid-latitude and polar cases of mixed-phase clouds. The motivations of the paper are indeed important, most of the studies are focused on warm clouds and not on mixed-phase clouds, as mentioned by the authors, even if there are more and more studies about them currently. The analysis is based on LES simulations to better understand the aerosol-cloud interactions. Interesting results have been derived from the study, for example ice processes increase the total cloud mass for the polar case compared to the polar warm cloud counterpart when ICNC/CDNC and IWC/LWC are high (and the opposite for mid-latitude clouds). The study is very comprehensive and the authors have done a lot of different simulations to explain their analysis, but I have concerns about the methods and the conclusions seem to be rushed or more clarification is needed. Also, I think clarifications are needed to better understand the paper as it is currently difficult to follow with the names of the simulations which are not clear to understand and remember what is associated with what. The current version needs to be rewritten to make it clearer. I do not have a strong expertise in models so I focused my review on the other parts. My concerns are detailed below.

General Comments
- Mixed phase clouds are different in the Arctic and mid-latitudes: In the Arctic, the liquid layer is usually on top of the ice layer, whereas in mid-latitudes mixed-phase clouds consist of pockets of liquid and ice within the clouds. These differences are not mentioned in the paper and have implications for ice processes, especially Wegener Bergeron process. How are mixed-phase clouds represented in the model? Can the authors comment on the possible biases arising from this?

As seen in Figures 4, 8 and 9, the liquid layer is not on the top of the ice layer in the case adopted by this study. Instead, the liquid layer is in the middle of the ice layer as seen in Figures 4, 8 and 9. Stated differently, the pockets of ice particles are not separated from those of liquid particles but mixed with those of liquid particles in the case adopted by this study as in the mid-latitude mixed-phase clouds including those in Lee et al. (2021). Hence, we do not believe that there are significant biases arising from differences between the Arctic and mid-latitude clouds in terms of the location of the liquid layer with respect to that of the ice layer. To indicate this, the following is added:

(LL495-498 on p17)

We hypothesized that ICNC/CDNC can be an important factor that determines above-described differences between the polar and midlatitude cases. Note that both in the polar and midlatitude cases, pockets of ice particles and those of liquid particles are mixed together instead of being separated from each other as seen in Figure 4 and Lee et al. (2021).

- It is usually difficult to disentangle the dynamical, microphysical changes and natural variability in model simulations, as both are bound in the parameterisation. I wonder if we can disentangle the effect of cloud evolution on ACI just by looking at different simulations, can the authors comment on that?

First of all, since the reviewer here talks about ACI, we want to limit the discussion here to the representative simulations for the polar case with varying aerosol concentrations, which are the standard simulations. The standard simulations are the 200_2, 2000_20, 2000_2 and 200_20 runs. Here, we want to emphasize that differences in these standard simulations are only in the initial aerosol concentrations and there are no other differences between them. For example, synoptic environmental conditions, as initial and boundary conditions, imposed on the simulations are identical among the standard simulations. Hence, differences in the simulated results regarding cloud processes and variables, and their evolutions are caused by differences in aerosol concentrations among the simulations. In particular, the differences in evolutions of clouds and associated cloud processes and variables can involve differences in feedback among cloud processes and variables, and aerosols. This

feedback is that differences in aerosol concentrations trigger those in cloud processes, such as cloud particle nucleation and precipitation, and then cloud variables, such as cloud-particle size and concentrations. Then, these aerosol-triggered differences in cloud processes and variables in turn affect aerosol concentrations and their impacts on clouds. In addition to that, aerosol-triggered changes in cloud processes and variables, such as cloud-particle nucleation, size and concentrations, and precipitation, affect latent-heat processes and dynamics (e.g., updrafts), and changed dynamics in turn affect latent-heat processes and cloud processes and variables, such as cloud-particle nucleation, size and concentrations, and precipitation. Here, we want to emphasize that all of the differences or changes in cloud processes and variables, such as cloud-particle nucleation, size and concentration, and precipitation, latent-heat processes and updrafts, their feedback among themselves and their feedback with aerosols are triggered by "differences or changes" in aerosol concentrations among the simulations. Here, changes in these cloud processes, variables and their feedback among themselves represent changes in cloud evolution, and changes in the feedback of cloud evolution with aerosols, which are associated with impacts of changes in cloud evolution on aerosols and their impact on clouds, represent impacts of changes in cloud evolution on ACI as termed by the reviewer here.

In this study, as mentioned, not only the changes in cloud evolution, but also the impacts of these changes in cloud evolution on ACI are triggered by changes in aerosol concentrations. In this study, we take interest in how changes in aerosol concentrations trigger these changes in cloud evolution and impacts of those changes in cloud evolution on ACI altogether. Stated differently, our main interest is in how all of these aerosol-triggered subsequent combined changes in cloud evolution and impacts of those changes in cloud evolution on ACI affect IWP and LWP, but not in the separation of aerosol-triggered changes in cloud evolution from impacts of those changes in cloud evolution on ACI. Hence, we perform analyses of results by looking at the averaged condensation, deposition, IWP, and LWP which are over the whole simulation period and thus final or combined products of aerosol-triggered changes in cloud evolution and impacts of these changes in cloud evolution on ACI.

- The study is difficult to read with all the different names of simulations that are not explicit about what they correspond to, and considering the number of simulation, it became difficult to figure out the corresponding simulation. I suggest that the names are renamed to better understand.

The other reviewer raised a similar issue as follows:

"Is stating INP meaningful in "noice" experiments? I suggest that the authors replace "200_2_noice" with "200_0" and "2000_2_noice" with "2000_0". In the same way, I suggest to replace "200_2_fac10" with "200_0.07" (also the others including "_fac10") as the names are neither intuitive nor consistent with others"

There are many ways to name simulations. In the paper, since we performed various simulations mostly by varying CCN and/or INP number concentrations, to reflect different CCN and/or INP number concentrations, due to the variation of CCN and/or INP number concentrations among simulations, CCN and INP number concentrations are explicitly put into the name of those simulations. We believe that this method of naming simulations is most effective in making readers understand how simulations are different from each other at a glance. Hence, we stick to the name method in the current manuscript. However, following the comment by the other reviewer above, some of names are changed.

- There are many sentences with the use of brackets to express two ideas in the same sentence, making it difficult to read. For example in lines 621, 632, 635,... I recommend simplifying those.

We minimized the use of those brackets by picking them, which are unnecessarily used, up, remove them and rephrase associated text.

- I did not find many details about the measurements (CCN measurements, Cloudnet...). Can the authors add information on where the data come from? How it is retrieved? Also, I would have liked the authors to show some plots comparing the model output with the observations when it is possible.

The details of CCN measurements are added as follows:

(LL279-289 on p10)

The measurement of the CCN concentration has been carried out at the Zeppelin research station in the domain, using the commercial droplet measurement technologies CCN counter with one column (CCNC-100), managed by the Korea Polar Research Institute, since year 2007. The CCNC-100 measures the CCN concentration at supersaturations of 0.2, 0.4, 0.6, 0.8 and 1% (Jung et al., 2018). The aerosol number size distribution is observed using a closed-loop differential mobility particle sizer (DMPS). The DMPS charges aerosol particles and exposing them into an electric field, which causes them to experience a force proportional to their electrical mobility, resulting in their classification according to size (Tunved et al., 2013). Aerosol composition is measured using aerosol mass spectrometry (AMS). The AMS measures the composition by vaporizing and ionizing aerosol particles.

The details of the Cloudnet observation are added as follows:

(LL241-251 on p9)

In the Svalbard area, Norway, a system of mixed-phase stratocumulus clouds existed over the horizontal domain marked by a red rectangle in Figure 1 and a period between 02:00 and 10:00 local solar time (LST) on March 29th, 2017. These clouds are observed by the Cloudnet ground observation that has been established to provide a systematic evaluation of clouds in forecast and climate models. The Cloudnet observation aims to establish a number of ground-based remote sensing sites, which would all be equipped with a specific array of instrumentation, using active sensors such as lidar and Dopplerized mm-wave radar, in order to provide vertical profiles of the main cloud variables (e.g., LWP and IWP), at high spatial and temporal resolution (Hogan et al., 2006). The Cloudnet observation provides data of important cloud variables such as LWP and IWP to the public and this study utilize these data.

(LL376-383 on p13)

This study adopts the Cloudnet ground observation to evaluate the 200_2 run. Observed LWP is provided by radiometer in the Cloudnet observation. The retrieval of IWP is performed by using radar reflectivity and lidar backscatter in the Cloudnet observation as described in Donovan et al. (2001), Donovan and Lammeren (2001), Donovan (2003) and Tinel et al. (2005). In the retrieval, the lidar signal and radar reflectivity profiles are combined and inverted using a combined lidar/radar equation as a function of the light extinction coefficient and radar reflectivity. The combined equation is detailed in Donovan and Lammeren (2001).

More plots comparing the model output with the observations are added in Figure 5.

- On line 721, the authors say that ICNC/CDNC increases with IWC/LWC and base their analysis on this. But this is not true for the 200_2 simulations, as the authors state. So I do not understand why they infer results and generalize it when one of the situations does not work. Can the authors comment on that?

Just want to emphasize that as described in text, this study is only about an "attempt" to test a factor which is ICNC/CDNC and can help us with the development of the general principle about the variation of IWC/LWC among different cloud systems but not about the establishment of a perfect, complete general principle as already stated in text as follows:

(LL162-169 on p6)

Here, we want to emphasize that this study does not aim to gain a fully established general principle, but aims to test the factor that can be useful to move ahead on our path to a more complete general principle. Hence, this study should be regarded a steppingstone to the established principle, and should not be considered a perfect study that get us the fully established principle. Taking into account the fact that even attempts to provide general factors for the general principle have been rare, the fulfilment of the aim is likely to provide us with valuable preliminary information that streamlines the development of a more established general principle.

Hence, this study does not aim to come up with a perfect, complete general principle which can be applicable without any limit or conditions. Instead, this study aims to come up with a principle which can be applicable with a certain degree of limit or conditions, though this study tries to minimize the magnitude of the degree. As stated in text, we want to emphasize that "Even attempts to provide general factors for the general principle have been rare". Thus, even a principle with a certain level of limit can provide us with valuable preliminary information that streamlines the development of a more established general principle. This research philosophy for this study is stated as follows:

(LL940-956 on p31-32)

This study finds that the relation between ICNC/CDNC and IWC/LWC is highly non-linear. This high non-linearity is closely linked to how the number concentrations of CCN and INP, and associated ICNC/CDNC change. For a specific situation where the ICNC/CDNC variation is relatively small and both the number concentrations of CCN and INP reduce, the increase in ICNC/CDNC can reduce IWC/LWC, although it is found that as a whole, the increase in ICNC/CDNC enhances IWC/LWC. Hence, mechanisms identified in this study, especially regarding the use of ICNC/CDNC as a simplified and useful tool to explain differences in IWC/LWC among different cloud systems, are not complete and entirely general.  In addition, results in this study are from only two cases in two specific locations in the midlatitude and Arctic regions and the more generalization of these results in this study merits more case studies over more locations in those regions, for example, in terms of above-mentioned sedimentation intensity, different factors (e.g., environmental factors) other than ICNC/CDNC, different sources and advection of aerosols, the magnitude of the variation of ICNC/CDNC and the way number concentrations of CCN and INP vary. Hence, findings particularly about relations between ICNC/CDNC and IWC/LWC in this study should be considered preliminary ones that initiate future work to streamline the development of the general parameterizations.

The limit or conditions to consider for the use of ICNC/CDNC to explain the IWC/LWC variation are described as follows:

(LL803-814 on p27)

The high-degree non-linearity in the variation of IWC/LWC is epitomized by the 1706 percent increase in IWC/LWC for the 163 percent increase in $ICNC_{avg}/CDNC_{avg}$ from the 200_0.7 run to the 2000_2 run. This 1706 percent increase in IWC/LWC is induced by increases in both the initial number concentrations of CCN and INP between the runs (Table 1). In other transition from a simulation in a row to that in the next row in Table 4, there are decreases in both the initial number concentrations of CCN and INP, or there is either a change in the initial number condensation of CCN or INP. When either the initial concentration of CCN or INP changes in the transition, less than a 100% increase in IWC/LWC is shown. The decreases in both the initial number concentrations of CCN and INP, which are from the 2000_20 run to the 200_2 run, result in the decrease in IWC/LWC. Hence, depending on how the initial number concentrations of CCN and INP change, the magnitude and sign of the change in IWC/LWC can vary substantially.

(LL793-802 on p26-27)

Here, it is notable that the percentage difference in $ICNC_{avg}/CDNC_{avg}$ is ~9% between the 2000_20 and 200_2 runs and the smallest among those differences in Table 4. The other differences are larger than 80%. Hence, the percentage difference in $ICNC_{avg}/CDNC_{avg}$ for a pair of the 2000_20 and 200_2 runs is at least ~one order of magnitude smaller than that for the other pairs of the runs in Table 4. This means that findings from the comparison between the 200_2 and 200_0.07 runs are not suitable to explain the variation of IWC/LWC among clouds when the variation of ICNC/CDNC is relatively insignificant. According to Table 4, it seems that the variation of ICNC/CDNC should be greater than a critical value above which those findings are useful to account for the IWC/LWC variation among clouds.

- The authors infer different glaciation processes from the change in ICNC/CDNC, but I am not

convinced that this is not directly related to the temperature difference between the Arctic and mid-latitudes. Can the authors make the argument explicit to rule out the temperature dependence?

Yes, it is true that in the comparison between the polar and midlatitude cases in this study, differences in ICNC/CDNC are mainly caused by differences in temperature between the cases as described in text:

(LL514-533 on p17-18)

ICNCavg/CDNCavg is 0.22 in the control run (i.e., the 200_2 run) for the polar case and 0.019 in the control run for the midlatitude case which is described in Lee et al. (2021). Henceforth, the control run for the midlatitude case is referred to as the control-midlatitude run. ICNCavg/CDNCavg is ~one order of magnitude higher for the polar case than for the midlatitude case. This is despite the fact that the ratio of the initial number concentration of aerosols acting as INP to that of acting as CCN is identical between the 200_2 and control-midlatitude runs. In addition, identical model, model setup such as vertical resolutions, and source of reanalysis data are used between the 200_2 and control-midlatitude runs, although there are differences in environmental conditions (e.g., temperature), cloud macrophysical variables such as cloud-top height and horizontal resolutions between the runs. Here, while taking these similarities and differences into account, we hypothesize that the significant differences in ICNCavg/CDNCavg between runs are mainly due to the fact that ice nucleation strongly depends on air temperature (Prappucher and Klett, 1978). When supercooling is stronger, in general, more ice crystals are nucleated for a given group of aerosols acting as INP. The average air temperature immediately below the cloud base over the simulation period is -16 ºC in the 200_2 run and -5 ºC in the control-midlatitude run. The average air temperature immediately above the cloud top is -33 ºC in the 200_2 run and -15 ºC in the control-midlatitude run. Hence, supercooling is greater and this contributes to the higher ICNCavg/CDNCavg in the polar case than in the midlatitude case.

However, it should be noted that although ICNC/CDNC differences are made by temperature differences between the cases in this study, for other cloud systems, we mentioned that ICNC/CDNC differences between those other cloud systems can be made by factors other than temperature differences as follows:

(LL897-904 on p30)

It should be emphasized that although this study mentions air temperature as a factor that affects ICNC/CDNC, ICNC/CDNC can be affected by other factors such as sources of aerosols acting as INP and those acting as CCN, and/or the advection of those aerosols. Hence, even for cloud systems that develop with a similar air-temperature condition, for example, when those systems are affected by different sources of aerosols and/or their different advection, they are likely to have different ICNC/CDNC, IWC/LWC, relative importance of impacts of INP on IWC and LWC as compared to those impacts of CCN, and relation between warm and mixed-phase clouds.

To effectively discern the impact of varying ICNC/CDNC on the variation of IWC/LWC across diverse cloud systems, this study specifically selected two cases characterized by notable temperature disparities, thereby yielding significant differences in ICNC/CDNC. However, as mentioned in text, discrepancies in ICNC/CDNC can arise from factors such as differences in sources and advection of aerosols beyond temperature differences between different cloud systems. While this study utilized temperature differentials to identify cases exhibiting significant disparities in ICNC/CDNC, its primary objective lies in comprehending the inherent role of ICNC/CDNC variations themselves in the discrepancies observed in IWC/LWC across diverse cloud systems, regardless of whether disparities in ICNC/CDNC are caused by those in temperature or in other factors, but not in how the disparities in ICNC/CDNC are caused. With the fulfillment of this objective, as long as we take interest in the role of ICNC/CDNC variations themselves among different cloud systems, the insights gleaned from this study regarding the influence of varying ICNC/CDNC on IWC/LWC can be extrapolated to scenarios where factors other than temperature differences contribute to discrepancies in ICNC/CDNC among different cloud systems. To clarify this, the following is added:

(LL904-910 on p30)

Regarding factors, which affect ICNC/CDNC, such as sources and advection of aerosols together with temperature , it should be noted that while this study utilizes differences in temperature among those factors to identify cases exhibiting significant disparities in ICNC/CDNC, its primary objective does not lie in the role of temperature differences in disparities in ICNC/CDNC, but in comprehending the inherent role of ICNC/CDNC variations themselves in the discrepancies observed, for example, in IWC/LWC, across diverse cloud systems.

- The nucleation mode between the Arctic and the mid-latitudes is very different due to the moisture regime. The Arctic region is less prone to immersion freezing and condensation freezing compared to the mid-latitudes because of the expected supersaturation of water vapour with respect to liquid. Is this related to the authors' conclusion? If so, would the information on relative humidity be helpful?

Yes, differences in dominant nucleation modes between the Arctic and mid-latitude cases can be considered to better understand findings from this study microphysically. However, we just want to emphasize that similar to what is mentioned in our response above, the primary objective of this study does not lie in the role of differences in factors such as not only aerosol sources and advection but also freezing modes as pointed out by the reviewer here, in ICNC/CDNC, but in comprehending the inherent role of ICNC/CDNC variations themselves in the discrepancies observed, for example, in IWC/LWC, across diverse cloud systems. Hence, we do not pursue the understanding of roles of freezing modes in differences in ICNC/CDNC between the cases.

- I think the authors' conclusion is too general (e.g., when the authors refer to polar regions, it actually corresponds to a region around Svalbard, which is a region of the Arctic). The conclusion should not be so promising because it could mislead readers.

The following is added:

(LL948-956 on p31-32)

In addition, results in this study are from only two cases in two specific locations in the midlatitude and Arctic regions and the more generalization of these results in this study merits more case studies over more locations in those regions, for example, in terms of above-mentioned sedimentation intensity, different factors (e.g., environmental factors) other than ICNC/CDNC, different sources and advection of aerosols, the magnitude of the variation of ICNC/CDNC and the way number concentrations of CCN and INP vary. Hence, findings particularly about relations between ICNC/CDNC and IWC/LWC in this study should be considered preliminary ones that initiate future work to streamline the development of the general parameterizations.

Specific comments
-Line 94: Mixed-phase clouds can also be found in convective clouds in the tropics.

The following is added:

(LL95-97 on p4)

When mixed-phase stratiform clouds are associated with convective clouds, they can form even in the tropical region.

- Line 105: "The radiative properties of liquid particles are substantially different from those of ice particles", I slightly disagree with this sentence. It is true that the radiative properties are not exactly the same, but the main difference is that ice crystals are larger and less numerous than cloud droplets (due to lower concentration of INP compare to CCN), so ice clouds tend to be less reflective than liquid clouds, and ice crystals precipitate, potentially altering the cloud lifetime.

The text pointed out here is removed and just added more references that indicate the importance of the relative proportion of liquid mass and ice mass in cloud radiative properties and in feedback between these radiative properties and changing climate, as follows:

(LL103-107 on p4)

The relative proportion of liquid mass, which can be represented by liquid-water content (LWC) or liquid-water path (LWP), and ice mass, which can be represented by ice-water content (IWC) or ice-water path (IWP), in mixed-phase stratiform clouds plays a critical role in cloud radiative properties and thus their climate feedbacks (Tsushima et al., 2006; Choi et al., 2010 and 2014; Gettelman et al., 2012; Zhang et al., 2019).

- Lines 134-136: "Many factors, such as environmental conditions..." The spatial distribution of the liquid and ice phases is also important as mentioned in the general comments.

See our response to the first general comment about the relative locations of liquid and ice phases. As seen in the response, the relative locations can be considered not to play an important role in differences in IWC/LWC between the two cases. However, in text pointed out here, we indicated the potential role of the relative locations of the layers of ice crystals and droplets or the spatial distribution of the liquid and ice particles in differences in IWC/LWC among different clouds as follows:

(LL134-137 on p5)

Lots of factors such as environmental conditions, which can be represented by variables such as temperature, humidity and wind shear, and macrophysical factors one of which is the relative locations of ice-crystal and droplet layers, can explain those differences.

- Line 165: "Get us the fully established principle": What would be needed to get the fully established principle?

As described in "Summary and conclusions", the more fully established principle may need to take factors, such as sedimentation, wind shear, stability, aerosol sources, aerosol advection and the variation of ICNC/CDNC in the context of how CCN and INP concentrations vary, into account to explain differences in IWC/LWC (or IWP/LWP) and aerosol-cloud interactions among different cloud systems. To indicate this, the corresponding text in "Summary and conclusions" is revised or additional text is added to "Summary and conclusions" as follows:

(LL880-885 on p29)

Thus, different mechanisms controlling the differentiation can be expected regarding factors such as stability and wind shear as compared to ICNC/CDNC. The examination of these different mechanisms among stability, wind shear and ICNC/CDNC deserves future study for more comprehensive understanding of the differentiation or for an above-mentioned more fully established general principle explaining the differentiation.

(LL893-896 on p30)

For more generalization of results here as a way to the more fully established general principle, this potential role of sedimentation needs to be investigated by performing more case studies involving cases with strong precipitation in the future.

(LL940-956 on p31-32)

This study finds that the relation between ICNC/CDNC and IWC/LWC is highly non-linear. This high non-linearity is closely linked to how the number concentrations of CCN and INP, and associated ICNC/CDNC change. For a specific situation where the ICNC/CDNC variation is relatively small and both the number concentrations of CCN and INP reduce, the increase in ICNC/CDNC can reduce IWC/LWC, although it is found that as a whole, the increase in ICNC/CDNC enhances IWC/LWC. Hence, mechanisms identified in this study, especially regarding the use of ICNC/CDNC as a simplified and useful tool to explain differences in IWC/LWC among different cloud systems, are not complete and entirely general. In addition, results in this study are from only two cases in two specific locations in the midlatitude and Arctic regions and the more generalization of these results in this study merits

more case studies over more locations in those regions, for example, in terms of above-mentioned sedimentation intensity, different factors (e.g., environmental factors) other than ICNC/CDNC, different sources and advection of aerosols, the magnitude of the variation of ICNC/CDNC and the way number concentrations of CCN and INP vary. Hence, findings particularly about relations between ICNC/CDNC and IWC/LWC in this study should be considered preliminary ones that initiate future work to streamline the development of the general parameterizations.

- Lines 197 to 203: I do not understand the logic, the authors refer to hydrometeors, then to aerosols, then back to aerosols. Can the authors rephrase the sentence?

The corresponding sentence is revised as follows:

(LL196-205 on p7)

Size distribution functions for each class of hydrometeors, which are classified into water drops, ice crystals (plate, columnar and branch types), snow aggregates, graupel and hail, are represented with 33 mass doubling bins, i.e., the mass of a particle $m_k$ in the kth bin is determined as $m_k = 2m_{k-1}$. Each of hydrometeors has its own terminal velocity that varies with the hydrometeor mass and the sedimentation of hydrometeors is simulated using their terminal velocity.
        Size distribution functions for aerosols, which act as cloud condensation nuclei (CCN) and ice-nucleating particles (INP), adopt the same mass doubling bins as for hydrometeors.

- Line 205: So aerosols do not sediment, is this true? Can the authors confirm that this hypothesis is close to reality?

The settling speed of aerosol particles in the accumulation mode, which is the most important mode among aerosol modes in terms of droplet or ice-crystal nucleation, ranges from 0.5 to 1 cm hr$^{-1}$, according to Seinfeld and Pandis (1997). This speed is negligible as compared to the terminal speed of droplets which is on average ~4-6 orders of magnitude greater than the settling speed of aerosols in the accumulation mode. Hence, we consider the settling speed of aerosols negligible as compared to the terminal speed of droplets, and based on this, we set the settling speed of aerosols at zero, while allowing droplets to fall down with its own terminal speed.

- Line 247: Since the altitude bins are not linear, would it be possible to have them in the supporting information? Otherwise it would be difficult to replicate the study.

The information is included in the supplement.

- Line 266 and line 281: Aerosol concentration is assumed to be constant with time and altitude in the PBL, can the authors confirm that this is an appropriate hypothesis?

The time-averaged observed aerosol concentration is imposed on the simulations as an initial aerosol concentration. During the simulation period, the imposed concentration of aerosols, including those in the PBL, is not fixed with respect to time and space but varies not only with time but also with the horizontal and vertical domains via processes as described in Section 2.1.

We do not have the observation of aerosol vertical distributions. Hence, we have no choice but to assume that aerosol number concentrations do not vary up to the PBL top and then above the PBL top, they reduce exponentially only at the first time step, based on the numerous previous observational studies (e.g., Gras, 1991; Jaenicke, 1993; Seinfeld and Pandis, 1998) that support this assumption. Since there is good consistency between modeling results and observation as described in Section 3.1.1, we believe that the assumption about aerosol concentrations is not that unreasonable.

The corresponding text is revised as follows by including related references and putting "at the frist time step" in the text:

(LL310-316 on p11)

This study takes an assumption that the interpolated CCN concentrations do not vary with height in a layer between the surface and the planetary boundary layer (PBL) top around 1 km in altitude at the first time step, following the previous studies such as Gras (1991), Jaenicke (1993) and Seinfeld and Pandis (1998). However, above the PBL top, they are assumed to decrease exponentially with height at the first time step, based on those previous studies, although the shape of size distribution and composition do not change with height.

To clarify the evolution of aerosol concentration, the following is added:

(LL308-310 on p11)

Note that although these parameters or the shape of aerosol size distribution does not vary, associated aerosol concentrations vary over the simulation domain and period via processes as described in Section 2.1.

- Line 291: The dust concentration is higher than the observed INP concentration, as is the INP concentration from the presentation. Can the authors state whether the INP concentration from the study is of the same order of magnitude as the dust events?

Yes, according to Hartmann et al. (2021), the INP concentration in this study is at the same order of magnitude as in the strong dust events. Based on this, the corresponding text is revised as follows:

(LL322-325 on p11)

However, Hartmann et al. (2021) observed the INP concentration that was at the same order of magnitude as assumed here in the Svalbard area when strong dust events occur, meaning that the assumed INP concentration is not that unrealistic.

- Line 385, is the temperature range from Choi et al. (2014) at the top of the cloud? Also, is the temperature range the same as the temperature range from the present study?

The temperature range in Choi et al. (2014) is based on the measurements of temperature throughout the cloud layer that includes the cloud top, cloud base and cloud parts between the cloud top and base.

The temperature range in Choi et al. (2014) is from 0 °C to -50 °C and this range includes the temperature range, which is between -16 and -33 °C, in the cloud layer in the case selected and simulated in this study. We picked up the supercooled cloud fraction (SCF) at the temperature between -16 and -33 °C from Choi et al. (2014) and the SCF picked up is shown in text.

- Line 462: The authors refer to the Lee et al. (2022) study for the mid-latitude case. Are the model and method similar and is CTH similar? I think the authors should describe this paper and the method used a bit more.

The simulated average CTH is 2.2 km in the midlatitude case and 3.3 km in the polar case. Both cases are simulated using an identical model, identical vertical resolutions, an identical source of reanalysis data for background conditions such as temperature, humidity and wind. However, the horizontal resolution is 100 m for the midlatitude case and 50 m for the polar case. To indicate this, the following is added:

(LL520-527 on p18)

In addition, identical model, model setup such as vertical resolutions, and source of reanalysis data are used between the 200_2 and control-midlatitude runs, although there are differences in environmental conditions (e.g., temperature), cloud macrophysical variables such as cloud-top height and horizontal resolutions between the runs. Here, while taking these similarities and differences into account, we hypothesize that the significant differences in ICNCavg/CDNCavg between runs are mainly due to the fact that ice nucleation strongly depends on air temperature (Prappucher and Klett, 1978).

Just want to say that there are similarities and differences in model setup (i.e., resolutions) and in CTH. However, this study focuses mostly on discrepancies in ICNC/CDNC and IWC/LWC, and through sensitivity tests, this study aims to understand how these discrepancies in ICNC/CDNC lead to those in IWC/LWC. The aim of this study is not in the examination of the cause of discrepancies in ICNC/CDNC between the cases, as detailed in our response to one of general comments above, although this study mentions temperature as a key factor causing the discrepancies in ICNC/CDNC. Factors like not only aerosol sources and advection as mentioned in "summary and conclusions" but also model setup and CTH can act as other factors than temperature causing the discrepancies in ICNC/CDNC. However, the examination of the cause of discrepancies in ICNC/CDNC between different cloud systems is out of scope of this study. We believe that this examination of the cause of discrepancies in ICNC/CDNC between different cloud systems requires significant research efforts, equivalent to those exerted on this paper, and can be done via additional future studies.

- Line 478-482. The much higher IWC than LWC in the polar case compared to midlatitude. The polar case is also associated with lower temperature, so I am not sure what more information can be derived from this.

As mentioned in our response above, while this study utilized temperature differentials to identify cases exhibiting significant disparities in ICNC/CDNC that control significant disparities in IWC/LWC, its primary objective lies in comprehending the inherent role of ICNC/CDNC variations themselves in the discrepancies observed in IWC/LWC across diverse cloud systems, regardless of the cause of those disparities in ICNC/CDNC. As mentioned in our response above, these disparities in ICNC/CDNC and IWC/LWC can be caused by other factors such as aerosol sources and advection than temperature. With the fulfillment of this objective, as long as we take interest in the role of ICNC/CDNC variations themselves among different cloud systems, the insights gleaned from this study regarding the influence of varying ICNC/CDNC on IWC/LWC can be extrapolated to scenarios where factors other than temperature differences contribute to discrepancies in ICNC/CDNC among different cloud systems.

- Line 495: The authors present new simulations here, which is confusing, why did they not present them in section 2.2.2?

Section 2.2.2 briefly introduces simulations, including new simulations pointed out by the reviewer here, to readers, so, they can get a glimpse of simulations performed for this study. Then, in the following sections, we describe and interpret results and based on this interpretation, we raise issues or hypothesis, for example, as in Section 3.1.3. To resolve the issues or hypothesis, we perform simulations and their results are described, for example, as in Section 3.1.4. We believe that it should be after presenting results and after finding issues or hypothesis to tackle from the results as in Section 3.1.3, we present simulations, tackling the issues or hypothesis, as in Section 3.1.4. In summary, readers can get a brief recognition of new simulations in Section 2.2.2, and then gain an understanding of why simulations are performed in Section 3.1.3 before getting to the simulations tacking the issues and hypothesis in Section 3.1.4. Hence, moving simulation results, as described in Section 3.1.4, to Section 2.2.2 removes chances for readers to understand why those simulations are performed and thus to better figure out implications of the results of those simulations. Thus, we believe that letting the presentation of the simulations pointed out by reviewer here stay in Section 3.1.4 is better for readers.

- The process associated with cloud formation and evolution does indeed depend on geographical regions. The Arctic is known to be prone to radiative cooling, whereas the mid-latitudes are prone to convective processes, so it is difficult to compare the two regions. Can the authors comment on this?

The simulations with and without radiative processes for the polar (or the Arctic) case in this study demonstrate that radiative cooling does not drive results from the polar case as in the midlatitude case. Hence, if we limit the driving force to two types, which are radiative cooling and convective processes as phrased by the reviewer here, there are no differences in the type of the driving force between the polar and midlatitude cases.

Anyway, this study aims to understand the substantial difference in IWP/LWP between the polar and midlatitude cases. For this understanding, we pick up and focus on ICNC/CDNC among many potential factors and examines its role in the difference in IWP/LWP between the cases. Yes, as mentioned in our responses above, there can be differences in other factors than in ICNC/CDNC which can explain the difference in IWP/LWP between different cloud systems. For the more fully established general principle, we need to carry out more studies to examine the roles of differences in other potential factors in the difference in IWP/LWP between different cloud systems in the future.

As mentioned in the manuscript, even attempts to identify a general factor to explain differences in IWP/LWP between different cloud systems have been rare and thus, findings in this study can be a valuable steppingstone to the fully established general principle, although this study deals with one factor, which is ICNC/CDNC.

- Line 597: 200_2_fac10, 100_2_fac10_CCN10, 200_2_fac10_INP10 represent mid-latitude cases. I thought the mid-latitude cases were from Lee et al. (2022) (line 159). Can the authors explicit that?

As mentioned in our responses above, this study focuses on only one factor, which is ICNC/CDNC, and tries to examine it as thoroughly as possible in terms of its role in the differentiation of IWP/LWP between the runs, as a steppingstone to the more general principle. Based on this, the 200_2_fac10 run in the old manuscript or 200_0.07 run in the new manuscript represents the mid-latitude case in Lee et al. (2022) or the control-midlatitude run in the sense that the 200_2_fac10 and control-midlatitude runs have similar ICNC/CDNC, although as mentioned in our responses above, there can be differences in factors, such as temperature, cloud-top height and model setup, between the 200_2_fac10 and control-midlatitude runs.

- Line 786: Should IFN be INP?

Corrected

First of all, we appreciate the reviewer's comments and suggestions. In response to them, we have made relevant revisions to the manuscript. Listed below are our answers and the changes made to the manuscript according to those comments and suggestions. Each comment of the reviewer (black) below is followed by our response (blue).

The authors have tried to make answers to my questions in the previous review. While some of them are adequately answered, others are not yet. Moreover, I would like to add some minor comments.

Major
1. The question in the previous review that points out the causality between the sedimentation and the total cloud mass is not properly answered. As sedimentation obviously acts as a sink term of cloud mass, the sentence "the droplet sedimentation tends to increase the total cloud mass in the 200_2 run" should be revised. In other words, to state the drop sedimentation as a cause of the increase in the total cloud mass, neither just mentioning some previous studies nor simply saying "it is not important" is not sufficient. The sentence should be revised like as: "Of the two runs, the drop sedimentation is greater in the case with the greater total cloud mass". I can find similar expressions at many points throughout the manuscript.
Related with this, I do not understand the phrase "changing sedimentation" in L527. Does changing mean increasing or decreasing? Also, I do not think that neither increasing nor decreasing of total cloud mass cannot be yielded by "changing sedimentation".

Following the comment here, sentences related to comments here are removed or revised. For details, see the revised text in Sections 3.1.2, 3.1.4 and 3.2.1.

2. By conducting additional experiments, the question that points out the importance of ICNC/CDNC is answered to some extent. However, it is still questionable whether ICNC/CDNC is indeed the critical factor. The experiments in the manuscript do not show any "intermediate" IWC/LWC values, saying 2–20 although ICNC/CDNC increases relatively gradually (Table 4). This means that the IWC/LWC shows an abrupt change, like a regime shift. I strongly recommend that the authors should clarify this in a relevant point of the manuscript.

To respond to this comment, the following is added:

(LL803-814 on p27)

The high-degree non-linearity in the variation of IWC/LWC is epitomized by the 1706 percent increase in IWC/LWC for the 163 percent increase in ICNCavg/CDNCavg from the 200_0.7 run to the 2000_2 run. This 1706 percent increase in IWC/LWC is induced by increases in both the initial number concentrations of CCN and INP between the runs (Table 1). In other transition from a simulation in a row to that in the next row in Table 4, there are decreases in both the initial number concentrations of CCN and INP, or there is either a change in the initial number condensation of CCN or INP. When either the initial concentration of CCN or INP changes in the transition, less than a 100% increase in IWC/LWC is shown. The decreases in both the initial number concentrations of CCN and INP, which are from the 2000_20 run to the 200_2 run, result in the decrease in IWC/LWC. Hence, depending on how the initial number concentrations of CCN and INP change, the magnitude and sign of the change in IWC/LWC can vary substantially.

3. I strongly suggest the authors to show more diverse types of analysis in the manuscript (not as supplementary). For example, the authors should add time-height plots of IWC or time series of IWP. For L585, the authors can add drop (ice) size distributions as the authors utilized a spectral bin microphysics model. For the description in the Section 3.1.1, the authors should add vertical profiles, time series, or time-height plots of RH with respect to liquid and ice.

The following is added:

(LL396-402 on p14)

To provide additional information of cloud development, Figure 5 shows the time evolution of the simulated and observed cloud-top and bottom heights, simulated and retrieved IWP and simulated and observed LWP together with the evolution of the simulated surface sensible and latent-heat fluxes; the simulated evolutions in Figure 5 are from the 200_2 run. This is based on the fact that the cloud-top and bottom heights, IWP and LWP are considered a good indicative of cloud development and the surface fluxes are considered important parameters controlling the overall development of clouds.

(LL508-513 on p17)

Figure 7 shows the time series of the averaged supersaturation over gird points where deposition occurs in the presence of both droplets and ice crystals in the 200_2 run. Figure 7 indicates that on average, supersaturation occurs for both droplets and ice crystals over those grid points. Hence, on average, the above-described situation of qv > qsw is applicable to deposition when droplets and ice crystals coexist in the 200_2 run.

(LL646-657 on p22)

In Figure 10a, we see that the number concentration of ice crystals with diameters smaller and larger than ~40 micron increases and decreases, respectively, as we move from the 200_2 run to the 200_20 run, which indicate a shift of the sizes of ice crystals to smaller ones. From the 200_2 run to the 200_20 run, the sedimentation of droplets at the cloud base decreases as shown in Table 2, mainly due to decreases in LWC. Figure 10b shows that the number concentration of drops decreases throughout almost all parts of the size range from the 200_2 run to the 200_20 run, which indicates a negligible shift in the drop size but a reduction in LWC. It is found that changes in the average rates of the droplet and ice-crystal sedimentation over the cloud base and simulation period are ~three to four orders of magnitude smaller than those in the average integrated condensation and deposition rates between the 200_2 and 200_20 runs (Table 2).

Minor
1. In the manuscript, the authors should specify the model integration time and the period where the average is calculated. Note that the first a few hours of simulation are generally excluded from calculating average as they are regarded as spin-up in a typical idealized cloud simulation.

Simulations in this study are not idealized simulations for clouds, but real-case simulations which generally do not use the spin-up time, and this study does not adopt the spin-up period.

The model integration time is indicated as follows:

(LL254-256 on p9)

The simulation of the observed system or case, i.e., the control run, is performed three-dimensionally over the red rectangle and the period between 02:00 and 10:00 LST on March 29th, 2017.

(LL294-296 on p10)

Note that the average of a variable with respect to time in the rest of this paper is performed over this period between 02:00 and 10:00 LST, unless otherwise stated.

2. The authors should describe about the model in more detail. For example, does the model include the source of aerosol concentration other than turbulent mixing and advection, such as regeneration by drop evaporation or surface emission? Otherwise, how can the aerosol concentration be maintained throughout the model integration? In addition, whether the model consider freezing of activated drops should be described, considering the experimental condition. Moreover, the way large-scale subsidence (L254) is considered in the model should also be described.

Aerosol regeneration is considered in the model. To indicate this, the following is added:

(LL205-209 on p7-8)

The evolution of aerosol size distribution and associated aerosol concentrations at each grid point is controlled by aerosol sinks and sources such as aerosol advection, turbulent mixing, activation and aerosol regeneration via the evaporation of droplets and the sublimation of ice crystals. Aerosol regeneration follows the method similar to that as described in Xue et al. (2010).

Yes, the model adopted in this study considers not only the homogeneous freezing of drops but the heterogeneous freezing of drops via contact, immersion and condensation-freezing pathways, as already described in text as follows:

(LL224-230 on p8)

To represent heterogeneous ice-crystal nucleation, the parameterizations by Lohmann and Diehl (2006) and Möhler et al. (2006) are used. In these parameterizations, contact, immersion, condensation-freezing, and deposition nucleation paths are all considered by taking into account the size distribution of INP, temperature and supersaturation. Homogeneous aerosol (or haze particle) and droplet freezing is also considered following the theory developed by Koop et al. (2000).

The large-scale subsidence in the reanalysis data is imposed on the model to consider the large-scale subsidence in the simulations. This imposition of large-scale subsidence is similar to the well-known imposition of the field of horizontal winds in the reanalysis data on the model in the sense that both the large-scale subsidence and the field of horizontal winds are imposed on the model as "background wind fields". Similar to the situation where the imposed horizontal winds affect those generated by processes such as cloud processes, the imposed background large-scale subsidence suppresses updrafts and intensifies downdrafts generated by those processes. To clarify this, the following is added:

(LL268-270 on p9-10)

This large-scale subsidence is imposed on the control run as a part of background wind fields and interacts with updrafts and downdrafts generated by relatively small-scale processes including those associated with clouds.

3. L424 and others: How is the entrainment rate defined? Why does it have a unit of cm s−1? The entrainment rate has the unit of s−1 in general.

The entrainment rate can be approximated by the difference between the cloud top growth rate (or the rate of increase in cloud-top height) in cm s$^{-1}$ and the large-scale vertical velocity (or large-scale subsidence) in cm s$^{-1}$, following studies such as Moeng et al. (1999), Jiang et al. (2002), Stevens et al. (2003a and 2003b) and Ackerman et al. (2004).

The following is added:

(LL468-471 on p16)

Here, entrainment rate is defined to be the difference between the rate of increase in cloud-top height and the large-scale subsidence, following Moeng et al. (1999), Jiang et al. (2002), Stevens et al. (2003a and 2003b) and Ackerman et al. (2004).

4. L621: All of the studies mentioned consider drop autoconversion, not ice autoconversion. Therefore, it is not proper to presume that ice autoconversion is roughly proportional to ice concentration based on these studies.

To support ice autoconversion, proportional to ice concentration, Murakami (1990) and Morrison et al. (2005, 2009, 2012) are added.

5. Is stating INP meaningful in "noice" experiments? I suggest that the authors replace "200_2_noice" with "200_0" and "2000_2_noice" with "2000_0". In the same way, I suggest to replace

"200_2_fac10" with "200_0.07" (also the others including "_fac10") as the names are neither intuitive nor consistent with others.

The simulation names are changed following the comment here.

6. As the descriptions about the "norad" experiments are very simple, I suggest not to include them in Table 1 for simplicity.

Done.

7. The authors stated that the reanalysis has a temporal resolution of 6 hours (L248). However, in another line, they stated that Figure 2c shows the time series of surface temperature in the reanalysis (L257), which varies with a temporal resolution of 1 hour.

We find errors in a plotting program constructing the evolution of the surface temperature in Figure 2c.  A corrected figure using the corrected program is put into the manuscript.

8. I strongly suggest the authors to split the L343–391 and to make them as another subsection, named "model validation". Related with this, if there is no other problem, I suggest the authors to include Fig. S1 in the manuscript.

Done.

9. L214: the most widely used -> a

Corrected.

10. L303: IFN -> INP

Corrected.

11. L398, 400: The order of IWC and LWC should be reversed.

Corrected.

12. L405: this -> that, L406: this -> the

Corrected.

---

## Author Response (AR3)

First of all, we appreciate the editor's comments and suggestions. In response to them, we have made relevant revisions to the manuscript. Listed below are our answers and the changes made to the manuscript according to those comments and suggestions. Each comment of the editor (black) below is followed by our response (blue).

Dear authors,

Thank you for revising your manuscript according to the referee's comments and for thoroughly responding to their various concerns. I recognize that this required significant effort on your part, and I am pleased to note that the manuscript is now much improved. Adding and discussing observations in comparison to your modeling results was particularly important.

Both referees have provided additional comments based on your responses, and I would ask you to review these and further revise your manuscript accordingly. In particular, I would like to emphasize two specific points from these comments:

- Representation of the mixed phase in the model compared to the observation: Your assumptions do not seem to align with current literature based on observations. This should, at a minimum, be acknowledged and discussed in the manuscript.

A discussion, which is about the assumption adopted by the model used here for the mixing of droplets and ice crystals, is given in Section 4.4 in the revised manuscript:

- Relationship between IWC/LWC and ICNC/CDNC: This is a significant highlight of your paper, yet it is not clearly represented in any figures. As suggested by the referee, a visual representation of the relationship (e.g., one ratio as a function of the other) could be very useful.

We believe that Table 4 shows the relation between ICNC/CDNC and IWC/LWC well. Hence, based on Table 4, Figure 11 with ICNC/CDNC for x-axis and IWC/LWC for y-axis is added.

Additionally, I would like to provide a few more comments:

- The title of the manuscript is becoming too long and should be made more concise for better impact and readability. Please refer to our Guidelines for Authors (https://www.atmospheric-chemistry-and-physics.net/policies/guidelines_for_authors.html).

The title is changed in a way that the title is shorter yet more succinct, following this comment and a comment by one of the reviewers. In the new title, we put the emphasis on the fact that this study examines the role of a microphysical factor, which is ICNC/CDNC,

in the development of mixed-phase stratocumulus clouds and their interactions with aerosols.

Similarly, your conclusion has also increased in length, and I suggest adapting it to provide a more concise summary for the readers. Again, please refer to the guidelines. This could for instance be achieved by moving some content to your "discussion" section, which is comparatively shorter than the conclusion.

Some text in "Summary and Conclusion" in the old manuscript is moved to "Discussion". For this, Section 4.3 is created. See text for details.

- The "data availability" section should include a statement about where the data you used can be accessed (https://www.atmospheric-chemistry-and-physics.net/policies/data_policy.html). You mention that the data processed on your private computer system is private, but you have also utilized publicly available data. An access statement for this original data should be provided, particularly for CloudNet, and perhaps for the reanalysis and CCN data (if it is not based on publicly available data, please state so in that section).

The corresponding section is revised as follows:

(LL1015-1023 on p34)

Our private computer system stores private data such as the model code and output, and the CCN data. Upon approval from funding sources, the data will be opened to the public. Projects related to this paper have not been finished, thus, the sources prevent the data from being open to the public currently. However, if information on the data is needed, contact the corresponding author Seoung Soo Lee (slee1247@umd.edu).
    The Cloudnet and reanalysis data used in this study are publicly available. The Cloudnet data are obtainable at "https://cloudnet.fmi.fi/search/data", while the reanalysis data can be obtained by contacting Met Office via "https://www.metoffice.gov.uk/about-us/contact"

- As mentioned earlier, comparisons to observations are a crucial aspect of your work. However, I still find that insufficient details are provided regarding these observations. For example,

how many CloudNet stations are available in your simulated region,

There is only one station in the simulated region. This is indicated as follows:

(LL244-246 on p9)

These clouds are observed by a ground station which is a part of the Cloudnet observation network and marked by a dot in Figure 1.

and where are they located (this could be useful to include in Fig 1)?

The location of the station is indicated in Figure 1.

What is the spatial resolution of the retrievals? Are they continuously available accross your simulation or are there discontinuities?

For the retrieval of IWC and IWP, the spatial resolution of ~ 10 m in the vertical direction is used. The CloudNet data, including IWC and IWP data, are provided in a continuous way with a time resolution of 30 seconds. To indicate this, the following is added:

(LL253-256 on p9)

The retrieval of IWC is performed by using radar reflectivity and lidar backscatter in the Cloudnet observation with a spatial resolution of ~10 m in the vertical direction and a temporal resolution of 30 seconds as described in Donovan et al. (2001), Donovan and Lammeren (2001), Donovan (2003) and Tinel et al. (2005).

(LL262-264 on p9)

The Cloudnet observation data including these IWC, LWC, IWP and LWP data are provided to the public with a temporal resolution of 30 seconds in a continuous manner.

For Fig. 5 and related discussions, have you compared domain-averaged model outputs with the observations from the one or few station(s), or have you co-located the simulations with the observations?

As described in the figure caption, the simulated cloud-top height is averaged over grid points with cloud tops, and cloud-bottom height is averaged over grid points with cloud bottoms in the domain at each time step. As also described in figure caption, the simulated IWP and LWP are averaged over grid points with non-zero IWP and LWP, respectively, in the domain at each time step. These simulated and averaged variables are compared to the counterparts from one Cloudnet station in the domain.

To indicate that observations from one station are used in Figure 5, the following is added in the caption for Figure 5:

Observed and retrieved values are from the ground station as marked in Figure 1.

Additionally, have CloudNet data been used / evaluated before for mixed-phase clouds,

Yes, Cloudnet data have been used to evaluate the simulated clouds including mixed-phase clouds as exemplified by studies such as Illingworth et al. (2007) and Hansen et al. (2018). To indicate this, the following is added:

(LL389-391 on p13)

This study adopts the Cloudnet observation, which has been used to assess cloud simulations as in Illingworth et al. (2007) and Hansen et al. (2018), to evaluate the 200_2 run.

Hansen, A., Ament, F., Grutzun, V., and Lammert, A.: Model evaluation by a cloud classification based on multi-sensor observations, Geosci. Model Dev. Discuss., https://doi.org/10.5194/gmd-2018-259, 2018.

Illingworth, A. J., Hogan, R. J., O'Connor, E. J., et al.: Cloudnet - continuous evaluation of cloud profiles in seven operational models using ground-based observations, Bull. Am. Meteorol. Soc., 88, 883-898, 2007.

and do they provide simultaneous ice and liquid cloud properties in a single vertical pixel or only one at a time?

As mentioned in text, LWC and LWP are measured by radiometer, while IWC and IWP are retrieved based on the measurement by radar and lidar. Radiometer measures LWC with a time resolution of 30 seconds and a spatial resolution of ~50 m in the vertical direction, while the IWC retrieval is performed with a time resolution of 30 seconds and a spatial resolution of ~10 m in the vertical direction. Here, in the Cloudnet data, the IWC-retrieval resolution is used as a master grid on to which LWC dataset are interpolated for comparisons between IWC and LWC at a specific spatial location.

Due to the use of the identical temporal resolution, ice and liquid cloud properties are considered provided simultaneously. However, due to the use of the higher spatial resolution for ice cloud properties than for liquid cloud properties, ice cloud properties are provided in more vertical pixels than liquid cloud properties, and in general, there is inconsistency between the location of pixels where ice properties are retrieved and that where liquid properties are measured before the interpolation is performed. However, since high resolutions finer than 100 m are used for both the measurement of liquid properties and the retrieval of ice properties, we believe that the discrepancy in the location between ice and liquid properties is minimal.

To deliver points here, the related text is revised as follows:

(LL251-264 on p9)

In the Cloudnet observation, particularly, LWC is measured by radiometer with a spatial resolution of ~50 m in the vertical direction and a temporal resolution of 30 seconds. The retrieval of IWC is performed by using radar reflectivity and lidar backscatter in the Cloudnet observation with a spatial resolution of ~10 m in the vertical direction and a temporal resolution of 30 seconds as described in Donovan et al. (2001), Donovan and Lammeren (2001), Donovan (2003) and Tinel et al. (2005). In the retrieval, the lidar signal and radar reflectivity profiles are combined and inverted using a combined lidar/radar equation as a function of the light extinction coefficient and radar reflectivity. The combined equation is detailed in Donovan and Lammeren (2001). In the Cloudnet data, LWC data with the coarser spatial resolution than IWC data are interpolated to observation locations of IWC data, and IWP and LWP data are obtained from these IWC and interpolated LWC data, respectively. The Cloudnet observation data including these IWC, LWC, IWP and LWP data are provided to the public with a temporal resolution of 30 seconds in a continuous manner. This study utilizes these publicized Cloudnet data.

All these aspects should be further described and discussed to enhance the relevance of the model-observation comparisons.

Overall, both referees have suggested that minor revisions are needed. Considering that I am also providing my own list of comments in addition, I will request reconsideration of the manuscript after (moderate) major revisions. Please note that these revisions are primarily aimed at further clarifying your manuscript.

Kind regards,
Odran Sourdeval

First of all, we appreciate the reviewer's comments and suggestions. In response to them, we have made relevant revisions to the manuscript. Listed below are our answers and the changes made to the manuscript according to those comments and suggestions. Each comment of the reviewer (black) below is followed by our response (blue).

I thank the authors for the changes they made to the manuscript, which improved the article, especially in the presentation of the results. As mentioned in my previous review, I think the paper is suitable for publication in ACP. However, I have minor comments that I would like the authors to address before the article is published.

I am still concerned about the representation of the mixed phase in the model compared to the observation. I do not understand what the authors mean by ice and liquid pockets in the sentence: "pockets of ice particles and those of liquid particles are mixed together". What is the size of the pockets considered? Usually, in models, ice and liquid hydrometeors are homogeneously mixed in mixed phase clouds. My problem is that this is not what observations show, as in Coopman and Tan (2023), Tan and Storelvmo (2016), Korolev and Milbrandt (2022), D'Alessandro et al. (2021), D'Alessandro et al. (2023), Schima et al. (2022). The different representation of homogeneously mixed hydrometeors or the presence of pockets has an impact on the microphysical processes (e.g., WBF...). Therefore, it should be highlighted in the article that the representation of mixed-phase clouds considered in the model of the study has hypotheses that do not agree with observations (the study that the authors refer to Lee et al. (2021) is also from models).

Coopman, Q., & Tan, I. (2023). Characterization of the Spatial Distribution of the Thermodynamic Phase Within Mixed-Phase Clouds Using Satellite Observations. Geophysical Research Letters, 50(24), e2023GL104977.

Korolev, A., & Milbrandt, J. (2022). How are mixed-phase clouds mixed?. Geophysical Research Letters, 49(18), e2022GL099578.

D'Alessandro, J. J., McFarquhar, G. M., Wu, W., Stith, J. L., Jensen, J. B., & Rauber, R. M. (2021). Characterizing the occurrence and spatial heterogeneity of liquid, ice, and mixed phase low-level clouds over the Southern Ocean using in situ observations acquired during SOCRATES. Journal of Geophysical Research: Atmospheres, 126(11), e2020JD034482.

D'Alessandro, J. J., & McFarquhar, G. M. (2023). Impacts of drop clustering and entrainment-mixing on mixed phase shallow cloud properties over the Southern Ocean: Results from SOCRATES. Journal of Geophysical Research: Atmospheres, 128(15), e2023JD038622.

Schima, J., McFarquhar, G., Romatschke, U., Vivekanandan, J., D'Alessandro, J., Haggerty, J., ... & Schnaiter, M. (2022). Characterization of Southern Ocean boundary layer clouds using airborne radar, lidar, and in situ cloud data: Results from SOCRATES. Journal of

Geophysical Research: Atmospheres, 127(21), e2022JD037277.

Tan, I., & Storelvmo, T. (2016). Sensitivity study on the influence of cloud microphysical parameters on mixed-phase cloud thermodynamic phase partitioning in CAM5. Journal of the Atmospheric Sciences, 73(2), 709-728.

To highlight that "the representation of mixed-phase clouds considered in the model of the study has hypotheses that do not agree with observations (the study that the authors refer to Lee et al. (2021) is also from models)", Section 4.4 is added in the revised manuscript.

Regarding my first specific comment, I suggest that the difference between stratiform and convective clouds be clarified and changed: "When mixed-phase stratiform clouds are associated with convective clouds, they can form even in the tropical region." to "When mixed-phase clouds are associated with convective clouds, they can form even in the tropical region."

Done.

I thank the authors for their efforts to remove the brackets, as mentioned in my previous review, but they should be careful that the sentences remain coherent. For example, the three repetitions of "respectively" in lines 122, 123, 124 make reading too difficult.

The text pointe out here is revised as follows:

(LL122-125 on p5)

This is because water vapor deposits on the surface of ice crystals, while it condenses on droplets. As a result, ice crystals act as sources of deposition and droplets act as sources of condensation. Consequently, ice crystals act as sources of IWC (or IWP) and droplets act as sources of LWC (or LWP).

Other text involving awkward repetitions of "respectively" is also revised. See text for this revision.

First of all, we appreciate the reviewer's comments and suggestions. In response to them, we have made relevant revisions to the manuscript. Listed below are our answers and the changes made to the manuscript according to those comments and suggestions. Each comment of the reviewer (black) below is followed by our response (blue).

The second review for the manuscript EGUsphere-2023-862 entitled "Examination of varying mixed-phase stratocumulus clouds in terms of their properties, ice processes and aerosol-cloud interactions between polar and midlatitude cases: An attempt to propose a microphysical factor to explain the variation" written by Lee et al..

The authors have adequately improved the manuscript. I suggest a few minor modifications before publishing.

1. Title: almost all contents of this manuscript are focused on the "polar case", instead of "the midlatitude case", with which was dealt by a previous study of the authors. I suggest the phrase "between polar and midlatitude cases" in the title simply to "a polar cloud case".

The title is changed in a way that it is shorter yet more succinct, following this comment and the editor's comment. In the new title, we put the emphasis on the fact that this study examines the role of a microphysical factor, which is ICNC/CDNC, in the development of mixed-phase stratocumulus clouds and their interactions with aerosols. Although as mentioned by the reviewer here, this study mainly talks about a polar case, its goal is to identify the role of the microphysical factor in the variation of cloud properties and their interactions with aerosols between different mixed-phase clouds at different geographical locations. Hence, we believe that as seen in the new title, not putting "polar" or "polar case" in the title makes it better represent this study.

2. I think that the most important result of this manuscript is the strong relationship between IWC/LWC and ICNC/CDNC, which is shown as a list of numbers in Tables 2, 3 and 5. I strongly suggest adding a figure that shows the relationship (e.g., ICNC/CDNC for x axis and IWC/LWC for y axis).

We believe that Table 4 shows the relation between ICNC/CDNC and IWC/LWC well. Hence, based on Table 4, Figure 11 with ICNC/CDNC for x-axis and IWC/LWC for y-axis is added.

3. L497: We cannot know whether pockets of ice particles and those of liquid particles are "mixed together" or "separated in a specific layer" from Figure 4, which just shows the horizontally averaged vertical mass contents profile of ice particles and liquid particles.

This comment is linked to the other reviewer's first comment which is on the mixing of liquid and ice particles. To respond to this comment and the other reviewer's first

comment, a discussion about an assumption of the mixing in the model used in this study is given in Section 4.4 in the revised manuscript.

4. L500–507: "higher", "more", "higher", "more", "higher", and "more" than what?

To remove confusion from the use of "higher" and "more", the corresponding text is revised as follows:

(LL516-526 on p17-18)

Thus, as ICNC/CDNC increases in a situation where qv > qsw, it is likely that the portion of water vapor, which is deposited onto ice crystals, increases. This is by stealing water vapor, which is supposed to be condensed onto droplets, from droplets in an air parcel. Here, qv and qsw represent water-vapor pressure and water-vapor saturation pressure for liquid water or droplets, respectively. As ICNC/CDNC increases in a situation where qsi< qv <qsw, the number of ice crystals, which absorb water vapor, increases per a droplet; here, water vapor absorbed by ice crystals includes that which is produced by droplet evaporation, and qsi represents water-vapor saturation pressure for ice water or ice crystals. Thus, as ICNC/CDNC increases, it is likely that the portion of water vapor, which is deposited onto ice crystals in an air parcel, increases as shown in Lee et al. (2021).

5. L507: I expect not only high ICNC/CDNC but also greater capacitance of ice particles than that of liquid drops can impact the more active growth of ice particles by vapor deposition.

To reflect the reviewer's point here, the following is added:

(LL526-527 on p18)

This is aided by the higher capacitance of ice crystals than that of droplets (Pruppacher and Klett, 1978).

6. Some paragraphs are too long and contain too many ideas so I suggest splitting them into two or more: for example, L479 and L533.

Long paragraphs are split into two or more paragraphs.